

# 1 Diurnal variation in the isotope composition of plant xylem

# 2 water biases the depth of root-water uptake estimates

Hannes P.T. De Deurwaerder [(1,2)*], Marco D. Visser [(2)], Matteo Detto [(2)], Pascal Boeckx [(3)],
Félicien Meunier [(1,4)], Liangju Zhao [(5,6)], Lixin Wang [(7)], Hans Verbeeck [(1)]
(1) CAVElab - Computational & Applied Vegetation Ecology, Faculty of Bioscience Engineering, Ghent
University, Ghent, Belgium
(2) Department of Ecology and Evolutionary Biology, Princeton University, Princeton, NJ, USA
(3) ISOFYS – Isotope Bioscience Laboratory, Faculty of Bioscience Engineering, Ghent University, Ghent,
Belgium
(4) Ecological Forecasting Lab, Department of Earth and Environment, Boston University, Boston,
Massachusetts, USA
(5) Shaanxi Key Laboratory of Earth Surface System and Environmental Carrying Capacity, College of
Urban and Environmental Sciences, Northwest University, Xi'an 710127, China
(6) Key Laboratory of Ecohydrology and Integrated River Basin Science, Northwest Institute of Eco-
Environment and Resources, Chinese Academy of Sciences, Lanzhou 730000, China
(7) Department of Earth Sciences, Indiana University-Purdue University Indianapolis (IUPUI),
Indianapolis, IN 46202, USA
*Correspondence to*: Hannes De Deurwaerder (Hannes_de_deurwaerder@hotmail.com)

## 21 Abstract

1. Stable isotopologues of water are a widely used tool to derive the depth of root water
uptake (RWU) in lignified plants. Uniform isotope composition of plant xylem water
(*i-$H_2O$-xyl*) along the stem length is a central assumption, which has never been properly
evaluated.



2. We studied the effects of diurnal variation in RWU, sap flux density and various other soil and plant parameters on $i$-$H_2O$-$xyl$ within a plant using a mechanistic plant hydraulic model and empirical field observations from French Guiana and northwestern China.

3. Our model predicts significant $i$-$H_2O$-$xyl$ variation arising from diurnal RWU fluctuations and vertical soil water heterogeneity. Moreover, significant differences in $i$-$H_2O$-$xyl$ emerge between individuals with different sap flux densities. In line with model predictions, field data show excessive $i$-$H_2O$-$xyl$ variation during the day or along stem length ranging up to 25.2‰ in $\delta^2H$ and 6.8‰ in $\delta^{18}O$, largely exceeding the measurement error range.

4. Our work show that the fundamental assumption of uniform $i$-$H_2O$-$xyl$ is violated both theoretically and empirically and therefore a real danger exists of significant biases when using stable water isotopologues to assess RWU. We propose to include monitoring of sap flow and soil water potential for more robust RWU depth estimates.

**Keywords**

Deuterium, Ecohydrology, Lianas, Root water uptake, Sap flow, Stable isotope compostion of water, Tropical trees, Water competition

## 1. Introduction

The use of stable isotope composition of water has greatly enhanced ecohydrology studies by providing insights into phenomena that are otherwise challenging to observe, such as depth of root water uptake (RWU) (Rothfuss & Javaux, 2017), below ground water competition and hydraulic lift (Hervé-Fernández *et al.*, 2016; Meunier *et al.*, 2017). Compared to root excavation, the technique is non-destructive, far less labor-intensive and informs on actual



RWU while excavation solely informs on root distribution and architecture. Moreover, its
flexibility allows use across multiple scales both spatial (i.e. individual to ecosystem) and
temporal (i.e. daily to seasonal; Dawson *et al.* 2002). The advantages and wide applicability of
this method make it a popular technique that pushes the boundaries of ecohydrology (Dawson
*et al.*, 2002; Yang *et al.*, 2010; Rothfuss & Javaux, 2017).

A variety of methods exist that infer RWU depth from the isotope composition of plant

xylem water (*i-H₂O-xyl*), but all rely on a direct relationship between the isotopic compositions
of plant xylem water and soil water (Ehleringer & Dawson, 1992). More precisely, all have two
key assumptions. The first is that the isotope composition of plant xylem water remains
unchanged during transport from root uptake to evaporative sites (e.g. leaves and non-lignified
green branches). Hence, isotope fractionation - processes that shift the relative abundance of
the water isotopolgues during root water uptake and water through non-evaporative tissues - is
neglected (Wershaw *et al.*, 1966; Zimmermann *et al.*, 1967; White *et al.*, 1985; Dawson &
Ehleringer, 1991; Walker & Richardson, 1991; Dawson *et al.*, 2002; Zhao *et al.*, 2016). Second,
all methods assume that xylem water provides a well-mixed isotope composition of water from
different soil layers: sampled xylem water instantaneously reflects the distribution and water
uptake of the roots independent of sampling time or height.

The first assumption is relatively well supported. Fractionation at root level has not

raised concerns for most RWU assessments using water isotopologues (Rothfuss & Javaux,
2017), with the exception of kinetic fractionation. Kinetic fractionation is a process driven by
the differences in molecule mass among the isotopologues that occurs only in extreme
environments (Lin & Sternberg, 1993; Ellsworth and Williams, 2007; Zhao *et al.*, 2016).
Similarly, isotopic fractionation of water within an individual plant, although possible, is not
considered a serious problem (Yakir, 1992; Dawson & Ehleringer, 1993; Cernusak *et al.*, 2005;
Mamonov *et al.*, 2007; Zhao *et al.*, 2016). However, the second assumption of time and space





invariance of the isotope composition of xylem water has, to our best knowledge, never been
assessed.

In principle, temporal variance in $i$-$H_2O$-$xyl$ within a plant during a day or along its height

can be expected on first principles. Here we hypothesize that it is in fact likely that various plant
physiological processes, ranging from very simple to more complex mechanisms could
influence within plant variance in $i$-$H_2O$-$xyl$ at short time scales. For instance, plant
transpiration during the course of the day is regulated by atmospheric water demand and leaf
stomata which have clear and well known diurnal patterns (Steppe & Lemeur, 2004; Epila *et*
*al.*, 2017). This results in changing water potential gradients within the soil-plant-atmosphere
continuum and therefore fluctuations in the depth RWU are also expected (Goldstein *et al.*,
1998; Doussan *et al.*, 2006; Huang *et al.*, 2017). Hence, as we expect plants capacity to take up
water at different soil layers to shift during the day, we should also expect diurnal variation in
the mixture of isotope composition from water taken up from various depths. As water moves
up along the xylem with velocity proportional to sap flow, different plants and species might
respond differently to diurnal variation in RWU. Therefore, from very basic principles we may
expect temporal variation in $i$-$H_2O$-$xy$ to propagate to different plant heights. As sap flux density
depends on plant hydraulic traits in relation to atmospheric water demand and soil moisture
gradient, this mechanism could make comparison of isotopic data among individuals and
species misleading.

In this study we provide a critical assessment of the assumption of $i$-$H_2O$-$xyl$ invariance

over time and along the length of plant stems. We test the hypothesis that major alterations in
the $i$-$H_2O$-$xyl$ along the length of lignified plants arise naturally during the day and that this
variation in $i$-$H_2O$-$xyl$ exceeds the expected measurement error. We test this hypothesis with a
twofold approach. First, we build a simple mechanistic model that incorporates basic plant
hydraulic realism. We use this model to specifically test that even rudimentary mechanistic





models of plant hydraulic functioning predict that diurnal changes in the soil-plant-atmosphere
continuum result in shifting mixtures of soil water absorption differing in isotope composition.
Second, we test whether the *i-H₂O-xyl* sampled at different plant heights or at different times
of the day show large variances with field observations from i) six Neotropical canopy trees
and six Neotropical canopy lianas sampled at different heights in French Guiana, and ii) high
temporal resolution *i-H₂O-xyl* data of 6 distinct plant species from the Heihe River Basin in
northwestern China (Zhao *et al.*, 2014).

**2.  Materials and Methods**
**2.1.  Part A: Modelling exploration**
**2.1.1.  Model derivation**
The expected *i-H₂O-xyl* at different stem heights within a tree during the course of the day can
be derived from plant and physical properties such as root length density, total fine root surface
area, water potential gradients and the isotope composition of soil water. We call this the
SWIFT model (i.e. Stable Water Isotopic Fluctuation within Trees). To derive the SWIFT
model, we first describe the establishment of *i-H₂O-xyl* entering the tree at stem base via a
multi-source mixing model (Phillips & Gregg, 2003). We subsequently consider vertical water
transport within the tree, which relates to the established sap flow pattern. Note that the model
presented here, focusses on deuterium but can easily be used to study stable oxygen
isotopologues. To ensure consistency and clarity in variable declarations we maintain the
following notation in the subscripts of variables: uppercase roman to distinguish the medium
through which water travels (X for xylem, R for root, S for soil) and lowercase for units of time
and distance (*h* for stem height, *t* for time and *i* for soil layer index). A comprehensive list of



variables, definitions and units is given in Table 1. A schematic representation of the model is
provided in Fig. 1a.

*i.*      *Isotope composition of plant xylem water at stem base.*

The deuterium isotope composition of xylem water ($\delta^2 H_{X,0,t}$) of an individual plant at

stem base (i.e. height zero; $h = 0m$; Fig. 1a) at time $t$, can theoretically be derived using the
multi-source mixing model approach introduced by Phillips & Gregg (2003). Considering a
root zone divided into $n$ discrete soil layers of equivalent thickness $\Delta z$, if the deuterium isotope
composition of soil water ($\delta^2 H_{S,i}$) in each soil layer is constant over time, a reasonable
assumption if the isotopic measurements are conducted during rain-free periods, $\delta^2 H_{X,0,t}$ can
be expressed as:
$$\delta^2 H_{X,0,t} = \sum_{i=1}^{n} f_{i,t} \cdot \delta^2 H_{S,i} \qquad\qquad \text{Eq. (1)}$$
where $f_{i,t}$ is the fraction of water taken up at the $i$[th] soil layer (Fig. 1a) defined as:
$$f_{i,t} = \frac{RWU_{i,t}}{\sum_{i=1}^{n} RWU_{i,t}} \qquad\qquad \text{Eq. (2)}$$
and $RWU_{i,t}$ is the net amount of water entering and leaving the roots at time $t$ in the $i$[th] soil layer
($RWU_{i,t}$ is defined positive when entering the root). The current representation of the model
assumes no water loss via the root system and no mixing of the extracted water from different
soil layers within the roots until the water enters the stem base. When tree capacitance is
neglected, the sum of $RWU_{i,t}$ across the entire root zone is equal to the instantaneous sap flow
at time $t$, $SF_t$:
$$SF_t = \sum_{i=1}^{n} RWU_{i,t} = \sum_{i=1}^{n} -k_i \cdot A_{R,i} \cdot \left[ \Psi_{X,0,t} - (\Psi_{S,i,t} - z_i) \right] \qquad\qquad \text{Eq. (3)}$$
Where $k_i$ is the plant specific total soil-to-root conductance, $\Psi_{X,0,t}$ is the water potential at the
base of the plant stem and $\Psi_{S,i,t}$ is the soil water matric potential (Fig. 1a). Total plant water

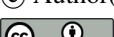



potential is generally defined as the sum of the pressure, gravity and matrix potential. Hence,
$\Psi_{X,0,t}$ represents the xylem pressure potential. The term $z_i$ is the gravimetric water potential
necessary to lift the water from depth $z_i$ to the base of the stem, assuming a hydrostatic gradient
in the transporting roots. The model considers $z_i$ to be a positive value (zero at the surface), thus
$z_i$ is subtracted from $\Psi_{S,i,t}$. $A_{R,i}$ is the absorptive root area distribution over soil layer $i$ (Fig. 1a).
This parameter can be derived from plant allometric relations (Čermák *et al.*, 2006) which is
subsequently distributed over the different soil layers via Jackson *et al.* (1995).
The total soil-to-root conductance is calculated assuming the root and soil resistances are
connected in series (Fig. 1a):

$$k_i = \frac{k_R \cdot k_S}{k_R + k_S}$$                Eq. (4)

where $k_R$ is the effective root radial conductivity (assumed constant and uniform), and $k_S =$
$K_{S,i}/\ell$ is the conductance associated with the radial water flow between soil and root surface.
$\ell = 0.53/\sqrt{\pi \cdot B_i}$ represents the effective radial pathway length of water flow between bulk soil
and root surface (De Jong van Lier *et al.*, 2008)(Vogel *et al.*, 2013). $B_i$ represents the overall
root length density distribution per unit of soil. $K_{S,i}$ is the soil hydraulic conductivity for each
soil depth. $K_{S,i}$ depends on soil water moisture and thus relates to the soil water potential $\Psi_{S,i,t}$
of the soil layer where the water is extracted. $K_{S,i}$ is computed using the Clapp & Hornberger
(1978) formulation:

$$K_{S,i} = K_{s,max} \cdot \left(\frac{\Psi_{sat}}{\Psi_{S,i,t}}\right)^{2+\frac{3}{b}}$$                Eq. (5)

where $K_{s,max}$ is the soil conductivity at saturation and $b$ and $\Psi_{sat}$ are empirical constants that
depend on soil type (here considered as constant through all soil layers).
Subsequently, $f_{i,t}$ can be restructured as:





$$f_{i.t} = \frac{k_i \cdot A_{R,i} \cdot \Delta\Psi_{i,t}}{\sum_{i=1}^{n} k_i \cdot A_{R,i} \cdot \Delta\Psi_{i,t}}$$ Eq. (6)
where the root to soil water potential gradient is represented as $\Delta\Psi_{i,t} = \Psi_{X,0,t} - (\Psi_{S,i,t} - z_i)$.
Combining Eq. (1) and Eq. (6) then allows derivation of $\delta^2 H_{X,0,t}$ as follows:
$$\delta^2 H_{X,0,t} = \sum_{i=1}^{n} \left( \frac{k_i \cdot A_{R,i} \cdot \Delta\Psi_{i,t}}{\sum_{j=1}^{n} k_j \cdot A_{R,j} \cdot \Delta\Psi_{j,t}} \cdot \delta^2 H_{S,i} \right)$$ Eq. (7)
This equation requires estimates of $\Delta\Psi_{i,t}$, which is preferably measured instantaneously in the
field (i.e. via stem and soil psychrometers for $\Psi_{X,0,t}$ and $\Psi_{S,i,t}$, respectively). However, as
measurements of $\Psi_{X,0,t}$ are not always available, estimated $\widehat{\Psi}_{X,0,t}$ can be derived from sap flow
by re-organizing Eq. (3) into:
$$\widehat{\Psi}_{X,0,t} = \frac{\sum_{i=1}^{n} [k_i \cdot A_{R,i} \cdot (\Psi_{S,i,t} - z_i)] - SF_t}{\sum_{i=1}^{n} k_i \cdot A_{R,i}}$$ Eq. (8)
which then allows replacement of $\Psi_{X,0,t}$ with $\widehat{\Psi}_{X,0,t}$ in Eq. (7).
*ii. Height-dependent isotope composition of plant xylem water*
In our model, from the stem base, the water isotopologues simply move upwards with
the xylem sap flow, hence diffusion and water fractionation during transportation are not
considered. The isotope composition in xylem water at height $h$ and time $t$ ($\delta^2 H_{X,h,t}$) is then the
isotope composition of xylem water at stem base at time $t - \tau$.
$$\delta^2 H_{X,h,t} = \delta^2 H_{X,0,t-\tau}$$ Eq. (9)
where $\tau$ is the lag before $\delta^2 H_{X,0,t}$ reaches stem height $h$ (Fig. 1a) which depends only on the
true sap flux density in the xylem ($SF_V$). True sap flux density indicates the real speed of vertical
water displacement within a plant, derived by dividing $SF_t$ over the lumen area of the plant ($A_x$;
Fig. 1a) i.e. the total cross-sectional area of the vessels. $\tau$ was derived from the mass
conservation equality:





$h \cdot A_x = \int_{t=0}^{\tau} SF_t \, dt$                                                      Eq. (10)
Note that since most scientific studies express sap flux density as the sap flow over the total
sapwood area ($SF_S$), rather than over the total vessel lumen area ($SF_V$), for consistency, we will
present the model outputs as functions of $SF_S$.
*iii.    Model parameterization and analyses*
We adopted the basic plant parameters from Huang *et al.* (2017) for a loblolly pine
(*Pinus taeda L.*) (Table S1). We started with synthetic sap flow patterns and volumes extracted
from the model runs of Huang *et al.* (2017) for a typical day (day 11 of the 30 days sequence),
and assumed no variation between days. Sap flow follows the plant's water demand which is
the result of daily cycles of transpiration driven by photosynthetic active solar radiation (PAR),
vapor pressure deficit (VPD) and optimal stomatal response (Epila *et al.*, 2017). Secondly, both
the soil water potential ($\Psi_{S,i,t}$) and deuterium isotope composition of soil water ($\delta^2 H_{S,i}$)
profiles with soil depth were adopted from Meißner *et al.* (2012) (Fig. S2, see Table S1 for
equations) and were assumed to stay constant over time. Since measurements of Meißner *et al.*
(2012) are derived from a silt loam plot in the temperate climate of central Germany, soil
parameters were selected accordingly from Clapp & Hornberger (1978). Subsequently, the
following model simulations were executed (see Fig.1a):
1) **Analysis A1: impact of temporal SF$_t$ variation on the isotope composition of**
**xylem water at a fixed stem height.** Temporal patterns in deuterium isotope
composition in xylem water ($\delta^2 H_X$) were evaluated for a typical situation, i.e.
measurement at breast height (*h*=1.30 m), conforming to standard practice of RWU
assessment.



2) **Analysis A2: impact of temporal SF$_t$ variation at different tree heights.** Temporal patterns in $\delta^2 H_{X,i}$ within a tree at various sampling heights (5, 10 and 15 m).

3) **Analysis A3: impact of temporal SF$_t$ variation on the isotope composition of xylem water and the timing of sampling.** Representation of the profile of $\delta^2 H_X$ along the full height of a tree, measured at different sampling times (9:00, 11:00 and 13:00), with the standard parameterization given in Table S1.

4) **Analysis B: variation in $\delta^2 H_X$ due to differences in absolute daily average sap flow speed.** Diurnal patterns in the deuterium isotope composition of xylem water in trees that differ solely in daily averaged $SF_V$, which are set to 0.56, 0.28 and 0.14 m h$^{-1}$ (respectively corresponding to $SF_S$ values of 0.08, 0.04 and 0.02 m h$^{-1}$).

All parameters (e.g. RWU) of the four analyses are given in Table S1.

Model runs for each analysis were compared to a null model. The null model adopts the standard assumption of zero variation in $\delta^2 H_X$ along the length of the plant body. We used extraction protocol related measurement errors with an accepted maximum error range of 3‰ when water extraction recovery rates are higher than 98% (Orlowski *et al.*, 2013). In our null model, this is represented by a normal distribution with a mean of 0‰ and a standard deviation of 1‰ , i.e. N(μ=0‰, σ=1‰ ), which makes the probability of an error of ≥3‰ highly unlikely (p ≤ 0.0027). Analytical errors introduced by the measurement device, i.e. a Picarro (California, USA), are considered negligible relative to the extraction error. Note that $SF_V$, which normalizes sap flow over total vessel lumen area, is correlated with plant diameter at breast height (DBH) which enables comparison with field measurements without the need for explicit consideration of DBH in the model. SWIFT was implemented in R version 3.4.0 (R Core Team, 2017), and is publicly available (see GitHub repository HannesDeDeurwaerder/SWIFT).





### 2.1.2. Estimation of rooting depth

RWU depths were derived from the simulated $\delta^2 H_X$ values by use of both the direct inference method and the end-member mixing analysis method, together representing 96% of the applied methods in literature (Rothfuss & Javaux, 2017). We refer readers to Rothfuss & Javaux (2017) for a complete discussion of these techniques. Here, average rooting depth is assumed to be the depth obtained by relating the simulated $\delta^2 H_X$ with the $\delta^2 H_{S,i}$ depth profile. We compared rooting depth estimates from simulated $\delta^2 H_X$, as described in the analyses above, with the true average rooting depth. The true average rooting depth was defined as the depth corresponding to the daily weighted average $\delta^2 H_X$, calculated as the weighted sum of $\delta^2 H_{X,i,t}$ and the relative fraction of water taken up at each depth.

### 2.1.3. Sensitivity analysis

We performed two sensitivity analyses to assess the relative importance of all parameters in generating variance in $\delta^2 H_X$ along the length of a plant. In both sensitivity analyses, we varied model parameters one-at-the-time to assess the local sensitivity of the model outputs. The sensitivity analysis provides insight into the design of field protocols, revealing potential key measurements in addition to any caveats.

We first assessed model sensitivity to (bio)physical variables by modifying model parameters of soil type, sap flow and root properties as compared to the standard parameterization (given in Table S1). The following sensitivity analyses were considered:

**Soil type:** The soil moisture content over all soil layers ($\theta_{S,i,t}$) can be deduced from the considered Meißner et al. (2012) $\Psi_{S,i,t}$ profile (see Fig. S2 and Table S1) using the Clapp & Hornberger (1978) equation:



$$\theta_{S,i,t} = \theta_{sat} \cdot \left(\frac{\Psi_{S,i,t}}{\Psi_{sat}}\right)^{-1/b}$$ Eq. (11)
Where $\theta_{sat}$, $\Psi_{sat}$ and $b$ are soil-type specific empirical constants that correspond to
sandy loam soil textures in the standard model parameterization (Clapp & Hornberger,
1978). The derived soil moisture profile ($\theta_{S,i,t}$), in turn, then provides a basis to study
the impact of other soil textures. A new soil texture specific $\Psi_{S,i,t}$ profile can then be
deduced by using $\theta_{sat}$, $\Psi_{sat}$ and $b$ values corresponding to different soil texture types
(values from Table 2 of Clapp & Hornberger (1978)). This enabled us to study $\Psi_{S,i,t}$
profiles for four distinct soil types, i.e. (i) sand, (ii) loam, (iii) sandy clay and (iv) clay
soils, in relation with the original silt loam $\Psi_{S,i,t}$ profile.
**Volume of water uptake:** We varied the total diurnal volume of water taken up by the
tree. New $SF_t$ values are scaled using algorithms from literature that provide an estimate
of the daily sap flow volume of a tree based on its DBH (Andrade *et al.*, 2005; Cristiano
*et al.*, 2015).
**Root conductivity:** We varied the root membrane permeability ($k_R$) to match multiple
species specific values found in literature (Sands *et al.*, 1982; Rüdinger *et al.*, 1994;
Steudle & Meshcheryakov, 1996; Leuschner *et al.*, 2004).
The second set of sensitivity analyses test the impact of root hydraulics, sap flux density
and sampling strategies on the sampled $\delta^2 H_X$. We obtained 1000 samples per parameter from
corresponding distributions and ranges (given in Table S2) with a Latin hypercube approach
(McKay *et al.*, 1979; McKay, 1988). This is a stratified sampling procedure for Monte Carlo
simulation that can efficiently explore multi-dimensional parameter space. In brief, Latin
Hypercube sampling partitions the input distributions into a predefined number of intervals
(here 1000) with equal probability. Subsequently, a single sample per interval is extracted in an





effort to evenly distribute sampling effort across all input values and hence reduce the number
of samples needed to accurately represent the parameter space.

### 2.2. Part B: Empirical exploration

#### 2.2.1. Data on variation in i-H$_2$O-xyl with plant height

We used data for six canopy trees and six canopy lianas sampled on two subsequent dry days
(24-25 August, 2017) at the Laussat Conservation Area in Northwestern French Guiana. The
sampling site (05°28.604'N-053°34.250'W) lies approximately 20 km inland at an elevation of
30 m a.s.l. This lowland rainforest site has an average yearly precipitation of 2500 mm yr$^{-1}$
(Baraloto *et al.*, 2011). Average and maximum daily temperatures of respectively 30°C and
36°C were measured during the sampling period. Sampled individuals are located in the white
sands forest habitat (Baraloto *et al.*, 2011), on a white sandy ultisol with typically high
percentage of sand.
Individuals (Table 2) were selected based on assessment of climbable tree, intactness of
leafy canopy vegetation and close vicinity with one another to optimize similarity in
meteorological and edaphic characteristics. Liana diameters were measured at 1.3 m from the
last rooting point (Gerwing *et al.*, 2006), tree diameters were measured at 1.3 m (Table 2).
Sampling was performed between 9am and 2pm to assure high sap flow. Liana and tree
sampling allowed highly contrasted sap flux density (Gartner *et al.*, 1990).

#### 2.2.2. Sampling strategy

The stem xylem tissue of individual plants was sampled at different heights (1.3, 5, 10, 15 and
20 m where possible) at the same radial position of the stem, between 8:00 and 15:00. The order
of sampling, i.e. ascending versus descending heights, was randomized. Tree stem xylem



samples were collected with an increment borer (5 mm diameter), resulting in wooden cylinders
from which bark and phloem tissues were removed. Coring was performed within the horizontal
plane at the predefined heights, oblique to the center of the stem to maximize xylem and
minimize heartwood sampling, and slowly to avoid heating up the drill head and kinetic
fractionation. Taking one sample generally took between 5 and 10 minutes. Since coring lianas
was not possible, we collected cross-sections of the lianas after removing the bark and phloem
tissue with a knife. All materials were thoroughly cleaned between sampling using a dry cloth
to avoid cross-contamination. Upon collection, all samples were placed in pre-weighed glass
collection vials, using tweezers, to reduce contamination of the sample. Glass vials were
immediately sealed with a cap and placed in a cooling box, to avoid water loss during
transportation.

### 317   2.2.3.  Sample processing

Sample processing was performed as in De Deurwaerder *et al.* (2018). Specifically, all fresh
samples were weighed, transported in a cooler and frozen before cryogenic vacuum distillation
(CVD). Water was extracted from the samples via CVD (4 h at 105°C). Water recovery rates
were calculated from the fresh weight, weight after extraction and oven dry weight (48 h at
105°C). Samples were removed from the analysis whenever weight loss resulting from the
extraction process was below 98% (after Araguás-Araguás *et al.*, 1998). The isotope
composition of the water in the samples was measured by a Wavelength-Scanned-Cavity Ring-
Down Spectrometer (WS-CRDS, L2120-i, Picarro, California, USA) coupled with a vaporizing
module (A0211 High Precision Vaporizer) through a micro combustion module to avoid
organic contamination (Martin-Gomez *et al.*, 2015; Evaristo *et al.*, 2016). Internal laboratory
references were used for calibration, with measurement precision of ±0.1‰ and ±0.3‰ for $\delta^{18}O$





and $\delta^2H$, respectively. Post-processing was performed using SICalib (version 2.16; Gröning,

2011)

Isotopic composition, expressed in terms of $[^{18}O]/[^{16}O]$ and $[^{2}H]/[^{1}H]$ ratios, is

represented by $\delta$-values (in our case, $\delta^{18}O$ and $\delta^2H$), which indicate the deviation from a
designated standard (i.e. V-SMOW, Vienna Standard Mean Ocean Water) in parts per thousand
(expressed in ‰):

$$\delta_{sample(‰)} = \left(\frac{R_{sample}}{R_{standard}} - 1\right) \cdot 1000$$    Eq. (12)

where $R$ is the heavy to light isotope ratio measured in the sample or standard. We calculate
normalized *i-H₂O-xyl* for each individual at every sampled height $h$ ($\varepsilon^2H_X$ and $\varepsilon^{18}O_X$) as being
the deviation of each sample from the stem mean (derived from all stem samples of that
individual):

$$\varepsilon^2H_{X,h} = \delta^2H_{X,h} - \frac{1}{N}\sum_{h=1}^{N}\delta^2H_{X,h}$$    Eq. (13)

with $N$ the number of heights sampled per individual.

**2.2.4. Data on high temporal-resolution variation in *i-H₂O-xyl***
We used data from three extensive field campaigns by Zhao *et al.* (2014) who sampled plant *i-*
*H₂O-xyl* at high temporal resolution in the Heihe River Basin (HRB), northwestern China. Four
distinct study locations differing in altitude, climatological conditions and ecosystem types
were selected. At each location, the dominant tree, shrub and/or herb species was considered
for sampling. In August 2009, *Populus euphratica* was sampled in the Qidaoqiao riparian forest
(42°01'N-101°14'E) and *Reaumuria soongorica* in the Gobi desert ecosystem (42°16'N-
101°17'E; 906-930 m a.s.l). In June–September 2011 *Picea crassifolia*, *Potentilla fruticose,*
*Polygonum viviparum* and *Stipa capillata* were measured in the Pailugou forest ecosystem
(38°33'N-100°18'E; 2700-2900 m a.s.l). All species were samples 2-hourly, with the exception



of P. crassifolia which was measured hourly. Stem samples were collected for trees and shrubs,
while root samples were obtained for the herb species (more details in Zhao *et al.* (2014)).
Upon collection, all samples were placed in 8 mL collection bottles and frozen in the
field stations before transportation to the laboratory for water extraction via CVD (Zhao *et al.*,
2011). Both $\delta^{18}O$ and $\delta^2H$ were assessed with an Euro EA3000 element analyzer (Eurovector,
Milan, Italy) coupled to an Isoprime isotope ratio mass spectrometer (Isoprime Ltd, UK) at the
Heihe Key Laboratory of Ecohydrology and River Basin Science, Cold and Arid Regions
Environmental and Engineering Research institute. Internal laboratory references were used for
calibration, resulting in measurement precision of ±0.2‰ and ±1.0 ‰ for $\delta^{18}O$ and $\delta^2H$,
respectively.

**3. Results**
**3.1. Part A: Modelling exploration**
**3.1.1. Simulated temporal fluctuation in isotope composition of plant xylem water**
*i. Isotope composition of xylem water at stem base and basic model behavior*
At the stem base, simulated $\delta^2H_{X,0,t}$ displays a diurnal fluctuation (Fig. 2) that corresponds to
the daily sap flow pattern (Fig. 2c). This pattern is caused by shifting diurnal RWU depth. Early
in the morning, when transpiration is low, most of the RWU occurs in deeper layers, where soil
water potential is less negative and isotopic composition of soil water is dominated by depleted
deuterium (Fig. S2a-b). As transpiration increases during the day, a significant proportion of
RWU is extracted from the drier shallow layers, which have an enriched isotopic composition.
In the afternoon, as transpiration declines, isotopic composition reflects again the composition
of the depleted deep soil and it remains constant throughout the night because SWIFT does not
consider mixing of the internal water in stem and roots nor hydraulic lift.



The most enriched $\delta^2H_X$-values (approx.-59‰) are found in alignment with the diurnal

minimum of $\Psi_{X,0,t}$ (approx.-0.85 MPa, Fig. 2c). At this moment, $\Delta\Psi_{i,t}$ are maximized, enabling

water extracting from the upper and driest soil layers. Most root biomass is located near the

surface (cf. Jackson *et al.*, 1995; Fig. S2c) and uptake in these layers will result in relatively

high contributions to the total RWU.

In contrast, $\Delta\Psi_{i,t}$ are smaller in the early morning and late afternoon causing root water

uptake in the upper soil layers to halt. The decreasing $\Delta\Psi_{i,t}$ translates into higher proportions of

RWU originating from deeper, more depleted soil layers. This causes $\delta^2H_X$ to drop to a baseline

of approx. -67‰. This afternoon depletion of $\delta^2H_X$ will henceforth be indicated as the $\delta^2H_X$-

baseline drop.

*ii.     Isotope composition of xylem water at different times, heights and $SF_V$*

Temporal fluctuation in $\delta^2H_X$ within a tree at 1.3 m (i.e. the standard sampling height;

Analysis A1; Fig. 1a) and at other potential sampling heights (e.g. branch collection; Analysis

A2; Fig. 1a), are provided in Fig. 2a-b, respectively. Both analyses show that fluctuations in

$\delta^2H_X$ depend on the height of measurement and the corresponding time needed to move the

water along the xylem conduits. Note that it depends on the selected temporal resolution

whether the $\delta^2H_X$-baseline drop at a given height equals the (stem base) minimum (here 1 min,

see Fig. S6). The relation between $\delta^2H_X$ variance and cumulative sap flow volumes is provided

in Fig. 2d. Here, the piston flow dynamics in SWIFT originate from lateral translation of the

$\delta^2H_X$ fluctuation at $\delta^2H_{X,0,t}$. In addition to sampling height, analysis A3 depicts the importance

of sampling time (Fig. 1b).

Analysis B outputs predict the occurrence and width of the $\delta^2H_X$-baseline drop as a function

of $SF_V$ (Fig. 1c). Moreover, depending on $SF_V$, the isotopic signal can take hours or days to

travel from roots to leaves - as is also observed experimentally (Steppe *et al.*, 2010). Low $SF_V$



allows multiple $\delta^2H_X$-baseline drops over the length of a single tree. This means that sampled
$\delta^2H_X$ can reflect soil isotopic composition of the past several days. This has direct implications
for comparing samples obtained at different times and heights and for species that experienced
different $SF_V$ histories.

### 3.1.2.  Potential biases in root depth estimation

Both timing of measurement (Fig. 3a) and $SF_V$ (Fig. 3b) influence rooting depth estimates
derived via the direct inference and end-member mixing analysis method (Fig. S2) (Rothfuss
& Javaux, 2017). Collection of tree samples at 1.30 m can result in erroneous estimation,
deviating up to 104% from the average daily RWU depth (Fig. 3). Plotting the relative error in
RWU depth as a function of time and $SF_V$ (Fig. 3c) shows that it is possible to time $\delta^2H_X$
measurements in a fashion that captures unbiased estimates of the average RWU depth. Xylem
water sampling should be timed to capture the $\delta^2H_X$ that corresponds to water extracted at peak
RWU, and the expected sampling time can be derived by considering the time needed for the
water to reach the point of measurement (i.e. at 1.30 m in Fig. 3). In general, SWIFT predicts
that plants with slow $SF_V$ should not be measured during the morning hours, as this results in
measuring the preceding days' absorbed water. In contrast, trees with higher $SF_V$ support earlier
sample collection.

### 3.1.3.  Sensitivity analysis

Our sensitivity analyses shows that the expected absolute error in RWU depth assessment is
directly related to both 1) maximum variance in and 2) the probability of sampling non-
representative $\delta^2H_X$ values. The maximum variance depends on the height, while the probability
of sampling non-representative areas depends on the width of the "$\delta^2H_X$-baseline drop"



respectively (defined above). Hence, bias in $\delta^2 H_X$ is predominantly a function of the sampling
strategy (timing and height of sampling; Fig. S3) in relation to the $SF_V$ of the plant (shown by
a strong effect of lumen area and total diurnal RWU volume in Fig. S3) and some biophysical
parameter (Fig. S4). We summarized the most important variables as predicted by SWIFT, that
should be considering in RWU studies below.
Plants on loam soils show larger diurnal $\delta^2 H_X$ variances (~8‰) in comparison with those of
clay soils (~3‰). Larger variances correspond to potentially larger error, but the steeper slope
of the $\delta^2 H_X$ curve results in a thinner $\delta^2 H_X$-baseline drop. Hence, loam soil can result in
potentially the largest errors but this is mediated by a lower probability of sampling non-
representative $\delta^2 H_X$ values during the day.
The volume of water taken up by the plant ($SF_t$; Fig. S4b) affects xylem water potential of
the plant at stem base ($\widehat{\Psi}_{X,0,t}$). Higher $SF_t$ requires more negative $\widehat{\Psi}_{X,0,t}$, enabling the plant to
access more shallow and enriched soil layers. Therefore, an increase in $SF_t$ results in the
increase of maximum $\delta^2 H_X$ values (increased maximum error) but also results in a smaller width
of the baseline drop (Fig. 1c). Lower $SF_t$ result in smaller error, but larger probability of
sampling an non-representative area (Fig. 1c).
Root properties, i.e. root membrane permeability (Fig. S4c) strongly influence both the total
range of $\delta^2 H_X$ variance and the width of the $\delta^2 H_X$-baseline drops. Decreasing root permeability
results in thinner $\delta^2 H_X$-baseline drops, but higher maximum $\delta^2 H_X$ variance.

### 3.2. Part B: Empirical exploration

The observed normalized deuterium isotope composition in xylem water ($\varepsilon^2 H_X$) along the height
of lianas and trees showed strong intra-individual variance exceeding the null model by a factor



of 3.2 and 4.3 respectively (Fig. 4a-b). Specifically, differences up to 13.1‰ and 18.3‰ in $\delta^2H$
and 1.3‰ and 2.2‰ in $\delta^{18}O$ are observed as intra-individual variances for trees and lianas
respectively (Table 2).
Similarly, excessive diurnal intra-individual $\delta^2H_X$ variances emerge in all considered
growth forms (Fig. 5). Observed maximums were 18.0‰, 21.0‰ and 25.2‰ in $\delta^2H_X$ for trees,
shrubs and herbs respectively (Fig. 5; 2.8‰, 6.8‰ and 6.5‰ in $\delta^{18}O_X$ in Fig. S5). The null
model expected diurnal variance was exceeded for each species during its measurement period,
with the exception of $\delta^2H_X$ measurements of _P. euphratica_. The latter is a riparian forest species,
living along the river course, where an easily accessible and abundant ground-water reservoir
drives its RWU and *i-H₂O-xyl*.

**4. Discussion**
**4.1. Dynamic diurnal isotope compositions of xylem water along plant stems**
Our model shows that basic plant hydraulic functioning will result in shifting mixtures of $\delta^2H_X$
entering the plant (Fig. 1a-2a). Daily $\Psi_{X,0,t}$ fluctuations interact with the $\Psi_{S,i,t}$ profile causing
different parts of the root distribution to be active during the day. The fluctuations in $\delta^2H_X$ at
the stem base propagate along the xylem with a velocity proportional to the sap flow and this
produces variability in sampled $\delta^2H_X$ that is much larger than the expected measuring error. In
addition, empirical field data show excessive *i-H₂O-xyl* variance along the stem length (Fig. 4)
and over a short time frame (i.e. sub-daily, Fig. 5). Therefore, the assumption of uniform $\delta^2H_X$
along the length of a lignified plant is rejected, both theoretically and empirically.
Consequently, rather than being static, $\delta^2H_X$ values along the height of a plant should be
considered a dynamic diurnal process.



471   Importantly, we show that violation of this assumption results in incorrect assessment

472 of differences in RWU depths between plants. Differences do not necessarily result from

473 variability in RWU depth, but may result from monitoring plants at different heights (Fig. 2),

474 at different times (Fig. 1b) or by comparing individuals which have different $SF_V$ (Fig. 1c). Our

475 sensitivity analysis reveals that various soil and plant characteristics have an important role in

476 determining both the daily maximum $\delta^2H_X$ variance as well as the width of the $\delta^2H_X$ -baseline

477 drop. These two characteristics directly impact (i) the expected maximum bias in estimates of

478 RWU depth and (ii) the chance of measuring $\delta^2H_X$ values that do not represent a mixture of all

479 rooting layers during peak RWU (i.e. the baseline drop). Our work supplements the recent

480 overview of Penna $et\ al.($ 2018) discussing challenges in using stable isotope composition of

481 water to study the terrestrial water fluxes. We additionally advocate that future research should

482 explore the minimum set of (bio)physiological drivers and processes that require quantification

483 to correctly interpret $\delta^2H_X$ along the length of a plant.

485 **4.2. General applicability of model and results**

486 A necessary condition for diurnal shifts in RWU is the existence of a water potential

487 heterogeneity, e.g. more negative water potentials in the upper layers where trees usually have

488 higher root density, which causes a disproportional partitioning of diurnal RWU between deep

489 and shallow roots. Since such a gradient is formed when the upper soil layers undergo

490 evaporation, these conditions are also necessary for the existence of a soil isotopic gradient.

491 Thus, the problem we have identified is intrinsic to the isotopic tracing method for RWU

492 assessment.

493   Plant transpiration results from complex interaction between atmospheric demands (i.e.

494 driven by VPD and radiation) and stomatal conductance which depends on tolerance of drought



stress and soil moisture content. We may expect diurnal fluctuation in radiation and VPD, and
hence in water transport and depth of water absorption, as modelled here to be a general
phenomenon in nature. Hence, there is a real risk of misinterpretation and calculation errors
within the existing literature whenever *i-H2O-xyl* are used to asses RWU and water competition
strategies. Moreover, much greater fluctuations in VPD and radiation should be expected under
natural conditions than the diurnal cycle described here, and these will increase variability of
transpiration fluxes, leading to even more complex dynamics of $\Psi_{X,0,t}$. For instance, slight
alterations in these variables, i.e. a changing degree in cloud cover, can influence $\Psi_{X,0,t}$ rather
abruptly (for e.g. lianas; Chen *et al.*, 2015) and lead to instantaneous changes in $\delta^2 H_X$. Clearly
this further complicates the comparison of samples from different plants and sampled at
different heights and times, to date overlooked in RWU assessments, and our model certainly
illustrates that these considerations are non-trivial.
Note that, based on our model, we expect that soil isotopic enrichment experiments will
generate extensive $\delta^2 H_X$ variation along the length of trees whenever diurnal RWU fluctuations
cause water extraction to shift between labeled and unlabeled soil layers. Furthermore, when
enrichment experiments target trees with different hydraulic properties (such as $SF_V$) care
should be taken as to determine when and where to sample these trees in order to assess an
enriched isotope composition. Researchers should be certain the signal will be present at the
sample height (Fig. 1-2).

### 4.3. Alternative causes of *i-H2O-xyl* fluctuation.

The SWIFT model provides a simple traceable and mechanistic explanation, using diurnal
variations in $SF_t$ and RWU, for the excessive variance and dynamic nature of the *i-H2O-xyl*
fluctuations with plant height and time of field samples (e.g. Fig. 4-5) and elsewhere (Cooper





*et al.* 1991). We believe that our model provides a plausible simple explanation for diurnal *i-*
*H₂O-xyl* variation, which contributes to the variation that is observed empirically. Nevertheless,
the model necessarily represents a simplified representation of plant hydraulic functioning and
is therefore limited. There may be alternative causes that contribute to the observed intra-
individual *i-H₂O-xyl* variances. We discuss these here.
*i.    Fractionation at root level*
An increasing body of observations show the occurrence of isotopic fractionation at the root
level governed by root membrane transport (Lin & Sternberg, 1993; Vargas *et al.*, 2017) or by
unknown reasons (Zhao *et al.*, 2016). Brinkmann *et al.* (2019) hypothesize that root level
fractionation causes disparity when RWU depth calculations based on $\delta^2 H_X$ measurements are
compared with those of $\delta^{18}O_X$. However, it is difficult to imagine a scenario where root
fractionation by itself can explain the observed diurnal fluctuations in *i-H₂O-xyl* with height
and time. Even if root fractionation significantly contributed to variation in *i-H₂O-xyl,* we would
still need to take into account diurnal fluctuation in RWU to explain the observed patterns.
ii.   *Temporal and spatial soil dynamics*
The dynamics of soil water movement is complex and soil water content can be extremely
heterogeneous in the three spatial dimensions and such variation is currently not represented in
SWIFT. Hydraulic lift is a process that generates a vertical redistribution of water in the soil
through the roots (Dawson & Ehleringer, 1993), which may change the soil water isotopic
composition and mixture drawn up by roots. However, hydraulic lift should redistribute and
mix the depleted isotopic signal of deeper layers with the enriched signal of shallower layers.
This should lead to lower variation in the soil profile, and less variation along plant length, as
such hydraulic lift cannot explain the observed patterns. Heterogeneity in horizontal
distribution of water pockets may also affect *i-H₂O-xyl* variance. Under these conditions, the





543 horizontal distribution of the absorptive root area becomes more important. However, as the

544 $\Psi_{S,i,t}$ and the isotope composition of soil water of these pockets are interlinked, the mechanistic

545 driver of water extraction is the diurnal fluctuation in water potential gradients in the plant,

546 conform SWIFT.

547 *iii. Storage tissue and phloem enrichment*

548 Storage tissues release water and sugars in the xylem conduits on a daily basis to support

549 water transpiration demand (Goldstein et al., 1998; Morris et al., 2016; Secchi et al., 2017) or

550 to repair embolism (Salleo et al., 2009; Secchi et al., 2017). Both water and sugars are

551 transported in and out storage tissue via symplastic pathways using plasmodesmata and

552 aquaporins (Knipfer et al., 2016; Secchi et al., 2017), a pathway which has been linked to

553 isotopic fractionation in roots (Ellsworth & Williams, 2007). Moreover, phloem transports

554 photosynthetic assimilates constructed at the leaf level potentially affected by transpiration

555 fractionation (Gessler *et al.*, 2013). Hence, these metabolic molecules might be constructed

556 from enriched $^{2}H$ and $^{18}O$ atoms. Water release from storage or phloem tissue might locally

557 alter *i-$H_2$O-xyl* (White et al., 1985). Additionally, time between water storage and release could

558 bridge multiple days, and corresponding isotopic composition may reflect different soil

559 conditions. It is evident that such dynamics are complex, and it is hard to predict how storage

560 tissue and phloem enrichment affect the *i-$H_2$O-xyl* patterns observed here. Xylem isotopic

561 sampling cannot differentiate between water resulting from RWU or storage, and therefore we

562 cannot discount the possibility that tissue and phloem enrichment play a role. At a minimum

563 this adds further uncertainty to RWU assessment.

564 Further studies should determine whether the implementation of additional mechanisms

565 such as tree capacitance, root and stem level fractionation, spatiotemporal soil water dynamics,



more detailed root systems or storage tissues impact the intra-individual *i-H₂O-xyl* and should
be accounted for to improve RWU assessment and interpretation.

**4.4.  The way forward**

Combining a plant hydraulic model with *in situ SF$_V$* and *in situ $\Psi_{S,i,t}$* can help improve

the robustness of RWU assessment and interpretation. Measurements of $\Psi_{S,i,t}$ at multiple
depths, i.e. by installing multiple soil water potential sensors that measure at high temporal
frequency, should be especially valuable since the SWIFT model showed high sensitivity to
alterations of this variable and these can be directly supplied as model inputs. At the same time,
the availability of *SF$_t$* measurements allows for identifying the moment when water uptake from
all root layers is at its maximum, which can be used to determine the optimal timing of sampling
at a given height providing a more robust estimation of RWU depth and uptake.

Alongside the modeling and theoretical approach presented here, new ways to study

$\delta^2H_X$ at a high temporal scale are strongly encouraged. For example, pioneering work of
Volkmann *et al.* (2016) to the development of an *in situ* continuous isotope measurement
technique that offers the possibility for monitoring *i-H₂O-xyl* at a sub hourly resolution. This
technique holds strong promise for further elucidating the natural $\delta^2H_X$ variances found within
plants and the physiology processes from which these variances result. Such high temporal
resolution of isotope measurements, coupled with *in situ* monitoring of various environmental
and plant biophysical metrics, are needed for both model improvement and further validation.
Moreover, these seem inevitable to eventually differentiate all causal mechanisms of the
observed intra-individual *i-H₂O-xyl* variance.

**5.  Conclusions**



We have demonstrated that the assumption of no intra-individual $i$-$H_2O$-$xyl$ variation is rapidly
violated once models incorporate even basic plant hydraulic functioning. Moreover, the
incorrectness of this assumption is confirmed by empirical field data, showing excessive
variance and high temporal fluctuations in $i$-$H_2O$-$xyl$. We expect the observed $i$-$H_2O$-$xyl$
variance and sub-daily fluctuations result, in part, from the mechanisms considered in the
SWIFT model, though they likely represent an end product of various physiological processes
which impact $i$-$H_2O$-$xyl$.

Our theoretical explorations warn that variability in the isotope composition of plant

xylem water can result in erroneous RWU depth estimation and will complicate the
interpretation and comparison of data: samples taken at different heights, times or plants
differing in $SF_V$ may incorrectly show differences in RWU depth. We further predict that
various soil parameters and plant hydraulic parameters affect (i) the absolute size of the error
and (ii) the probability of measuring $i$-$H_2O$-$xyl$ values that do not represent the well-mixed
values during the plants' peak RWU. Hydraulic models, such as SWIFT, should be used to
design more robust sampling regimes that enable improved comparisons between studied
plants. We advocate the addition of $SF_t$, which indirectly reflects diurnal RWU fluctuations,
and $\Psi_{S,i,t}$ monitoring as a minimum in future RWU assessments since these parameters were
predicted to be the predominant factors introducing variance in $i$-$H_2O$-$xyl$ from the SWIFT
model exploration. However, soil texture and root permeability are also key considerations to
measure especially when comparing across species and sites.

Our findings do not exclude additional factors that impact the observed intra-individual

$i$-$H_2O$-$xyl$ variance and temporal fluctuation. Therefore, we strongly emphasize the need for
more testing. Directed studies that validate and quantify the relative impact of other plant
physiological processes towards variance in $i$-$H_2O$-$xyl$ are a prerequisite before improved
modeling tools can be developed, and bias in RWU assessments eliminated.



**Acknowledgement**
This research was funded by the European Research Council Starting Grant 637643
(TREECLIMBERS), the FWO grants (1507818N, V401018N to HDD), the Carbon Mitigation
Initiative at Princeton University (MD, MDV), Agence Nationale de la Recherche
"Investissement d'Avenir" grant (CEBA: ANR-10-LABX-25-01), the Belgian American
Educational Foundation (BAEF to FM) and the WBI (FM). We are grateful to Samuel Bodé,
Megan Bartlett, Isabel Martinez Cano and Pedro Hervé-Fernández who provided feedback on
analytical and interpretative aspects of the study. We thank Dries Van Der Heyden, Wim Van
Nunen, Laurence Stalmans, Oscar Vercleyen, Katja Van Nieuland, Stijn Vandevoorde and
Clément Stahl for data collection and lab processing. We credit Pascal Petronelli and Bruce
Hoffman for species identification, and Cora N. Betsinger for proofreading. Cheng-Wei
Huang's work provided inspiration for this research.

**Author contribution**

H.V., M.D.V and P.B. supervised and provided guidance throughout all aspects of the research.
H.D.D., M.D.V and H.V. designed the study. H.D.D., L.Z. and L.W. collected the samples and
data during the field campaign and performed the processing and analysis of the samples. The
model was developed and coded by H.D.D, M.D.V, M.D. and F.M. All authors contributed to
interpretation of the results and to the text of the manuscript.

**Data availability**

Both the data and the SWIFT model are available on the GitHub repository
HannesDeDeurwaerder/SWIFT





**Competing interests**
The authors declare that they have no conflict of interest.



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





**Tables**
**Table 1.** Nomenclature.

| Symbol | Description | Unit |
|---|---|---|
| $A_{R,i}$ | The absorptive root area distribution over soil layer i | m$^2$ |
| $A_{Rtot}$ | The plants' total active fine root surface area | m$^2$ |
| $A_{SAPWOOD}$ | Sapwood area | m² |
| $A_x$ | Total lumen area | m² |
| b | Shape parameter for the soil hydraulic properties (Clapp & Hornberger, 1978) | dimensionless |
| $B_i$ | The overall root length density distribution per unit of soil, not necessarily limited to the focal plant. | m m$^{-3}$ |
| $\delta^2 H_{X,0,t}$ | Isotope composition of plant xylem water at stem base at time $t$ | in ‰ VSMOW |
| $\delta^2 H_{X,h,t}$ | Isotope composition of plant xylem water at height $h$ and time $t$ | in ‰ VSMOW |
| $\delta^2 H_{S,i}$ | Isotope composition of soil water of the $i^{th}$ soil layer (constant over time) | in ‰ VSMOW |
| $\delta_{sample}$ | Isotope composition of water within a sample | in ‰ VSMOW |
| $\Delta\widehat{\Psi}_{i,t}$ | Estimated water potential gradient between stem base and the $i^{th}$ soil layer at time $t$ derived from Eq. (8) | m |
| $\Delta\Psi_{i,t}$ | Soil water potential gradient between soil and roots at the $i^{th}$ soil layer at time $t$ | m H$_2$O |
| $\varepsilon^2 H_X$ ; $\varepsilon^{18} O_X$ | Normalized isotope composition of plant xylem water | in ‰ VSMOW |
| $f_{i,t}$ | Fraction of water taken up in the $i^{th}$ soil layer at time $t$ | dimensionless |
| $h$ | Measurement height | m |
| $i$ | Soil layer index | dimensionless |
| $i$-$H_2O$-$xyl$ | Isotope composition of plant xylem water | in ‰ VSMOW |
| $k_i$ | Soil-root conductance of the $i^{th}$ soil layer | s$^{-1}$ |
| $K_{max}$ | Maximum soil hydraulic conductivity | m s$^{-1}$ |
| $k_R$ | Effective root radial conductivity | s$^{-1}$ |
| $k_S$ | The conductance associated with the radial water flow between the soil and the root surface | s$^{-1}$ |
| $K_{S,i}$ | Soil hydraulic conductivity at the $i^{th}$ soil layer | m s$^{-1}$ |
| $\ell$ | The approximated radial pathway length of water flow between bulk soil and root surface | m |
| LF | Lumen fraction per unit sapwood area | m² m$^{-2}$ |
| n | Number of unique contributing water sources | # |
| $\Psi_{sat}$ | Soil water potential at soil saturation | m |
| $\Psi_{S,i,t}$ | Soil water potential of the $i^{th}$ soil layer at time $t$ | m |
| $\Psi_{X,0,t}$ | Water potential at the base of the plant stem at time $t$ | m |





| | | |
|---|---|---|
| R | Heavy to light isotope ratio measured in the sample or standard | % |
| $RWU_{i,t}$ | Net amount of water entering and leaving the root tissues per unit of time in the $i^{th}$ soil layer at time $t$ | $m^3\ s^{-1}$ |
| $SF_t$ | Instantaneous sap flow at time $t$ | $m^3\ s^{-1}$ |
| $SF_S$ | Sap flow velocity, calculated as the sap flow per sapwood area | $m\ h^{-1}$ |
| $SF_V$ | True sap flux density, calculated as the sap flow per lumen area | $m\ h^{-1}$ |
| $\tau$ | Delay before the isotope composition of xylem water at stem base reaches stem height $h$ | s |
| $\theta_{sat}$ | Soil moisture content at soil saturation | $m^3\ m^{-3}$ |
| $\theta_{S,i,t}$ | Soil moisture content of the $i^{th}$ soil layer at time $t$ | $m^3\ m^{-3}$ |
| $z_i$ | Soil depth of the $i^{th}$ soil layer | m |









**Table 2.** Sampled liana and tree individuals, provided with their species, respective diameter at breast height (DBH, in cm) and their $\delta^2H$ and $\delta^{18}O$ ranges (in ‰, VSMOW) measured per individual.

| Code | Growth form | DBH [cm] | Family | Species name | $\delta^2H_X$-range [in ‰, VSMOW] | $\delta^{18}O_X$-range [in ‰, VSMOW] |
|---|---|---|---|---|---|---|
| SP1 | Tree | 15.6 | Moraceae | *Coussapoa sp.* | -30.1; -25.5 | -2.8; -2.6 |
| SP2 | Tree | 50.9 | Fabaceae | *Vouacapoua americana* | -23.9; -18.1 | -3.1; -2.2 |
| SP3 | Tree | 44.6 | Vochysiaceae | *Erisma nitidum* | -27.7; -20.8 | -3.2; -1.9 |
| SP4 | Tree | 26.1 | Sapotaceae | *Micropholis guyanensis* | -29.8; -28.0 | -3.0; -2.9 |
| SP5 | Tree | 21.0 | Anacardiaceae | *Tapirira guyanensis* | -31.1; -18.0 | -3.2; -2.2 |
| SP6 | Tree | 49.7 | Fabaceae | *Albizia pedicellaris* | -26.9; -22.1 | -3.2; -2.6 |
| SP1 | Liana | 2.8 | Polygonaceae | *Coccoloba sp* | -27.9; -20.7 | -3.9; -2.3 |
| SP2 | Liana | 2.7 | Convolvulaceae | *sp.* | -29.3; -24.0 | -4.4; -2.9 |
| SP3 | Liana | 0.8 | Moraceae | *sp.* | -40.8; -22.6 | -4.5; -2.3 |
| SP4 | Liana | 3.8 | Combretaceae | *cf. rotundifolium Rich.* | -23.6; -15.2 | -2.9; -2.0 |
| SP5 | Liana | 0.7 | Convolvulaceae | *Maripa cf violacea* | -31.6; -19.7 | -3.8; -2.7 |
| SP6 | Liana | 3.8 | Convolvulaceae | *Maripa sp.* | -35.3; -24.4 | -4.8; -3.1 |



**Figures**

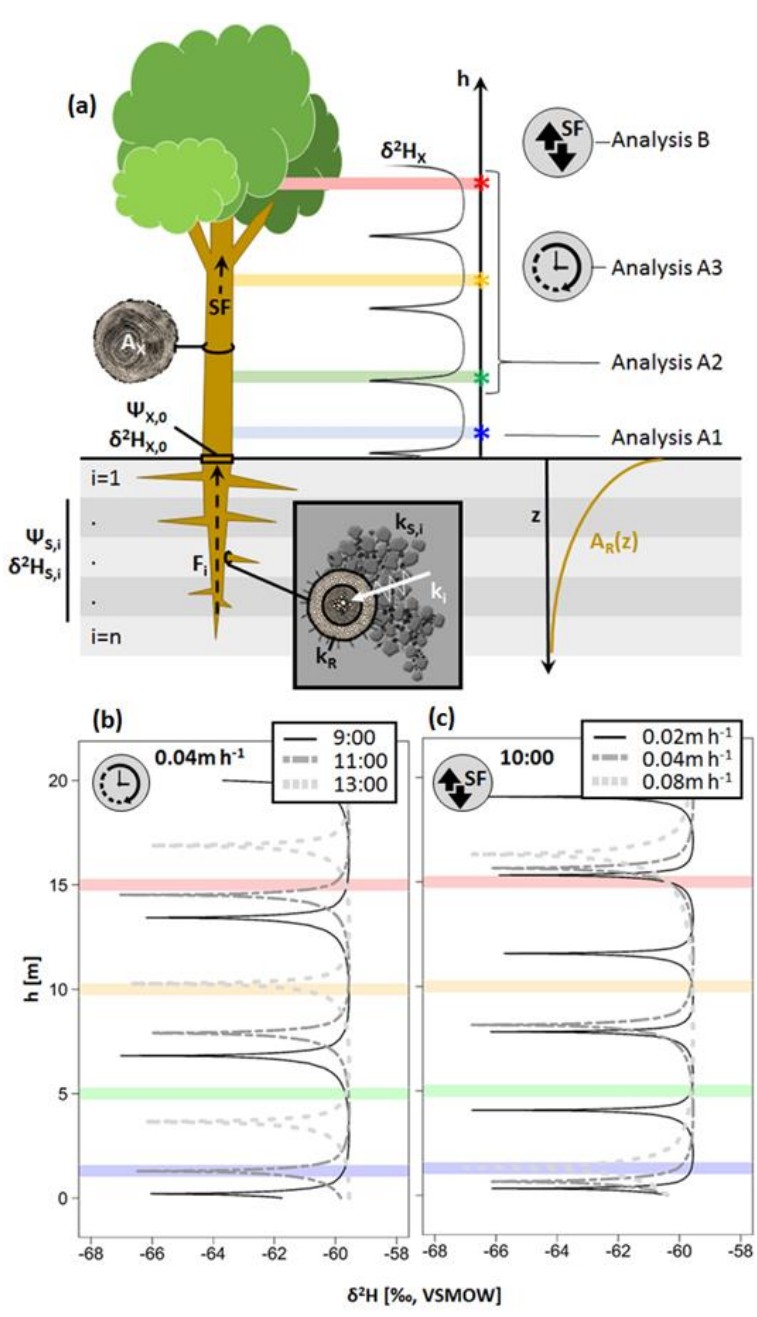







**Fig. 1. Panel a:** Schematic representation of the model and considered analysis detailed in the
text. **Panel b:** Model outputs for model analysis A3 representing the deuterium isotope
composition of xylem water ($\delta^2H_X$) as a function of the tree height simulated for different
sampling times (9:00, 11:00 and 13:00). The modeled tree has an average daily sap flux density
of 0.04 m h$^{-1}$ (SF$_S$), which corresponds to an average daily true sap flux density of 0.28 m h$^{-1}$
(SF$_V$). Panel c: Model outputs for model analysis B where $\delta^2H_X$ in relation with stem height is
shown at 10:00 h, but parameterized with distinct average sap flux density, i.e. 0.08, 0.04 and
0.02 m h$^{-1}$ (corresponding to an average true sap flux density SF$_V$ of. 0.56, 0.28 and 0.14 m h$^{-1}$
$^{1}$, respectively). The standard parameterization used for both study analysis is detailed in Table
S1.



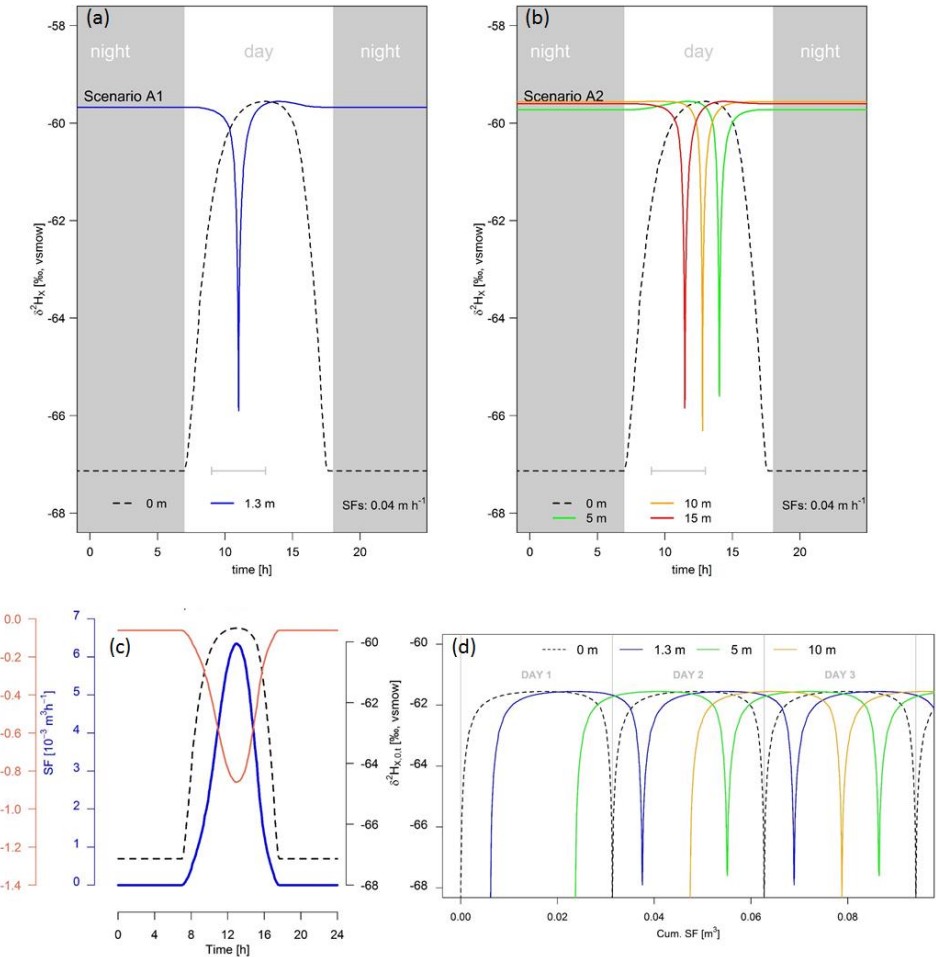


**Fig. 2. Panel a & b:** Diurnal patterns of simulated in deuterium isotope composition of plant

xylem water ($\delta^2H_X$) fluctuation as a function of time for various tree heights. The modeled tree

has an average daily sap flux density ($SF_S$) of 0.04 m h⁻¹, which corresponds to an average daily

true sap flux density of 0.28 m h⁻¹ ($SF_V$), and the standard parameterization is detailed in Table

S1. Panel (a) shows analysis A1 output where diurnal $\delta^2H_X$ patterns are provided at stem base

(0 m, black dashed line) and at general tree coring height at breast level, i.e. at 1.3 m (blue).

Panel (b) shows analysis A2 outputs demonstrating diurnal patterns in $\delta^2H_X$ within a standard

tree at various heights, i.e. at 0 m (black dotted), 5 m (green), 10 m (orange) and 15 m (red).



These heights represent random branch sample collection and conform to the standard practice
of RWU assessment. Grey lines with whiskers indicate the common sampling period (9:00 until
13:00) according to standard practice. **Panel c:** Sap flow rate ($SF$, blue line), deuterium isotope
composition of xylem water ($\delta^2 H_{X,0,t}$ black dashed line) and water potential at stem base
($\Psi_{X,0,t}$, red line) are shown over the period of a single day. **Panel d:** Simulated $\delta^2 H_X$ fluctuations
in function of the cumulative sap flow volume measured at various heights: stem base (0 m,
black dashed), 1.3 m (blue), 5 m (green) and 10 m (red). Days are delineated by grey vertical
lines.









**Fig. 3**. a) Relative error on the inferred root water uptake (RWU) depth (i.e. bias between the
average daily and the instantaneous derived RWU depth), for a tree measured at standard tree
coring height (i.e. 1.30 m) which has a sap flux density ($SF_S$) of 0.04 m h$^{-1}$ (i.e. $SF_V = 0.28$ m
h$^{-1}$), over the common sampling period (9:00 until 13:00). b) Relative error on the inferred
RWU depth considering a tree measured at standard tree coring height (1.30 m) at 11:30, but
which differs in $SF_S$. c) Relative error on the inferred RWU depth over the duration of the
common sampling period (9:00 until 13:00) and over a range of potential $SF_S$ (in m h$^{-1}$) –
corresponding to $SF_V$ range of 0.15–1.25 m h$^{-1}$. Dotted lines a (black) and b (grey) correspond
to their respective representation in panel a and b. day= -1 and day= 0 indicate whether the
derived RWU depth error corresponds to the previous or current day of measurement.



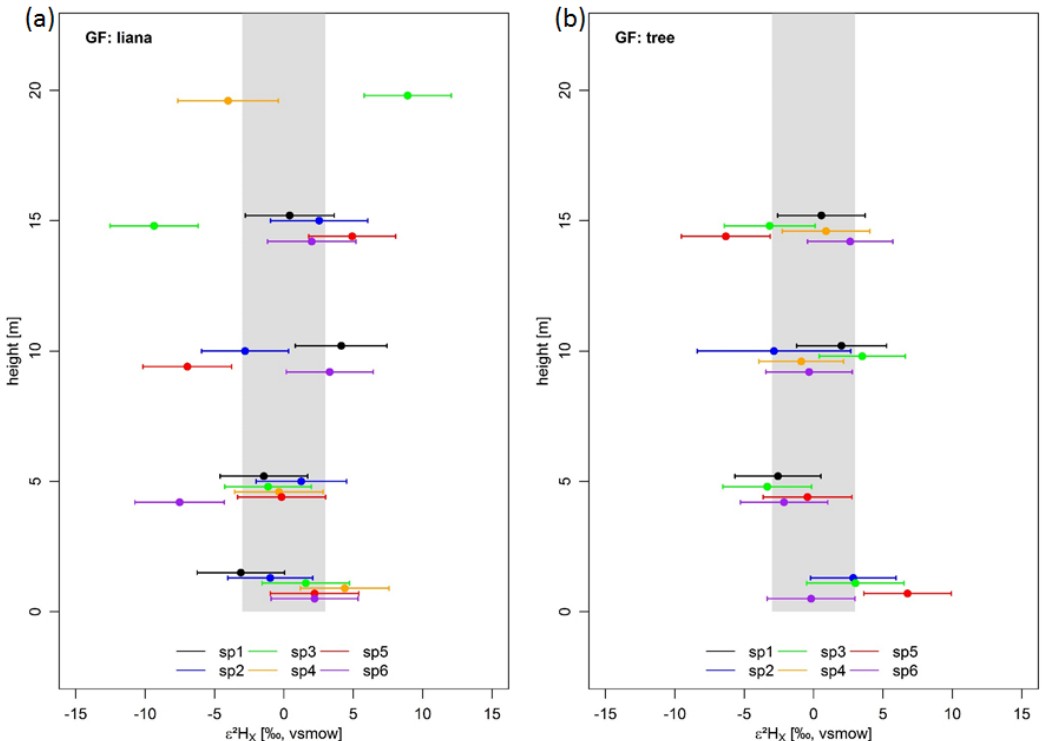

**Fig. 4.** Field measurements of normalized intra-individual $\delta^2H_X$ ($\varepsilon^2H_X$) for six lianas (panel a) and six trees (panel b). Individuals are provided in different colors; species names can be derived from Table 2. Error whiskers are the combination of potential extraction (± 3‰) and measurement errors of the isotope analyzer. The full grey envelope delineates the acceptable variance from the stem mean (i.e. 3‰) according to the standard assumption of no variance along the length of a lignified plant, i.e the null model.









**Fig. 5**. High temporal field measurements of deuterium isotope composition of xylem water
($\delta^2 H_X$) of two tree (red, stem samples), two shrub (blue, stem samples) and two herb (green,
root samples) species sampled in the Heihe River Basin (northwestern China) shown for the
respective measurement periods. Timing and location of sampling are provided in the panel
titles. The full colored envelope per respective species delineates the acceptable variance from
the stem mean (i.e. 3‰) according to the standard assumption of no variance along the length
of a lignified plant. Grey vertical lines mark the transition of days. The table provides the
maximum measured diurnal $\delta^2 H_X$ range per species.