# Peer review of "Causes and consequences of pronounced variation in the"

_Biogeosciences, 2019_

## Referee Comment (RC1) · Anonymous Referee #1 · 19 Feb 2020

De Deurwaerder and colleagues present a composite work where they (i) run a model simulating diurnal variations and vertical heterogeneity in xylem water isotopic composition ($\delta$xyl) and perform a multi-variate sensitivity analysis. They also (ii) present results of sampling campaigns where $\delta$xyl temporal and spatial variations were observed in twelve tree and liana species. The authors explain these variations and thus the departure of the generally accepted hypothesis of homogeneous $\delta$xyl on account of their model output. Finally they warn the isotopic community against the "danger" in using water stable isotopes as tracers for RWU analysis.

The manuscript is well written, figures and tables are of good quality and appropriate

referencing supports the text. Finally the manuscript content falls within the scope of BG.

My general comments are listed below:

1- I note that the authors do not confront their model results to collected data, nor thoroughly test their model hypotheses on independent data. I do not see a particular problem, but it should be mentioned clearly that aforementioned items (i) and (ii) are only "softly" coupled in the study;

2- highlighting the temporal as well as spatial (longitudinal) dynamics of $\delta$xyl is of evident interest. However the prevalence of such dynamics may not put in "danger" – as the authors say – the determination of fractional root water uptake for other non-wooden species. The abstract should be rewritten accordingly. The isotopic community should be on the "safe" side if researchers extract water from a plant tissue for which it has been proven that its stable isotopic composition reflects that of RWU. Of course, this should be investigated for each investigated plant species, preferably under controlled conditions (see for example: Barnard et al., 2006);

3- The authors provide no information about the soil compartment; what about the soil water isotopic composition profile temporal and spatial variabilities? Are the isotopic differences in xylem water reflected by the span of isotopic composition values in soil water? This would offer the possibility to rule out possible evaporation effects mostly during sampling and transport (which is not listed as other reasons for the observed diurnal variations of $\delta$xyl). If soil water isotopic information is not available, it should be stated as limitation of the study;

4- I found on several occasions that the authors did not fully understand basic principles driving isotopic fractionation (see my specific comments);

5- In general, I do not think that such field experiments, where a significant number of environmental driving factors are unknown, should be used to question the entire

isotopic research methodology. I urge the authors to discuss this point as well and measure their words.

The authors will also find a list of specific questions/remarks/corrections/issues:

L24ff. What does "i-H2O-xyl" refer to? To "plant xylem water uniform isotope composition" or "plant xylem water isotope composition"? In either case, "$\delta$" is to be preferred over "i-H2O-"

L32-33. "field data show excessive i-H2O-xyl variation during the day or along stem length ranging up to 25.2‰ in $\delta$2H and 6.8‰ in $\delta$18O" does not read well. I propose something like: "the hydrogen (oxygen) isotope composition of plant xylem water showed strong temporal (i.e., daily) and spatial (i.e., along the stem) variation ranging up to 25.2 (6.8) ‰'

L36. Please rephrase: "danger" is not the proper word.

L46-47. There is no such thing as the depth of root water uptake in the case of several soil water sources. Only in the context of direct inference is this true. But the authors do not refer to the later (and outdated) technique.

L49. This is not true: the isotopic technique is of course destructive (you have to take a soil core), very labor intensive (e.g. extraction of soil and plant xylem water).

L50 (also L47). You should mention that it is fractional RWU and not absolute RWU you are talking about. You cannot solve for water mass balance with the isotope technique, which constitutes its greatest limitation when compared to other techniques.

L52. How would you determine fractional root water uptake at the ecosystem level?

L56. This should be "$\delta$xyl". Why the "i" instead of "$\delta$" here? Also, why the "H2O"? (is there another molecule investigated here?)

L58-60. Peclet effect is measureable in the xylem vessels upstream of the evaporative sites. This assumption is not systematically made. Instead, authors investigate the

prevalence of isotopic fractionation depending on the plant tissues they sample, e.g. in Barnard et al. (2006). Please revise.

L67-69. There can only be kinetic fractionation playing a role during the transport of water through the root membranes, since there is no liquid-vapor phase change that would involve equilibrium fractionation. Please revise.

L69-72. Not only kinetic fractionation is a result of the difference in mass of the water isotopologues, but fractionation in general (e.g., equilibrium and kinetic fractionations).

L94-95. Why would you make the assumption that $\delta$xyl is constant over time (over which period of time anyway)? At this point of the MS, it is not clear. Actually, no one makes this assumption in the field, rather they sample from e.g. the base stem among individuals at e.g. a sub-hourly temporal resolution and sub-daily temporal extent.

L100-101. What do you mean by "diurnal changes in the soil-plant-atmosphere continuum"? Which changes?

L113. What exact "water potential gradients" do you refer to?

L114-116. Why would you need to use a mixing model, especially since you did not sample soil water and determine its isotopic composition? You may as well simulate a sinusoidal pattern for the $\delta$xyl. Please elaborate/explain.

L117-119. You should write the isotopic equations with "$\delta$" instead of "$\delta$2H" as the model does not focus on 1H2HO, to the contrary of what the authors say. For the model to focus on 1H2HO, it would mean that 1H2HO and 1H218O would follow different physical processes, which is not the case (both isotopologues undergo mass-dependent fractionations, i.e. $\varepsilon$eq(2H)/ $\varepsilon$eq(18O)$\approx$8 and $\varepsilon$k(2H)/ $\varepsilon$k(18O)$\approx$0.88). Also write 2H instead of Deuterium and do it consistently throughout the MS. The latter is just an element's isotope and does not deserve (anymore) its own letter (see IAEA tech reports guidelines).

L130-131. This assumption is only reasonable when soil water redistribution no longer

occurs, e.g., this does not stand shortly after a rain event.

L125ff. Report the dimension of each variable and parameter throughout the MS.

L142. From Eq. (3), I understand that the water "potentials" are in fact "hydraulic heads". This should be clarified.

L143-144. Add here that $k_i$ and ðİŻźðİŚ"ðİŚą are also specific to the ith soil layer.

L193-196. I am missing background information to understand what the "30 days sequence" of the "model runs of Huang et al. (2017)" refers to. Please elaborate.

L201-203. Why would you need external data (Meissner et al. 2012) and not simply do your model exploration on basis of a synthetic experiment?

L208-209. I did not hear of such standard practice and I doubt there is. Could you add a reference for this?

L223-225. Split the sentence and add detail. It is hard to understand. Also following the Rayleigh distillation model, the error should always be negative in the case of incomplete water recovery, which does not match to your normal distribution of error in the null model.

L228-229. How so? And why would it be relevant to take into account the analyzer systematic error at this point of your model testing?

L235-244. Are you talking about RWU depth of rooting depths here? How do you define the latter term? Why would you use the direct inference model (which is a very simplistic view on RWU, i.e., one single root sampling from on single layer at a time) if you use a multi-source mixing model (Phillips and Gregg, 2003), which allows the plant to sample simultaneously from different layers? Please explain this apparent contradiction. Overall this section is quite difficult to read and I ask that the authors simplify it.

L258. Is there a specific reason why you did not use Van Genuchten's soil retention

curve?

L309. Delete "kinetic". It is not even sure that you would have fractionation at all, considering that you may boil (==fractionation free process) the water here rather than evaporate it.

L321. Fresh weight does not take into account possible loss of water during transport/storage. You should have weighted the samples prior extraction again.

L331-336. Since you are measuring with a Picarro, which does not give ratio (but performs already the delta conversion), you need to say that you "corrected the Picarro raw delta readings into calibrated delta values thanks to the values of the aforementioned 'internal laboratory References' expressed on the international V-SMOW scale". No need to display the equation (12) but you may detail these "internal laboratory References" (e.g., value).

L336-334. Still at this point, I do not know what the difference is between i-H2O-xyl and δxyl…If there is none, please use the latter term. In addition, use another letter than ε for the normalized "i-H2O-xyl": it usually stands for isotopic fractionation, defined as the deviation of the fractionation factor to unity. It seems even odd that you would consider such a letter…

L369. I still do not understand what is the concept of RWU depth if you consider the multi-source mixing model approach.

Fig. 2. Panel (a): how do you come up with a night δxyl at 1.3 m above –60‰ Also, I don't see why panels (a) and (d) look so different for day 1, since if I understand correctly, the cumulative SF is a function of time (if sap flow remains constant).

L371. "isotopic composition of soil water is dominated by depleted deuterium". Please correct phrasing: soil water can be depleted in 2H in comparison to another water volume, but there is no such thing as "depleted 2H".

L373. An isotopic composition, which is a number, cannot be "enriched". Please correct.

L375. "depleted deep soil water"

L384. "...RWU originating from deeper, more depleted soil layers". Please correct: water from a given soil layer might be depleted, not the soil layer in itself.

L399-400. This belongs to the discussion section.

L407-418. Nowadays no study is published where RWU depth is investigated with the direct inference method. Analyses are performed with Bayesian mixing models. So I wonder if this section, although interesting theoretically, would benefit practically to the community.

L446 and Fig. 4. See my previous comment on the use of "$\varepsilon$". The caption of a figure should not point to another figure or table. Write here the name of the species (no need to write them in the figures though).

L452. Add in the text that growth forms refer to lianas and trees.

L455-457. This belongs to the discussion section. Also the link between "easily accessible and abundant groundwater reservoir" and the fact that the diurnal intra-individual variance is minimized is not clear. I suggest moving to the discussion and elaborating on this.

L471-472 and Table 2. How many individuals (which you could consider as replicates) of each species were sampled during the experiment? Discuss the implication of having n=1 with respect to $\delta$xyl variance.

L486-492. The authors say that the intrisinc problem of the "isotopic tracing method" is that there is a soil water isotopic gradient in case there is evaporation and under heterogeneous soil water potential gradient? I don't understand this at all (!) The isotopic methodology for studying plant RWU relies on heterogeneous isotopic gradients in soil water. This is a solution, not a problem here...

L493-506. I disagree. There is a clear problem in determining fractional RWU profiles on basis of measurements of the transpiration isotopic composition, which is highly temporally dynamic and spatially heterogeneously distributed; many observation of leaf water confirm the non-reaching of isotopic steady state. In addition, how would a "change of cloud cover degree" have an "instantaneous" influence on $\delta$xyl? This contradicts the results of your synthetic experiments, where depending on sap flow rate, there is a marked isotopic memory effect of the antecedent water moving upward it the xylem vessel.

L516-523. The model provides an explanation, sure, but does not validate your hypotheses from the confrontation with experimental data. This is missing from your study and should be mentioned.

L534-546. My understanding from the literature is that hydraulic redistribution is intermittent and localized, thus does not affect that much the bulk soil water isotopic composition, rather it affects the direct environment of the roots.

L578-587. Not to forget we need to monitor soil water isotopic composition to verify if $\delta$xyl spreads within the range of isotopic values observed in the soil profile.

References:   Barnard, R. L., de Bello, F., Gilgen, A. K., and Buchmann, N.: The $\delta$18O of root crown water best reflects source water $\delta$18O in different types of herbaceous species, Rapid Commun. Mass Spectrom., 20, 3799-3802, doi:10.1002/Rcm.2778, 2006.

---

## Referee Comment (RC2) · Anonymous Referee #2 · 20 Feb 2020

The manuscript by De Deurwaerder and colleagues challenges the idea that, in absence of precipitation or other rapid changes in climate, the water isotope composition of plant xylem should stay fairly constant over diurnal time scales or along stem height. Their analysis is based mostly on a model (!) of root water uptake and isotopic transport within the roots, up to the stem base. Their model considers that (1) the isotope composition of stem water at the base of a tree ($\delta^2 H_x(0,t)$) is the average isotope composition of soil water over the root zone, weighted by the fractional root water uptake rates at each depth (Eqs. 1 or 7) and (2) the isotope composition of stem water at any height $h$ ($\delta^2 H_x(h,t)$) is the isotope composition of stem water at the base, delayed by the travel time $\tau$ of sap between stem base and height $h$ ($\delta^2 H_x(h,t) = \delta^2 H_x(0,t-\tau)$)

(Eq. 9). Soil properties are used as boundary conditions that do not change over the day in terms of soil water potential and isotopic composition. With such model, they predict large diurnal variations of xylem water isotopes at stem base, but also large variations along the stem (see their Figs. 1 and 2). Based on this modelling exercise, and separate observations of the $^2H/^1H$ ratio in water extracted from tree stems and lianas at different heights within a tropical forest canopy, and showing some scatter sometimes larger than $\pm3$‰ (the estimated error from water extraction and isotope analysis), they conclude that (1) the common assumption that the isotope composition of stem water is fairly constant over time is violated and (2) it can cause significant biases when using water isotopes to identify plant water origin.

I think it is good that the authors bring forward the point that xylem water may sometimes exhibit rather dynamic variations in its isotope composition. However, I am afraid the proposed model is inadequate and the dataset is too limited for illustrating this point. To me, the study does not prove anything; it shows that there are variations in the data and that there are variations in the model but there is no model-data comparison. Besides, variations in the data are not very large and can be explained by lots of other processes, and variations in the model are mostly caused by its lack of realism. These two points are explained in more details below.

The dataset accompanying this study only consists of a few water isotope data from tree stems and lianas collected over a couple of days. No soil water data is shown, or even sap flow or rooting depths. I doubt it is the best dataset to test the proposed theory, or draw any conclusion about plant water uptake. The data shown in Fig. 5 is interesting but it comes from another study (Zhao L, Wang L, Liu X, Xiao H, Ruan Y, Zhou M (2014) The patterns and implications of diurnal variations in the d-excess of plant water, shallow soil water and air moisture. Hydrology and Earth System Sciences, 18, 4129–4151). Many processes (stem evaporation, different proportions of storage tissues or even atmospheric vapour use) and measurement artefacts (during sampling and transport, water extraction, isotopic analysis...) could explain significant

variations in the water composition of stems from trees and lianas of different statures. Accounting for uncertainties in the extraction and analysis is certainly not enough.

More importantly, I find the modelling analysis flawed and totally unrealistic. As explained above, the proposed model simulates water isotope gradients along the stem based on the average travel time of sap between two stem points (i.e. assuming the water isotope composition of xylem water at height $h$ is that at stem base at an earlier time corresponding to the travel time between stem base and height $h$). By doing so, the model neglects the mixing of water isotope by diffusion during water transport. If we neglect pit structure and consider vessels as regular pipes, the Péclet number $\wp$ that compares advection and diffusion is, using their notations: $\wp = SF_V h/D_l$. Taking an average sap flow velocity of $SF_V = 0.3 mh-1$ (see caption of Fig. 2) a typical height (diffusion length) $h$ of 1m and a self-diffusion of liquid water of $D_l = 2.5 10^{--9} m^2 s^{-1}$ this leads a Péclet number $\wp$ around 30000, i.e. high enough to justify neglecting (a posteriori) water mixing by diffusion. However, mixing with storage tissues also occurs and tree sap does not move like a slab. In their model, as soon as transpiration stops, root uptake stops and sap flow at any height stops too so that the $\delta^2 H$ of xylem water at any height remains to its value at dusk over the entire night until the following morning plus the time delay $\tau$ (see for example Fig. 2a, the curve for $h$=1.3m). In reality, at night, sap flow does not stop immediately because plant elastic tissues need to be replenished. Root uptake will continue until full replenishment of the elastic tissues is done. This will contribute to homogenisation of xylem water over night. Also when sap flow becomes small diffusion is not negligible anymore (low Péclet), which will reinforce the isotopic mixing by water diffusion. In other words, in the real world, xylem water should not exhibit large isotopic gradients along the stem such as shown in their Figs. 1 or 2. Mixing with storage tissues is briefly discussed (section 4.3.iii) but not in the same direction as above. If night-time mixing of xylem water in roots and stems was accounted for, this should strongly minimise the predicted diurnal variations of $\delta^2 H_x(h, t)$, even at stem base. Not accounting for diurnal variations in soil conditions (water potential and isotopic composition) is also a strong limitation of the model.

In conclusion, I find the argument raised by De Deurwaerder and colleagues not supported by their data nor by their model simulations. More realism would need to be brought to the model and the dataset should be complemented with additional information before drawing any conclusion on how variable the isotopic of xylem water in tree stems and lianas is over diurnal time scales or with height.

---

## Author Comment (AC1) · 6 Apr 2020

Dear Editor,

We hereby submit our final response and proposal for improvements to the manuscript "Diurnal variation in the isotope composition of plant xylem water biases the depth of root-water uptake estimates" to be considered for publication as a research article in *Biogeosciences*.

First, we would like to thank both referees for their thorough assessment of our manuscript, as their suggestion will greatly improve its quality. We are pleased that both reviewers acknowledge the importance of our study (e.g. reviewer #1: *"highlighting the temporal as well as spatial (longitudinal) dynamics of δxyl is of evident interest"*, reviewer #2: "*I think it is good that the authors bring forward the point that xylem water may sometimes exhibit rather dynamic variations in its isotope composition*"). We were also happy to notice that the essence of our work (i.e. investigating the diurnal variability in isotopic composition along woody stems) and its merits are not questioned. We noticed that most of the (major) criticisms arise from problems in the presentation, formulation and overstatement of our work. We have no doubts that these oversights can and will be addressed. In the revised version of this manuscript, we will therefore focus more on the experimental results we obtained while providing a more balanced presentation of the limitations of our study and of the model we developed as a plausible explanation of the observed variability. In particular,

**I.** both reviewers indicated that our empirical dataset is not ideal (i) for model validation, and (ii) to support some stronger statements regarding the implications of our findings. We acknowledge these points and will address them by:

1. **restructure the manuscript giving more emphasis on the strong points of our empirical datasets,** which are unique in the field, and shows excessive variability (temporal and longitudinal) in the isotopic composition of xylem water ($\delta^2 H_X$);

2. **emphasize clearly that the model analysis is intended as a theoretical exploration** to build hypotheses and to understand when to expect large variance in $\delta^2 H_X$. We will clearly state that the coupling between the data and the model is only qualitative at this stage;

3. **toning down the manuscript title and softening some statements** (see details in our response to the reviewer comments), especially regarding the limitations of the isotope method for determining RWU;

4. **expand the existing discussion section, by elaborating** our existing section on alternative hypotheses that could contribute to the observed variability;

5. **shed a more positive light on the implications** of diurnal variability in $\delta^2 H_X$ as this can lead to novel information and opportunities in water acquisition and plant performance studies.

**II.** concerns were raised about the realism of the presented model. Our model considers basic physical and physiological processes, and we agree that it is inevitably - as every other model - a simplification of reality. We stress, however, that the suggested implementations, while improving realism, will not change the conclusion of the paper: excessive changes in $\delta^2 H_X$ can be expected along the stem of woody plants (or similarly at one vertical position over time). We highlight this by implementing some suggestions, and providing further details in our discussion where alternative hypotheses seem appropriate – we note that some

of the reviewers' concerns were already included in the discussion section and will be elaborated on. In particular, changes include:

1.  Reviewer 2 suggested including a diffusion term in the transport equation as it can lead to homogenization of $\delta^2 H_X$ within the plant. We explored this possibility using analytical solutions of the advection-diffusion equation. **These simulations show that the impact of diffusion is small and cannot cause complete homogenization** (see figure 1, below). Diffusion will - very slowly, i.e. over multiple days - reduce the absolute range of variability in $\delta^2 H_X$ by smearing the isotopic composition (See figure: 5cm in 24h), but it also leads to broader $\delta^2 H_X$-baseline drops. This implies that while the absolute range of variability might slightly decrease over time (or along tree height). The probability sampling in the $\delta^2 H_X$-baseline drop will in fact increase, **strengthening the importance of our main message.** This new finding will be included in the new version of the manuscript.

2.  There is potential for water exchange between storage tissues and xylem water, we discuss this implication in the discussion, but decided not to include such a process in the present model version as (i) it depends on the assumption that storage water is representative of soil water uptake by the roots, (ii) we do not have information on storage water isotopic signature and dynamics and (iii) we are not aware of any existing dataset that could validate/parameterize this model process. In the discussion, we particularly highlight that, if storage water is not representative of soil water uptake by the roots, then exchange with storage water **likely exacerbates potential bias in the isotope tracing technique, strengthening the main issues raised in our paper. Moreover, no complete homogenization is visible in the presented empirical data (or in the literature in general) despite** the likely **exchange from storage cells**.

3.  Similarly, including variations of soil water isotopic composition and water potential over time may improve the model realism and affect the absolute range and the dynamics of xylem water isotopic composition **but would not lead to homogenization. We will also elaborate more on this in the revised discussion.**

Please find our detailed responses to reviewer comments below (responses to reviewers in bold).

We hope that the proposed adjustments, inspired by the reviewers' suggestion, will improve the manuscript allowing publication in *Biogeosciences*.

The authors

[Figure]

[Figure]

**Figure 1: (a) Simulation of the impact of diffusion flux (D=3.0 10$^{-5}$ cm/h; Meng et al., 2018) on the isotopic composition of xylem water while assuming no advection flux (i.e., Sap flow = 0 cm h$^{-1}$). Colored lines show the smearing of the isotopic composition due to diffusion at different time intervals: 1, 6, 12 and 24h. (b) Simulation of the impact of diffusion (D=3.0 10$^{-5}$ cm/h; Meng *et al.*, 2018) on the isotopic composition of xylem water during sap flow activity (i.e. Sap flow velocity = 25cm h$^{-1}$). Colored lines show the combined impact of diffusion and advection on displacement and smearing of the isotopic composition at different time intervals: 1, 6, 12 and 24h. The simulations are analytical solutions of the advection-diffusion equation.**

*Anonymous Referee #1*

De Deurwaerder and colleagues present a composite work where they (i) run a model simulating diurnal variations and vertical heterogeneity in xylem water isotopic composition (δxyl) and perform a multivariate sensitivity analysis. They also (ii) present results of sampling campaigns where δxyl temporal and spatial variations were observed in twelve tree and liana species. The authors explain these variations and thus the departure of the generally accepted hypothesis of homogeneous δxyl on account of their model output. Finally they warn the isotopic community against the "danger" in using water stable isotopes as tracers for RWU analysis.

The manuscript is well written, figures and tables are of good quality and appropriate referencing supports the text. Finally the manuscript content falls within the scope of BG.
**We thank the reviewer for his/her appreciation of the quality of our work, and the detailed assessment of the study and constructive feedback on the manuscript. We feel that the paper improved considerably thanks to his/her suggestions.**

My general comments are listed below:

1- I note that the authors do not confront their model results to collected data, nor thoroughly test their model hypotheses on independent data. I do not see a particular problem, but it should be mentioned clearly that aforementioned items (i) and (ii) are only "softly" coupled in the study.
**The reviewer makes a fair point. As our field data is unique but limited, they do not allow direct validation. Our model presents a theoretical exploration of one of the potential causes of the observed high variability in isotopic composition in xylem water ($\delta^2H_X$) along the stem of a woody plant. For illustrative and interpretative purposes, our model explores ideal, simplified environmental conditions. The empirical data present a much more complex situation, which we were unable to characterize fully due to financial and logistical restrictions. The new version of the manuscript will mention that our study presents only a soft, qualitative coupling between model and data, as suggested by the reviewer. In particular, the new manuscript version will first present the unique dataset that we collected, and then the model that we develop as a potential explanation of the variability found along the plant stems. In the manuscript discussion, we intend to discuss further other potential explanations of such variability and how they could be integrated and tested within our modelling framework.**

2- highlighting the temporal as well as spatial (longitudinal) dynamics of δxyl is of evident interest. However the prevalence of such dynamics may not put in "danger" – as the authors say – the determination of fractional root water uptake for other non-wooden species. The abstract should be rewritten accordingly. The isotopic community should be on the "safe" side if researchers extract water from a plant tissue for which it has been proven that its stable isotopic composition reflects that of RWU. Of course, this should be investigated for each investigated plant species, preferably under controlled conditions (see for example: Barnard et al., 2006).
**The abstract of the paper will be rewritten accordingly. We agree that a distinction between woody and non-woody plants should be considered as described by Barnard *et al.*, (2006), as highlighted by the reviewer. Our model targets woody species (i.e. > 70% of all isotopic studies, Rothfuss and Javaux (2017)), and it is therefore not appropriate to speculate about non-woody species. We re-formulate our statements in the new version of**

**the manuscript (i.e. expressions such as "to put in danger" will be dropped and replaced by more informative and appropriate formulations).**

3- The authors provide no information about the soil compartment; what about the soil water isotopic composition profile temporal and spatial variabilities? Are the isotopic differences in xylem water reflected by the span of isotopic composition values in soil water? This would offer the possibility to rule out possible evaporation effects mostly during sampling and transport (which is not listed as other reasons for the observed diurnal variations of δxyl). If soil water isotopic information is not available, it should be stated as limitation of the study;

**The empirical data collection indeed has the limitation of the absence of soil characterization during field setup (i.e., soil water potential and isotopic composition of soil water), which will be clearly stated in the new version of the manuscript.**

**In addition, evaporation effects during sampling and transport can never be excluded in field studies but have been minimized by the applied protocol, as detailed in the manuscript. We will add evaporative effects during sampling and transport as a potential reason of additional variation in the discussion section. However, we expect evaporation effects to be low because of (a) the imposed strict protocol. Specifically, fast sampling with cautious care to avoid heating the extraction instruments, was followed by fast capping of the sample vials (sealing caps with rubber and glass vials having a minimum of two full closing coils), immediate cooling of the vials in the field, and freezing of vials upon return in the lab; and, (b) cross validation of the obtained $\delta^2H_X$ with other isotope experiments performed during the dry season at Laussat which do suggest that our $\delta^2H_X$ lie within the natural span of the soil water. These observations, although not related to the paper itself, can be provided in supplementary material.**

4- I found on several occasions that the authors did not fully understand basic principles driving isotopic fractionation (see my specific comments);

**We regret that we left the impression of a less than full understanding of the basic principles behind isotopic fractionation. We removed all instances of careless representation and wording in the new version of the manuscript (see our answers to the specific comments for more details).**

5- In general, I do not think that such field experiments, where a significant number of environmental driving factors are unknown, should be used to question the entire isotopic research methodology. I urge the authors to discuss this point as well and measure their words.

**We agree with the reviewer and have toned down the message to be more in line with the uncertainties in the data. We now realize that our original tone could have been perceived as questioning the entire isotopic research methodology – which was not our intention. Therefore, in the revised manuscript, we will use more appropriate statements, as well as including some positive aspects of diurnal variability in $\delta^2H_X$. These could present new opportunities in water acquisition and plant performance studies. However, we remain convinced that our findings indeed show the need for caution when applying isotopic research methodology in multiple situations and configurations, as large variability of stem isotopic composition are expected and could plausibly lead to significant bias in**

**average RWU depth determination. Our main objective remains to build increased awareness of the potential of diurnal variability to bias future isotope endeavors.**

The authors will also find a list of specific questions/remarks/corrections/issues:

L24. What does "i-H$_2$O-xyl" refer to? To "plant xylem water uniform isotope composition" or "plant xylem water isotope composition"? In either case, "δ" is to be preferred over "i-H$_2$O-"
**This is a good suggestion. The "δ"-notation suggested by the reviewer will be adopted, as i-H$_2$O-xyl originally referred to the plant xylem water isotope composition.**

L32-33. "field data show excessive i-H2O-xyl variation during the day or along stem length ranging up to 25.2‰ in δ2H and 6.8‰ in δ18O" does not read well. I propose something like: "the hydrogen (oxygen) isotope composition of plant xylem water showed strong temporal (i.e., daily) and spatial (i.e., along the stem) variation ranging up to 25.2 (6.8) ‰´'
**This sentence will be adjusted accordingly.**

L36. Please rephrase: "danger" is not the proper word.
**This will be adjusted accordingly:** *"Our work shows that the fundamental assumption of uniform δ$_{xyl}$ is violated, both theoretically and empirically, which might generate significant biases when using stable water isotopes to assess RWU under certain field conditions".*

L46-47. There is no such thing as the depth of root water uptake in the case of several soil water sources. Only in the context of direct inference is this true. But the authors do not refer to the later (and outdated) technique.
**We agree that the terminology: 'average root water uptake depth' is more appropriate. This will be implemented accordingly throughout the manuscript.**

L49. This is not true: the isotopic technique is of course destructive (you have to take a soil core), very labor intensive (e.g. extraction of soil and plant xylem water).
**This statement presents a comparison with root excavation endeavors, which are extremely time consuming, laborious and destructive. We adjusted the statement to read that in comparison to root excavation, isotope techniques are far less destructive and time consuming, and are hence definitely preferred when studying multiple individuals at once.**

L50 (also L47). You should mention that it is fractional RWU and not absolute RWU you are talking about. You cannot solve for water mass balance with the isotope technique, which constitutes its greatest limitation when compared to other techniques.
**This will be adjusted accordingly.**

L52. How would you determine fractional root water uptake at the ecosystem level?
**This statement, which is taken from Dawson *et al.*, (2002), can - for instance - embody δ$_{xyl}$ analysis performed on the dominant tree species of a forest stand (i.e. forest stands on Mount Kilimanjaro - Bodé *et al.*, 2019), or on classified groups of plant individuals (juvenile versus adult – Stahl *et al.*, 2013; liana versus tree – De Deurwaerder *et al.*, 2018). These measurements inform on the average depth of water acquisition (i.e., "strategy") of the species/group, which can then be extrapolated to estimate the expected dynamics/strategies at the ecosystem scale.**

L56. This should be "δxyl". Why the "i" instead of "δ" here? Also, why the "H2O"? (is there another molecule investigated here?)

**As indicated in previous comment, we will adopt the "$\delta_{xyl}$"- notation suggested by the reviewer.**

L58-60. Peclet effect is measurable in the xylem vessels upstream of the evaporative sites. This assumption is not systematically made. Instead, authors investigate the prevalence of isotopic fractionation depending on the plant tissues they sample, e.g. in Barnard et al. (2006). Please revise.

**This remark of the reviewer will be addressed by (i) emphasizing that this study targets woody plants, as non-woody plant are indeed subjected to "stem" fractionation processes (Barnard et al, 2006, a reference which will be included), and notify that (ii) the Péclet effect might be observed in branches upstream of evaporative surfaces. The later presents a rather local phenomenon and should not, or very limitedly, impact stem samples at distance from the evaporative surface, as performed in this study. If impacted, an upstream enrichment can be expected (we will add that potential impact in the discussion as well).**

L67-69. There can only be kinetic fractionation playing a role during the transport of water through the root membranes, since there is no liquid-vapor phase change that would involve equilibrium fractionation. Please revise.

**The reviewer is correct, and this will be revised accordingly**

L69-72. Not only kinetic fractionation is a result of the difference in mass of the water isotopologues, but fractionation in general (e.g., equilibrium and kinetic fractionations).

**This is correct, and was unfortunately dropped out during editing of this version of manuscript. We revise our definition accordingly emphasizing that this entails transport of water through a root membrane.**

L94-95. Why would you make the assumption that δxyl is constant over time (over which period of time anyway)? At this point of the MS, it is not clear. Actually, no one makes this assumption in the field, rather they sample from e.g. the base stem among individuals at e.g. a sub-hourly temporal resolution and sub-daily temporal extent.

**We agree with the reviewer that we should be more precise in our formulation of the hypothesis and the time frequency considered (sub-daily and even sub-hourly). This will be addressed in the new version of the manuscript. We also note that there may be a misunderstanding here regarding the assumption made in the field.**

**It is indeed correct that a few high frequency measurements of $\delta_{xyl}$ exist. However, it should be noted that these are (a) rather rare at the moment; and (b) predominantly target sampling of the leaves. Sampling of leaves, however, is less relevant to the 'isotopic tracing technique for RWU assessment' as multiple other processes impact the isotopic composition of leaf water (i.e. the aforementioned Péclet effect). In this study, we do not address leaf water monitoring because of the decoupling between source water and measured signature. To date, most studies where isotopic composition of xylem is used for RWU assessment have - at best - a daily, but more often a monthly or seasonal temporal sequence. Moreover, many of the studies (including ours) consider only one-time sampling (including ours, see e.g. De Deurwaerder *et al.,* 2018). These studies do assume a constant**

$\delta_{xyl}$ over time. Hence, sub-hourly/daily $\delta_{xyl}$ variances are generally not accounted for in studies on lignified stem sampling.

**Finally, we acknowledge that coring close to the base of an individual stem is generally applied in non-woody, herbaceous plants as recommended by Barnard et al., 2006. However, to the best of our knowledge, this is not standard practice in woody plants. We acknowledge that it might be more general than we know, as implied by the reviewer, but this is not reflected in the existing literature where height of coring is rarely provided, and when so, coring is generally performed where stem diameter is measured (i.e. 1.3m in metric system, and at 4.5 feet in imperial system) (e.g. White *et al.*, 1985; Meinzer *et al.*, 1999; Goldsmith *et al.*, 2012; Hervé-Fernández *et al.*, 2016; De Deurwaerder *et al.*, 2018; Muñoz-Villers *et al.*, 2019)**

100-101. What do you mean by "diurnal changes in the soil-plant-atmosphere continuum"? Which changes?
**This statement indeed needs further clarification, which will be pursued in the new version of the manuscript. In short, with "diurnal changes in the soil-plant-atmosphere continuum" we imply: changes in water potential differences between leaf and soil along the day. These gradients will determine the vertical distribution of root water uptake.**

L113. What exact "water potential gradients" do you refer to?
**Here, we refer to the water potential gradient between soil and the evaporative surfaces (leaves) of the plant. This will be added to the manuscript.**

L114-116. Why would you need to use a mixing model, especially since you did not sample soil water and determine its isotopic composition? You may as well simulate a sinusoidal pattern for the $\delta xyl$. Please elaborate/explain.
**We indeed did not measure soil water ourselves in this study, but target a model representation which is practically implementable and repeatable. Here, the approach of Phillips and Gregg (2003) presents a widely used and implementable approach to mathematically represent fractional water uptake in the soil. The apparent sinusoidal pattern of the xylem water isotopic composition observed by the reviewer results from the diurnal fluctuations in leaf water potentials and corresponding changes in the distribution of the root water uptake in the various soil layers. Specifically, here, the pattern in leaf water potential is imposed by a bell shaped sap flow curve obtained from Huang *et al.* (2017). Hence, this apparent sinusoidal pattern naturally emerges from the source mixing model approach and was not hard coded as such in the model.**

L117-119. You should write the isotopic equations with "$\delta$" instead of "$\delta 2H$" as the model does not focus on $1H2HO$, to the contrary of what the authors say. For the model to focus on $1H2HO$, it would mean that $1H2HO$ and $1H218O$ would follow different physical processes, which is not the case (both isotopologues undergo mass dependent fractionations, i.e. $\varepsilon eq(2H)/ \varepsilon eq(18O) \approx 8$ and $\varepsilon k(2H)/ \varepsilon k(18O) \approx 0.88$). Also write 2H instead of Deuterium and do it consistently throughout the MS. The latter is just an element's isotope and does not deserve (anymore) its own letter (see IAEA tech reports guidelines).
**We will adopt the notation suggested by the reviewer throughout the manuscript, and we will rephrase the statement as we indeed focus on the water isotope element instead of the water isotopologue as inaccurately implied in the manuscript.**

L130-131. This assumption is only reasonable when soil water redistribution no longer occurs, e.g., this does not stand shortly after a rain event.

**For the sake of simplicity we present a model which assumes rain-free periods and prevents soil redistribution, as indicated by our statement L130-132: "… *a reasonable assumption if the isotopic measurements are conducted during rain-free periods, …*". We acknowledge that this assumption can be presented more clearly in the new version of the manuscript.**

L125ff. Report the dimension of each variable and parameter throughout the MS.

**For readability of the text, we prefer the use of a dedicated table listing the variable/parameter dimensions all together, as done in Huang *et al.* (2017) (here Table 1, as indicated in the text L123).**

L142. From Eq. (3), I understand that the water "potentials" are in fact "hydraulic heads". This should be clarified.

**This will be clarified in the next version of the manuscript by using hydraulic head (or soil matric potential) instead of the confusing and generic potential term, whenever appropriate.**

L143-144. Add here that $k_i$ and $\Psi_{S,i,t}$ are also specific to the ith soil layer.
**This will be adjusted accordingly.**

L193-196. I am missing background information to understand what the "30 days sequence" of the "model runs of Huang et al. (2017)" refers to. Please elaborate.

**We agree that our statement is unclear for readers that are not familiar with the paper of Huang *et al.* (2017) in which a 30 day drought simulation study of loblolly pine was conducted. An average day within this representation was selected based on both representativeness and data availability. We will elaborate on this topic in more detail in the new version of the manuscript.**

L201-203. Why would you need external data (Meissner et al. 2012) and not simply do your model exploration on basis of a synthetic experiment?

**The reviewer is correct as applying a complete synthetic experiment is indeed possible. We chose to use the Meißner *et al.*, (2012) data as this presents a realistic dataset (in terms of range and variation in both soil water potential and soil water isotopic composition) obtained during field studies, and therefore find it relatable for both interpretation as well as providing insights for model requirements guiding field setups.**

L208-209. I did not hear of such standard practice and I doubt there is. Could you add a reference for this?

**Here, we assume researchers followed standard procedure in using an increment borer in forest inventory, i.e., coring where stem diameter is typically measured (i.e. breast height: at 1.3m according to the metric system, in at 4.5 feet according to the imperial system). This method has been applied multiple times (e.g. see White *et al.*, 1985; Meinzer *et al.*, 1999; Goldsmith *et al.*, 2012; Hervé-Fernández *et al.*, 2016; De Deurwaerder *et al.*, 2018; Muñoz-Villers *et al.*, 2019), but we agree that it does not need to be presented as a standard practice, as several isotope tracing studies applying an increment borer to collect xylem cores do not specify the height of coring. We also acknowledge that (a) many studies sample branches (ignoring the effect of evaporative enrichment from the leaves to upstream plant organs), and (b) that our assumption that researchers follow the standard**

**increment borer approach could be incorrect. We will rephrase this statement, providing the here presented references in support of the followed approach.**

L223-225. Split the sentence and add detail. It is hard to understand. Also following the Rayleigh distillation model, the error should always be negative in the case of incomplete water recovery, which does not match to your normal distribution of error in the null model.
**The indicated sentence will be split up and clarified. We thank the reviewer for this excellent suggestion. It is true that the expected error should be negative, which we can easily implement in the model structure by using a truncated distribution instead of a normal one. This will be implemented in the next model version.**

L228-229. How so? And why would it be relevant to take into account the analyzer systematic error at this point of your model testing?
**Analyzers always have an embedded error which is generally very small. But, if known, the user can opt to implement these in SWIFT model. In this study - as indicated in L229 - we consider these errors negligible and have indeed ignored them as it has little relevance in the model testing at this point, as indicated by the reviewer. For sake of clarity, this sentence will be removed.**

L235-244. Are you talking about RWU depth of rooting depths here? How do you define the latter term? Why would you use the direct inference model (which is a very simplistic view on RWU, i.e., one single root sampling from on single layer at a time) if you use a multi-source mixing model (Phillips and Gregg, 2003), which allows the plant to sample simultaneously from different layers? Please explain this apparent contradiction. Overall this section is quite difficult to read and I ask that the authors simplify it.
**Here we are talking about average depth of RWU (i.e. a weighted mean of the depths of root water uptake, with the root flows at the different depths as weights), and hence, this section/title will be changed accordingly for clarification. We applied both methods for completion of the presented study, as combined, they embody 96% of all applied methods (Rothfuss and Javaux, 2017). While direct inference might be considered as very simplistic, to date, it remains the most applied technique in literature (46% according to Rothfuss and Javaux, 2017). In short, the direct inference approach compares hydrogen and oxygen isotopes between the soil water profile and the xylem water of the stem. The depth of soil water having similar isotope values to the stem water indicate the main depth of soil water sources used by the plant (e.g. see Wang *et al.*, (2010)). Hence, this approach does not exclude that the plant can take up water from multiple soil layers, but just assumes that the signature found in the xylem reflects the dominant signature of bulk water uptake. It is therefore unclear to us what the reviewer exactly means with "the apparent contradiction".**

L258. Is there a specific reason why you did not use Van Genuchten's soil retention curve?
**There is no specific reason not to use the van Genuchten's soil water retention curve. As we do not know the soil hydraulic properties at the site (soil retention and conductivity curves), we do not have any reason to prefer Clapp and Hornberger (1978) close-form equation instead of the Mualem-van Genuchten model. We will implement more soil hydraulic models (including Mualem-van Genuchten) in future model versions.**

L309. Delete "kinetic". It is not even sure that you would have fractionation at all, considering that you may boil (==fractionation free process) the water here rather than evaporate it.
**This will be adjusted accordingly.**

L321. Fresh weight does not take into account possible loss of water during transport/ storage. You should have weighted the samples prior extraction again.

**We agree with the reviewer that measuring before extraction itself could provide extra information on whether or not water was lost during transport/storage of the samples. We did not do this, and can therefore not provide such insights to the reader.**

**However, as we used glass vials with sealing caps (including sealing rubber, and at least two complete loops of closing coil), water losses during transport and storing should be negligible/absent. Besides, it should be noted that measuring samples after storage, i.e. before extraction, might itself impose sample contamination and inaccurate assessment of the percentage of water extracted by the cryogenic water extraction method. Specifically, frozen vials will attract frost and condense water onto the vial exterior, which can substantially impact the weight of the vial itself. This then should be accounted for, for instance by warming the samples to room temperature before weighing, a practice which arguably is also not recommended.**

L331-336. Since you are measuring with a Picarro, which does not give ratio (but performs already the delta conversion), you need to say that you "corrected the Picarro raw delta readings into calibrated delta values thanks to the values of the aforementioned 'internal laboratory References' expressed on the international V-SMOW scale". No need to display the equation (12) but you may detail these "internal laboratory References" (e.g., value).

**The suggestion of the reviewer will be implemented in the new version of the manuscript.**

L336-334. Still at this point, I do not know what the difference is between i-H2O-xyl and $\delta$xyl: : :If there is none, please use the latter term. In addition, use another letter than $\varepsilon$ for the normalized "i-H2O-xyl": it usually stands for isotopic fractionation, defined as the deviation of the fractionation factor to unity. It seems even odd that you would consider such a letter…

**As indicated above, we will replace the symbol of i-H$_2$O-xyl by $\delta_{xyl}$ as suggested by the reviewer. In addition, we agree with the reviewer that our choice to use $\varepsilon$ here was unfortunate. While '$\varepsilon$' is commonly used in statistics to indicate bias in sample set, we now see that this indeed can result in confusion for the isotope community. Hence, another Greek letter (i.e. '$\beta$') will be used in the revised manuscript.**

L369. I still do not understand what is the concept of RWU depth if you consider the multi-source mixing model approach.

**As indicated above, we will clarify this definition. Throughout this study we consider 'average depth of root water uptake' (i.e. a weighted mean of the depth of root water uptake, with the root flows at the different depths as weights), as will be adjusted in the new version of the manuscript.**

Fig. 2. Panel (a): how do you come up with a night $\delta$xyl at 1.3 m above −60‰ Also, I don't see why panels (a) and (d) look so different for day 1, since if I understand correctly, the cumulative SF is a function of time (if sap flow remains constant).

**The patterns in $\delta^2$H$_X$ results of both the isotopic composition of the water taken up by the roots at any time – and – the volume displacement of water moving as a slab along the tree stem. At each time step considered, a specific volume of water and isotopic composition is extracted by the plant. This presents $\delta^2$H$_X$ at stem base which is not limited by volume of the tree yet. However, this quantity of water is subsequently pushed, as a slab, upwards in the limited volume of the stem, i.e. our model presents a piston-flow**

**approach. At this point, the quantity of water taken up by the plant also impacts the observed δ²H_x pattern.**

**Specifically, the water movement within the tree can be visualized by 'a stack of disks' of water each having a time specific δ²H_x, where stack-height is defined by quantity of water taken up and the stack area corresponds to the lumen area of the tree. Step by step, new disks are introduced at the bottom, pushing previous disks upwards, i.e. water moves as a slab through the stem. When root water uptake activity stops, i.e. sap flow is zero, the stacks remain at their respective position. When measuring at 1.3m height, the entire volume of water taken up in the late afternoon, with values of -66‰ is simply to small too reach the measurement height. For this reason, δ²H_x at 1.3m represents the water isotopic composition of water taken up earlier during the day (i.e. around midday), which has a more enriched isotopic composition.**

L371. "isotopic composition of soil water is dominated by depleted deuterium". Please correct phrasing: soil water can be depleted in 2H in comparison to another water volume, but there is no such thing as "depleted 2H".
**This will be corrected accordingly.**

L373. An isotopic composition, which is a number, cannot be "enriched". Please correct.
**This will be corrected accordingly.**

L375. "depleted deep soil water"
**This will be corrected accordingly.**

L384. ": : :RWU originating from deeper, more depleted soil layers". Please correct: water from a given soil layer might be depleted, not the soil layer in itself.
**This will be corrected accordingly.**

L399-400. This belongs to the discussion section.
**This sentence will be moved to the discussion section.**

L407-418. Nowadays no study is published where RWU depth is investigated with the direct inference method. Analyses are performed with Bayesian mixing models. So I wonder if this section, although interesting theoretically, would benefit practically to the community.
**It is true that Bayesian mixing approaches become more commonly used in current literature. However, we argue that the potential issues in RWU assessment unraveled in our research apply to all existing literature, of which direct inference method still embodies the majority of studies (see Rothfuss and Javaux, 2017). For this reason, we are convinced that this section can be relevant when critical assessment of former studies is pursued. We will emphasize this more clearly.**

L446 and Fig. 4. See my previous comment on the use of "ε". The caption of a figure should not point to another figure or table. Write here the name of the species (no need to write them in the figures though).
**This suggestion will be implemented in the figure, and the notation "ε" will be changed.**

L452. Add in the text that growth forms refer to lianas and trees.
**This will be adjusted accordingly**

L455-457. This belongs to the discussion section. Also the link between "easily accessible and abundant groundwater reservoir" and the fact that the diurnal intra-individual variance is minimized is not clear. I suggest moving to the discussion and elaborating on this.

**These sentences will be moved to the discussion section.**

L471-472 and Table 2. How many individuals (which you could consider as replicates) of each species were sampled during the experiment? Discuss the implication of having n=1 with respect to $\delta_{xyl}$ variance.

**It is true that only one replicate per species was obtained for this study. That was because we did not target the intra- or interspecific variances in $\delta_{xyl}$ in our experimental protocol but instead we investigated the intra-individual $\delta_{xyl}$ variability, and the theoretical exploration of a likely cause of this phenomenon.**

L486-492. The authors say that the intrinsic problem of the "isotopic tracing method" is that there is a soil water isotopic gradient in case there is evaporation and under heterogeneous soil water potential gradient? I don't understand this at all (!) The isotopic methodology for studying plant RWU relies on heterogeneous isotopic gradients in soil water. This is a solution, not a problem here...

**We acknowledge that the text was not clearly formulated in support of the argument envisioned. What we wanted to convey is that the soil water conditions required to perform the 'isotopic tracing method', also facilitate a large variance in $\delta^2H_X$, which could have important consequences for the RWU assessment. An altered, non-ambiguous discussion will be presented in the new version of the manuscript.**

L493-506. I disagree. There is a clear problem in determining fractional RWU profiles on basis of measurements of the transpiration isotopic composition, which is highly temporally dynamic and spatially heterogeneously distributed; many observation of leaf water confirm the non-reaching of isotopic steady state. In addition, how would a "change of cloud cover degree" have an "instantaneous" influence on $\delta xyl$? This contradicts the results of your synthetic experiments, where depending on sap flow rate, there is a marked isotopic memory effect of the antecedent water moving upward it the xylem vessel.

**Cloud cover will result in reduced water environmental demand and thus impact the sap flow velocity (and thus the water and isotope dynamics in the stem). Hence, cloud cover of a tree will reflect in distinct patterns of RWU uptake dynamics and the bulk isotopic composition of water extracted from different soil layers. The statement does not contradict our model findings but we acknowledge that the presentation could have been more clear. What we wanted to say is that the intra-individual variability of $\delta^2H_X$, according to our model simulations, reflects indeed the past changes of root water uptake dynamics (including due to dynamic changes of environmental demands).**

L516-523. The model provides an explanation, sure, but does not validate your hypotheses from the confrontation with experimental data. This is missing from your study and should be mentioned.

**We fully agree with the reviewer and acknowledge that we have overstated our findings. The next version of the manuscript will clearly describe limitations of our study. This will also be clarified by the paper structure that will change: we will present our model simulations as a potential explanation of the isotopic composition variability observed in the field.**

L534-546. My understanding from the literature is that hydraulic redistribution is intermittent and localized, thus does not affect that much the bulk soil water isotopic composition, rather it affects the direct environment of the roots.

**Correct, we agree with the reviewer that hydraulic redistribution will predominantly impact the rhizosphere of the plant, rather than the isotopic composition of the water in soil layers. We will rewrite this paragraph as such. The main message within this paragraph, suggesting that hydraulic lift will reduce the $\delta_{xyl}$ variance, remains valid as the variance of isotopic composition of water accessed by the plant can be reduced.**

L578-587. Not to forget we need to monitor soil water isotopic composition to verify if $\delta xyl$ spreads within the range of isotopic values observed in the soil profile.

**Correct, and we will add this suggestion to the new version of the manuscript.**

The manuscript by De Deurwaerder and colleagues challenges the idea that, in absence of precipitation or other rapid changes in climate, the water isotope composition of plant xylem should stay fairly constant over diurnal time scales or along stem height. Their analysis is based mostly on a model (!) of root water uptake and isotopic transport within the roots, up to the stem base. Their model considers that (1) the isotope composition of stem water at the base of a tree ($\delta^2 Hx_{(0;\,t)}$) is the average isotope composition of soil water over the root zone, weighted by the fractional root water uptake rates at each depth (Eqs. 1 or 7) and (2) the isotope composition of stem water at any height h ($\delta^2 Hx_{(h;\,t)}$) is the isotope composition of stem water at the base, delayed by the travel time $^\tau$ of sap between stem base and height h ($\delta^2 Hx_{(h;\,t)} = \delta^2 Hx_{(0;\,t-\tau)}$) (Eq. 9). Soil properties are used as boundary conditions that do not change over the day in terms of soil water potential and isotopic composition. With such model, they predict large diurnal variations of xylem water isotopes at stem base, but also large variations along the stem (see their Figs. 1 and 2). Based on this modelling exercise, and separate observations of the $^2H=^1H$ ratio in water extracted from tree stems and lianas at different heights within a tropical forest canopy, and showing some scatter sometimes larger than 3‰ (the estimated error from water extraction and isotope analysis), they conclude that (1) the common assumption that the isotope composition of stem water is fairly constant over time is violated and (2) it can cause significant biases when using water isotopes to identify plant water origin.

I think it is good that the authors bring forward the point that xylem water may sometimes exhibit rather dynamic variations in its isotope composition. However, I am afraid the proposed model is inadequate and the dataset is too limited for illustrating this point. To me, the study does not prove anything; it shows that there are variations in the data and that there are variations in the model but there is no model-data comparison. Besides, variations in the data are not very large and can be explained by lots of other processes, and variations in the model are mostly caused by its lack of realism. These two points are explained in more details below.

**We thank the reviewer for his/her detailed assessment of the study and helpful feedback on the manuscript. We do want to stress that our study is based on both a theoretical model exploration and empirical, novel field data. Moreover, we present two independent datasets (a dataset collected in French Guiana, and a dataset collected in China), which both show excessive variability in isotopic composition. We do not agree that all these data can be dismissed easily. Additionally, we believe that all models can be criticized due to a lack of realism but their value depends on the insights they bring. In fact, process-based model explorations are proven tools in many scientific fields, because of the insights they provide and not because of their subjective realism. We hope to convince the reviewer that adding supplementary processes would indeed improve model realism and might impact the dynamics and absolute range of $\delta^2 H_X$, but it will not alter our conclusion: large variability of isotopic composition along woody stems is expected in many situations. Moreover, including some of the suggested realism strengthened our results.**

**We agree with the reviewer, however, that the message brought in our original manuscript should be toned down to better reflect the limitations of the analysis and data. Therefore, we propose to revise our manuscript, providing a more appropriate message by (i) down toning our statements and by (ii) including the positive aspects of diurnal variability in $\delta^2 H_X$, which could present new opportunities in water acquisition and plant performance studies. At the moment we are unable to fully validate the model with the dataset obtained. However, the presented model serves as a theoretical exploration of one**

possible explanation that could cause the observed variance in $\delta^2 H_X$. Here we apply generally accepted plant hydraulic processes which shows that large variance in $\delta^2 H_X$ is expected under the simulated (and realistic) conditions. We remain convinced that our findings, though not conclusive, can help build increased awareness of the potential of diurnal variability to bias future isotope endeavors. At a minimum it calls for more research. To meet the concerns of the reviewer, the new version of the manuscript will more clearly mention the limited coupling between model and data, as also suggested by the reviewer 1. We acknowledge that other mechanisms could contribute to the observed $\delta^2 H$ variance, and will extend existing sections on other processes. Future model developments and targeted datasets are encouraged, as mentioned in the text, and will be highlighted even more in the new version of the manuscript.

In short, we agree with most of reviewer's comments, but we do not share his/her conclusion on the data and model.

- **The French Guiana dataset (i.e. measuring isotopic composition along stems of lianas and trees) is indeed limited but is the first of its kind and does show intra-individual variations that are too large to be explained by extraction error only. We do not think that variances up to 20‰ $\delta^2 H$ in a natural system should be considered as negligible.**
- **The model lacks realism for certain processes. It however does provide the insight that naturally arising changes in evaporative demand should lead to isotopic composition variability in woody stems due to their coupling to variable isotopic and soil water potential gradients (which are the basis of the isotopic studies). Adding model complexity, as the reviewer suggests, would allow us to refine both the ranges and the dynamics of the variation but will not prevent it. As we illustrate using the reviewer's suggestions below.**

Hence our main objective (to illustrate the fact the xylem water isotopic composition does exhibit dynamic variations) still stands.

The dataset accompanying this study only consists of a few water isotope data from tree stems and lianas collected over a couple of days. No soil water data is shown, or even sap flow or rooting depths. I doubt it is the best dataset to test the proposed theory, or draw any conclusion about plant water uptake. The data shown in Fig. 5 is interesting but it comes from another study (Zhao L, Wang L, Liu X, Xiao H, Ruan Y, Zhou M (2014) The patterns and implications of diurnal variations in the d-excess of plant water, shallow soil water and air moisture. Hydrology and Earth System Sciences,18, 4129–4151). Many processes (stem evaporation, different proportions of storage tissues or even atmospheric vapour use) and measurement artefacts (during sampling and transport, water extraction, isotopic analysis…) could explain significant variations in the water composition of stems from trees and lianas of different statures. Accounting for uncertainties in the extraction and analysis is certainly not enough.

We will more clearly discuss such limitation in the new version of the manuscript.

This study indeed presents data collected by Zhao *et al.*, (2014). We note that this study is in collaboration with the authors who collected the data (please see the author list). The data we present show high temporal xylem water observations not presented in Zhao *et al.* (2014). This type of data is very rare in literature. Zhao *et al.* (2014) focused their paper

on d-excess variability throughout the day, which is a derivative of δ²H and δ¹⁸O data. In our paper, we provide the raw δ²H and δ¹⁸O temporal data.

**Finally, we note that the factors the reviewer mentions were included in our discussion section. Not all of them will be an issue, while others will exacerbate bias. We will expand this discussion section to address the concerns of the reviewer. Here in short:**

- **Stem evaporation: This is indeed a good suggestion for non-woody plants. However, we target woody/lignified plants (this might have not been stated clearly enough, but now will be). Stem evaporation, especially when measuring relatively low in the stem (at 1.3m), should be negligible.**
- **Different proportions of storage tissues: We fully agree with the reviewer that this presents a potential explanation of observed patterns, as discussed in the discussion section "iii. *Storage tissue and phloem enrichment*" (L 547-564).**
- **Atmospheric vapor use: The reviewer presents another excellent argument why applying the 'tracer isotopic technique' and sampling protocols should be re-assessed, addressing large variances in δ²Hx and all potential contributing factors. We can include foliar water uptake and implications on the variability in δ²Hx in the discussion of the manuscript. However, the quantitative contribution of foliar water uptake is rather limited, and impact on δ²Hx will mainly restrict to leaves and upstream branches, and will generally be negligible in the plant stem, the main focus of our analysis.**
- **Sampling protocol and extraction procedure: While sampling protocols and extraction procedures are never perfect, our extraction protocol is based on best practice as suggested by Orlowski *et al.*, (2013). Hence, considering extraction error rates of 3‰ are very cautious estimates, and actual error is most likely much less. Here we remark that despite these cautious estimates, we observe significant variability (as compared to the null model), which is remarkable and should be reported.**

More importantly, I find the modelling analysis flawed and totally unrealistic. As explained above, the proposed model simulates water isotope gradients along the stem based on the average travel time of sap between two stem points (i.e. assuming the water isotope composition of xylem water at height h is that at stem base at an earlier time corresponding to the travel time between stem base and height h). By doing so, the model neglects the mixing of water isotope by diffusion during water transport. If we neglect pit structure and consider vessels as regular pipes, the Péclet number $\wp$ that compares advection and diffusion is, using their notations: $\wp = SFV\,h = D_l$. Taking an average sap flow velocity of $SFV = 0.3\,mh^{-1}$ (see caption of Fig. 2) a typical height (diffusion length) h of 1m and a self-diffusion of liquid water of $D_l = 2.5\ 10^{-9}\,m^2s^{-1}$ this leads a Péclet number $\wp$ around 30000, i.e. high enough to justify neglecting (a posteriori) water mixing by diffusion. However, mixing with storage tissues also occurs and tree sap does not move like a slab. In their model, as soon as transpiration stops, root uptake stops and sap flow at any height stops too so that the $\delta^2H$ of xylem water at any height remains to its value at dusk over the entire night until the following morning plus the time delay $\tau$ (see for example Fig. 2a, the curve for h=1.3m). In reality, at night, sap flow does not stop immediately because plant elastic tissues need to be replenished. Root uptake will continue until full replenishment of the elastic tissues is done. This will contribute to homogenisation of xylem water over night. Also when sap flow becomes small diffusion is not negligible anymore (low Péclet), which will reinforce the isotopic mixing by water diffusion. In other words, in the real world, xylem water

should not exhibit large isotopic gradients along the stem such as shown in their Figs. 1 or 2. Mixing with storage tissues is briefly discussed (section 4.3.iii) but not in the same direction as above. If night-time mixing of xylem water in roots and stems was accounted for, this should strongly minimize the predicted diurnal variations of _2Hx(h; t), even at stem base. Not accounting for diurnal variations in soil conditions (water potential and isotopic composition) is also a strong limitation of the model.

**We thank the reviewer for this excellent suggestion to use the Péclet effect to support neglecting diffusion in the model when sap flow is large enough. We have now performed an analysis to evaluate the impact of diffusion at night, when sap flow is zero (Fig 1a), as assumed by the model at night (i.e. Péclet number becomes low as advective flow rate goes to zero, while diffusive flow rate remains constant, hence flow is dominated by diffusion at night). As the diffusion coefficient is low (i.e. $D_l = 3 \cdot 10^{-5}$ cm² s⁻¹) the impact of diffusion at night is mostly negligible (see Fig 1a where 12 hours of diffusion results in a smearing of the signature ± 4cm). Diffusion causes an increase in the width of the $\delta^2H_X$-baseline drop, which means that the probability of sampling a non-representative section within this $\delta^2H_X$-baseline drop will increase. Including more realism hence increases bias in RWU estimates. It should be noted that diffusion will indeed reduce the absolute range of variability in $\delta^2H_X$ over time (but very slowly and very little), and hence with height of the plant. However, it will not lead to homogenization overnight (which is by the way not observed in our supporting datasets). This result and corresponding figures will now be implemented in the manuscript. In addition, we like to stress that for simplicity of the theoretical exploration, we deliberately chose for zero sap flow at night, as indicated in the model description. However, the model is flexible, and direct sap flow data complying the wishes of the user can be implemented without problem.**

**As we explored in the discussion section (see section "*Storage tissue and phloem enrichment*"; L546-564), homogenization of xylem water over night depends on the assumption that storage water is representative of the water taken up by the roots. In fact, for a number of reasons, the isotopic composition of storage tissues is likely to deviate from the isotopic composition in soil water. This decouples the isotopic signature observed in xylem from the isotopic composition of the water mixture obtained by RWU, and exacerbates potential bias in the isotope tracing technique. Unfortunately, empirical data on isotopic composition of storage tissue is absent in literature to our knowledge, and this hampers theoretical exploration of such an hypothesis. We will highlight more clearly the importance of research targeting evaluation of the impact of storage water use by future studies, which would then allow implementation of storage tissue in the model. However, the presented empirical data itself do not show any indications of complete homogenization despite obtained from lianas and trees during dry season. This suggests that either storage tissue does not completely succeed in homogenizing $\delta^2H_X$ overnight as suggested by the reviewer. Therefore, in our opinion, the diurnal root water uptake fluctuation still remains a convincing explanation for the observed variability in $\delta^2H_X$.**

**Finally, the reviewer is correct in pointing out that the absence of diurnal variations in soil conditions (water potential and isotopic composition) presents a limitation of the model. But this is already discussed in the discussion section "*Temporal and spatial soil dynamics*" (L534-546). However, temporal and spatial soil dynamics are generally very small given (a) the timeframe and (b) conditions in which stable isotopic tracing technique**

**are studied, i.e., one day sampling during dry conditions without rain are generally preferred. Hence, for all these conditions, the simplification of our model is acceptable in our opinion. Besides, our model implementations are flexible and if variable soil condition data are available, they can easily be implemented.**

In conclusion, I find the argument raised by De Deurwaerder and colleagues not supported by their data nor by their model simulations. More realism would need to be brought to the model and the dataset should be complemented with additional information before drawing any conclusion on how variable the isotopic of xylem water in tree stems and lianas is over diurnal time scales or with height.

**We understand the reservations of the reviewer for this study, as coupling between data and theoretical model was not fully possible. Therefore, we will tone down statements in the manuscript to better represent the limitations of the data and models. We do not agree that our data can be dismissed easily: we stress that these are two independent datasets that both show excessive variability in δ²H$_X$, which is illustrated for the first time. In addition, we also stress that the processes that reviewer suggested to increase realism does not change our conclusion itself: along the stem of woody plants, we can expect changes of water isotopic composition. We believe that arguments raised by both reviewer's present additional incentives to re-assess and therefore further refine and improve the stable isotopic tracer technique.**

**REFERENCES**

**Barnard RL, De Bello F, Gilgen AK, Buchmann N**. **2006**. The δ18O of root crown water best reflects source water δ18O in different types of herbaceous species, Rapid Commun. Mass Sp., 20, 3799–3802.

**Bodé S, De Wispelaere L, Hemp A, Verschuren D, Boeckx P**. **2019**. Water-isotope ecohydrology of Mount Kilimanjaro. *Ecohydrology*: e2171.

**Dawson TE, Mambelli S, Plamboeck AH, Templer PH, Tu KP**. **2002**. Stable isotopes in plant ecology. *Annual review of ecology and systematics* **33**: 507–559.

**De Deurwaerder H, Hervé-Fernández P, Stahl C, Burban B, Petronelli P, Hoffman B, Bonal D, Boeckx P, Verbeeck H**. **2018**. Liana and tree below-ground water competition—evidence for water resource partitioning during the dry season. *Tree Physiology*.

**Goldsmith GR, Muñoz-Villers LE, Holwerda F, McDonnell JJ, Asbjornsen H, Dawson TE**. **2012**. Stable isotopes reveal linkages among ecohydrological processes in a seasonally dry tropical montane cloud forest. *Ecohydrology* **5**: 779–790.

**Hervé-Fernández P, Oyarzún CE, Woelfl S**. **2016**. Throughfall enrichment and stream nutrient chemistry in small headwater catchments with different land cover in Southern Chile. *Hydrological Processes*: n/a-n/a.

**Huang C, Domec J, Ward EJ, Duman T, Manoli G, Parolari AJ, Katul GG**. **2017**. The effect of plant water storage on water fluxes within the coupled soil–plant system. *New Phytologist* **213**: 1093–1106.

**Meinzer FC, Andrade JL, Goldstein G, Holbrook NM, Cavelier J, Wright SJ**. **1999**. Partitioning of soil water among canopy trees in a seasonally dry tropical forest. *Oecologia* **121**: 293–301.

**Meißner M, Köhler M, Schwendenmann L, Hölscher D**. **2012**. Partitioning of soil water among canopy trees during a soil desiccation period in a temperate mixed forest. *Biogeosciences* **9**: 3465–3474.

**Meng W, Xia Y, Chen Y, Pu X**. **2018**. Measuring the mutual diffusion coefficient of heavy water in normal water using a double liquid-core cylindrical lens. *Scientific reports* **8**: 1–7.

**Muñoz-Villers LE, Geris J, Alvarado-Barrientos S, Holwerda F, Dawson TE**. **2019**. Coffee and shade trees show complementary use of soil water in a traditional agroforestry ecosystem. *Hydrology and Earth System Sciences Discussion*.

**Orlowski N, Frede HG, Brüggemann N, Breuer L**. **2013**. Validation and application of a cryogenic vacuum extraction system for soil and plant water extraction for isotope analysis. *J. Sens. Sens. Syst* **2**: 179–193.

**Phillips DL, Gregg JW**. **2003**. Source partitioning using stable isotopes: coping with too many sources. *Oecologia* **136**: 261–269.

**Rothfuss Y, Javaux M**. **2017**. Reviews and syntheses: Isotopic approaches to quantify root water uptake: a review and comparison of methods. *Biogeosciences* **14**: 2199.

**Stahl C, Herault B, Rossi V, Burban B, Brechet C, Bonal D**. **2013**. Depth of soil water uptake by tropical rainforest trees during dry periods: does tree dimension matter? *Oecologia* **173**: 1191–1201.

**Wang P, Song X, Han D, Zhang Y, Liu X**. **2010**. A study of root water uptake of crops indicated by hydrogen and oxygen stable isotopes: A case in Shanxi Province, China. *Agricultural Water Management* **97**: 475–482.

**White JWC, Cook ER, Lawrence JR, Broecker WS**. **1985**. The D/H ratios of sap in trees - implications for water sources and tree-ring D/H ratios. *Geochimica et Cosmochimica Acta* **49**: 237–246.

**Zhao L, Wang L, Liu X, Xiao H, Ruan Y, Zhou M**. **2014**. The patterns and implications of diurnal variations in the d-excess of plant water, shallow soil water and air moisture.

---

## Author Comment (AC2) · 6 Apr 2020

To Whom It May Concern,

Please find our responses to all comments in the pdf document "bg-2019-512-supplement.pdf". We have compiled all responses into a single document, for clarity.

Sincere Regards,

De Deurwaerder and colleagues

Please also note the supplement to this comment:

https://www.biogeosciences-discuss.net/bg-2019-512/bg-2019-512-AC2-supplement.pdf

---

## Author Response (AR1)

Dear Editor,

We hereby submit our final response and proposal for improvements to the manuscript "Causes and consequences of pronounced variation in the isotope composition of plant xylem water" to be considered for publication as a research article in *Biogeosciences*.

First, we would like to thank both referees for their thorough assessment of our manuscript, as their suggestions have greatly improved its quality. We are pleased that both reviewers acknowledge the importance of our study (e.g. reviewer #1: *"highlighting the temporal, as well as spatial (longitudinal) dynamics of δxyl, is of evident interest"*, reviewer #2: "*I think it is good that the authors bring forward the point that xylem water may sometimes exhibit rather dynamic variations in its isotope composition*"). We were also happy to notice that the essence of our work (i.e. investigating the diurnal variability in isotopic composition along woody stems) and its merits are not questioned. We noticed that most of the (major) criticisms arise from problems in the presentation, formulation, and overstatement of our work. These oversights are addressed in the new version of the manuscript. In addition, we provide a more balanced presentation of the limitations of our study and of the model we developed as a plausible explanation of the observed variability. In particular,

I.  both reviewers indicated that our empirical dataset is not ideal (i) for model validation, and (ii) to support some stronger statements regarding the implications of our findings. We acknowledge these points and have addressed them as follows:

1. **We restructured the manuscript giving more emphasis on the strong points of our empirical datasets,** which are unique in the field, and show pronounced variability (temporal and longitudinal) in the isotopic composition of xylem water ($\delta^2H_X$). Moreover, **a new dataset obtained in Germany** (Magh *et al.*, 2020) extends the original datasets of French Guiana and China. This new dataset describes pronounced intra-individual $\delta^2H_X$ variance observed during high temporal resolution monitoring of $\delta^2H_X$ in Silver Fir and Beech.

2. **We now emphasize clearly that the model analysis is intended as a theoretical exploration** to build hypotheses and to understand when to expect large variance in $\delta^2H_X$ (L35). We clearly indicate that the coupling between the data and the model is only qualitative at this stage (L284:286; L361:363);

3. **We toned down the manuscript title and softened some potentially inflammatory statements** (see details in our response to the reviewer comments), especially regarding the limitations of the isotope method for determining RWU;

4. **We expand the existing discussion section, by elaborating** our existing section on alternative hypotheses that could contribute to the observed variability (L558:564; L575:576; L604:616);

5. **We shed a more positive light on the implications** of diurnal variability in $\delta^2H_X$ as this can lead to novel information and opportunities in water acquisition and plant performance studies (L618:622).

II.  concerns were raised about the realism of the presented model. Our model considers basic physical and physiological processes, and we agree that it is inevitably - as every other model - a simplification of reality. We stress, however, that the suggested implementations, while improving realism, will not change the conclusion of the paper: pronounced changes in $\delta^2H_X$ can be expected along the stem of woody plants (or similarly at one vertical position over time). We highlighted this by implementing some suggestions, and providing further details in our discussion where alternative hypotheses seemed appropriate – we note that some of the reviewers' concerns were already included in the discussion section and are now elaborated on. In particular, changes include:

1. Reviewer 2 suggested including a molecular diffusion term in the transport equation as it can lead to the homogenization of $\delta^2H_X$ within the plant. We explored this possibility using analytical solutions of the advection-diffusion equation. **These simulations show that the impact of diffusion is negligible when sap flux densities are high** (see figure 1, below)**, as is the case for our experimental examples.** Diffusion will - very slowly, i.e. over multiple days - reduce the absolute range of variability in $\delta^2H_X$ by smearing the isotopic composition (See figure: ±5cm in 24h), but it also leads to broader $\delta^2H_X$-baseline drops. This implies that while the absolute range of variability might slightly decrease over time (or with tree height). The probability of sampling in the $\delta^2H_X$-baseline drop will in fact increase, **strengthening the importance of our main message. However, we also indicate that diffusion could become more important at very low sap flux densities** as this implies an accumulated effect over multiple days. This results in a time-lag between $\delta^2H_X$ and isotopic composition of soil water, presenting another complication for RWU assessment. **Diffusion is now extensively discussed** in the manuscript (L466:482; L603:615; and supplementary methods B).

2. The impact validation of molecular diffusion did not show strong impacts on $\delta^2H_X$ dynamics along the length of the stem. However, this suggestion of reviewer 2 instigated **a more in-depth assessment if other processes besides molecular diffusivity might contribute to isotope transport through the plant** (e.g. variable flow velocities within vessels and among vessels of the xylem network). We extend our study with a new analysis comparing the xylem transport in the model against a recent $^2$H enrichment study of Marshall *et al.* (2020) (L474:483; L604:616; Fig 6; Supplementary methods B). Marschall *et al.* (2020) applied a novel *in situ* borehole equilibration technique for continuous monitoring $\delta^2H_X$ dynamics in a *Pinus pinea* individual. This new analysis highlights the need for an improved understanding of $\delta^2H_X$ uptake and transport along trees. It further emphasizes the current lack in understanding various important processes, besides diurnal fluctuations in RWU-activity, that might alter $\delta^2H_X$. These processes are currently ignored in the usual approach of using stable water isotopes for RWU assessment. **Therefore, we further highlight and discuss the need for more intra-individual physiological and hydrological understanding via targeted studies,** for the betterment of the current implementation of the stable water isotopic technique for RWU as well as the presented model (L604:616; L664:668).

3. There is potential for water exchange between storage tissues and xylem water, we discuss this implication in the discussion (L580:603), but decided not to include such a process in the present model version as (i) it depends on the assumption that storage water is representative of soil water uptake by the roots, (ii) we do not have information on storage water isotopic signature and dynamics, and (iii) we are not aware of any existing dataset that could parameterize this model process. Moreover, we highlight that no homogenization is visible in the presented empirical data despite the likely exchange from storage cells (Fig 3c and supplementary methods B). Furthermore, in the discussion, we particularly highlight that, if storage water is not representative of soil water uptake by the roots, then exchange with storage water **likely exacerbates potential bias in the isotope tracing technique, strengthening the main issues raised in our paper.**

4. Similarly, including variations of soil water isotopic composition and water potential over time may improve the model realism and affect the absolute range and the dynamics of xylem water isotopic composition **but would not lead to homogenization.** (L565:579)

Finally, we like to highlight that Kathrin Kuehnhammer, Ruth-Kristina Magh and John D. Marshall have been added to the author list, as they provided (a) the empirical data in Germany and (b) the dataset used in the new validation of $\delta^2 H_X$ transport dynamics through the plant at very low sap flow velocities, and (c) helped with the corresponding analysis and revision.

Please find our detailed responses to reviewer comments below (responses to reviewers in bold).

We hope that the implemented adjustments, inspired by the reviewers' suggestion, improved the manuscript allowing publication in *Biogeosciences*.

The authors

[Figure]

**Fig 1:** Analytical solutions of advection-diffusion equation on a semi-infinite 1-D domain with 12 ‰ step-change in isotope signature for different values of flow velocity and diffusivity. The plots show the impact of diffusion on the isotopic composition of xylem water. Colored lines show the solution at different time intervals: 0, 12, 24, 48, and 96 hr. Note that the values of diffusivity are much higher than these reported for heavy water (e.g. D=0.1 $cm^2$ $h^{-1}$; Meng et al., 2018)

_**Anonymous Referee #1**_

De Deurwaerder and colleagues present a composite work where they (i) run a model simulating diurnal variations and vertical heterogeneity in xylem water isotopic composition ($\delta$xyl) and perform a multivariate sensitivity analysis. They also (ii) present results of sampling campaigns where $\delta$xyl temporal and spatial variations were observed in twelve tree and liana species. The authors explain these variations and thus the departure of the generally accepted hypothesis of homogeneous $\delta$xyl on account of their model output. Finally, they warn the isotopic community against the "danger" in using water stable isotopes as tracers for RWU analysis.

The manuscript is well written, figures and tables are of good quality and appropriate referencing supports the text. Finally, the manuscript content falls within the scope of BG.
**We thank the reviewer for his/her appreciation of the quality of our work, and the detailed assessment of the study and constructive feedback on the manuscript. We feel that the paper improved considerably thanks to his/her suggestions.**

My general comments are listed below:

1- I note that the authors do not confront their model results to collected data, nor thoroughly test their model hypotheses on independent data. I do not see a particular problem, but it should be mentioned clearly that aforementioned items (i) and (ii) are only "softly" coupled in the study.
**The reviewer makes a fair point. As our field data is unique but limited, they do not allow direct validation. Our model presents a theoretical exploration of one of the potential causes of the observed high variability in isotopic composition in xylem water ($\delta^2H_X$) along the stem of a woody plant. For illustrative and interpretative purposes, our model explores ideal, simplified environmental conditions. The empirical data present a much more complex situation, which we were unable to characterize fully due to financial and logistical restrictions. The new version of the manuscript mentions that our study presents only a qualitative coupling between model and data, as suggested by the reviewer** (L283:285; L360:362);

2- highlighting the temporal, as well as spatial (longitudinal) dynamics of $\delta$xyl, is of evident interest. However, the prevalence of such dynamics may not put in "danger" – as the authors say – the determination of fractional root water uptake for other non-wooden species. The abstract should be rewritten accordingly. The isotopic community should be on the "safe" side if researchers extract water from a plant tissue for which it has been proven that its stable isotopic composition reflects that of RWU. Of course, this should be investigated for each investigated plant species, preferably under controlled conditions (see for example Barnard et al., 2006).
**The abstract of the paper is rewritten accordingly (see L33:38). We agree that a distinction between woody and non-woody plants should be considered as described by Barnard _et al._, (2006), as highlighted by the reviewer. Our model targets woody species (i.e. > 70% of all isotopic studies, Rothfuss and Javaux (2017)), and it is therefore not appropriate to speculate about non-woody species. We re-formulate our statements in the new version of the manuscript (i.e. expressions such as "to put in danger" is dropped and replaced by more informative and appropriate formulations).**

3- The authors provide no information about the soil compartment; what about the soil water isotopic composition profile temporal and spatial variabilities? Are the isotopic differences in xylem water reflected by the span of isotopic composition values in soil water? This would offer the possibility to rule out possible evaporation effects mostly during sampling and transport (which is not listed as other reasons for the observed diurnal variations of δxyl). If soil water isotopic information is not available, it should be stated as a limitation of the study;

**The empirical data collection indeed has the limitation of the absence of adequate soil characterization during field setup (i.e., soil water potential and isotopic composition of soil water), which is now clearly stated in the new version of the manuscript (supplementary methods A; Fig S1).**

**In addition, evaporation effects during sampling and transport can never be excluded in field studies but have been minimized by the applied protocol, as detailed in the manuscript. However, we expect evaporation effects to be low because of (a) the imposed strict protocol. Specifically, fast sampling with cautious care to avoid heating the extraction instruments was followed by fast capping of the sample vials (sealing caps with rubber and glass vials having a minimum of two full closing coils), immediate cooling of the vials in the field, and freezing of vials upon return in the lab; and, (b) cross-validation of the obtained $\delta^2H_X$ with potential source water isotopic composition performed during the dry season at Laussat do suggest that our $\delta^2H_X$ lie within the natural span of the soil and precipitation water sources. These observations, although not of adequate quality as most samples did fail the 98% recovery validation, is now provided as a supplementary figure (Fig S1, and supplementary method A). In addition, we highlight that the pronounced intra-individual $\delta^2H_X$ variance is now observed in 3 independent datasets collected by 3 independent research groups.**

4- I found on several occasions that the authors did not fully understand basic principles driving isotopic fractionation (see my specific comments);

**We regret that we left the impression of a less than full understanding of the basic principles behind isotopic fractionation. We removed all instances of careless representation and wording in the new version of the manuscript (see our answers to the specific comments for more details).**

5- In general, I do not think that such field experiments, where a significant number of environmental driving factors are unknown, should be used to question the entire isotopic research methodology. I urge the authors to discuss this point as well and measure their words.

**We agree with the reviewer and have toned down the message to be more in line with the uncertainties in the data. We now realize that our original tone could have been perceived as questioning the entire isotopic research methodology – which was not our intention. Therefore, in the revised manuscript, we use more appropriate statements, as well as included some positive aspects of diurnal variability in $\delta^2H_X$ (L618:622). These could present new opportunities in water acquisition and plant performance studies. However, we remain convinced that our findings indeed show the need for caution when applying isotopic research methodology in multiple situations and configurations, as large variability of stem isotopic composition are expected and could plausibly lead to significant bias in average RWU depth determination. Our main objective remains to (a) build increased awareness of the potential of diurnal variability to bias future isotope**

**endeavors, and (b) to advocate for more targeted intra-individual physiological and hydraulic studies to further our understanding in how isotopes are taken up and transported throughout the tree and how these processes might impact the current RWU assessment approach using stable water isotopes.**
* * *
The authors will also find a list of specific questions/remarks/corrections/issues:

L24. What does "i-$H_2O$-xyl" refer to? To "plant xylem water uniform isotope composition" or "plant xylem water isotope composition"? In either case, "$\delta$" is to be preferred over "i-$H_2O$-"
**This is a good suggestion. The "$\delta$"-notation suggested by the reviewer is adopted, as i-$H_2O$-xyl originally referred to the plant xylem water isotope composition.**

L32-33. "field data show pronounced i-H2O-xyl variation during the day or along stem length ranging up to 25.2‰ in $\delta 2H$ and 6.8‰ in $\delta 18O$" does not read well. I propose something like: "the hydrogen (oxygen) isotope composition of plant xylem water showed strong temporal (i.e., daily) and spatial (i.e., along the stem) variation ranging up to 25.2 (6.8) ‰´"
**This sentence is adjusted accordingly (L39:42).**

L36. Please rephrase: "danger" is not the proper word.
**This is adjusted accordingly (L46:49).**

L46-47. There is no such thing as the depth of root water uptake in the case of several soil water sources. Only in the context of direct inference is this true. But the authors do not refer to the later (and outdated) technique.
**We agree that the terminology: 'average root water uptake depth' is more appropriate. This is implemented accordingly throughout the manuscript.**

L49. This is not true: the isotopic technique is of course destructive (you have to take a soil core), very labor-intensive (e.g. extraction of soil and plant xylem water).
**This statement presents a comparison with root excavation endeavors, which are extremely time-consuming, laborious, and destructive. We adjusted the statement to read that in comparison to root excavation, isotope techniques are far less destructive and time-consuming, and are hence definitely preferred when studying multiple individuals at once. (L63:67)**

L50 (also L47). You should mention that it is fractional RWU and not absolute RWU you are talking about. You cannot solve for water mass balance with the isotope technique, which constitutes its greatest limitation when compared to other techniques.
**This is adjusted accordingly (L66).**

L52. How would you determine fractional root water uptake at the ecosystem level?
**This statement, which is taken from Dawson *et al.*, (2002), can - for instance - embody $\delta_{xyl}$ analysis performed on the dominant tree species of a forest stand (i.e. forest stands on Mount Kilimanjaro - Bodé *et al.*, 2019), or on classified groups of plant individuals (juvenile versus adult – Stahl *et al.*, 2013; liana versus tree – De Deurwaerder *et al.*, 2018). These measurements inform on the average depth of water acquisition (i.e., "strategy") of**

the species/group, which can then be extrapolated to estimate the expected dynamics/strategies at the ecosystem scale.

L56. This should be "δxyl". Why the "i" instead of "δ" here? Also, why the "H2O"? (is there another molecule investigated here?)
**As indicated in previous comment, we now adopted the "$\delta_{xyl}$"- notation suggested by the reviewer.**

L58-60. Peclet effect is measurable in the xylem vessels upstream of the evaporative sites. This assumption is not systematically made. Instead, authors investigate the prevalence of isotopic fractionation depending on the plant tissues they sample, e.g. in Barnard et al. (2006). Please revise.
**This remark of the reviewer is addressed by (i) emphasizing that this study targets woody plants, as non-woody plant are indeed subjected to "stem" fractionation processes (Barnard et al, 2006, a reference which will be included), and notify that (ii) the Péclet effect might be observed in branches upstream of evaporative surfaces (L29-31 in Supplementary method A). The later presents a rather local phenomenon and should not, or very limitedly, impact stem samples at distance from the evaporative surface, as performed in this study.**

L67-69. There can only be kinetic fractionation playing a role during the transport of water through the root membranes since there is no liquid-vapor phase change that would involve equilibrium fractionation. Please revise.
**The reviewer is correct, and this is revised accordingly.**

L69-72. Not only kinetic fractionation is a result of the difference in mass of the water isotopologues, but fractionation in general (e.g., equilibrium and kinetic fractionations).
**This is correct and was unfortunately dropped out during the editing of the manuscript. We revise our definition accordingly emphasizing that this entails the transport of water through a root membrane.**

L94-95. Why would you make the assumption that δxyl is constant over time (over which period of time anyway)? At this point of the MS, it is not clear. Actually, no one makes this assumption in the field, rather they sample from e.g. the base stem among individuals at e.g. a sub-hourly temporal resolution and sub-daily temporal extent.
**We agree with the reviewer that we should be more precise in our formulation of the hypothesis and the time-frequency considered (sub-daily and even sub-hourly). This is now addressed in the new version of the manuscript (L99:100, but see Fig 1). We also note that there may be a misunderstanding here regarding the assumption made in the field.**

**It is indeed correct that a few high-frequency measurements of $\delta_{xyl}$ exist. However, it should be noted that these are (a) rather rare at the moment; and (b) predominantly target sampling of the leaves. Sampling of leaves, however, is less relevant to the 'isotopic tracing technique for RWU assessment' as multiple other processes impact the isotopic composition of leaf water (i.e. the aforementioned Péclet effect). In this study, we do not address leaf water monitoring because of the decoupling between source water and measured signature. To date, most studies where the isotopic composition of xylem is used for RWU assessment have - at best - a daily, but more often a monthly or seasonal temporal sequence. Moreover, many of the studies (including ours) consider only one-time sampling (including ours, see e.g. De Deurwaerder et al., 2018). These studies do assume**

**a constant $\delta_{xyl}$ over time. Hence, sub-hourly/daily $\delta_{xyl}$ variances are generally not accounted for in studies on lignified stem sampling.**

**Finally, we acknowledge that coring close to the base of an individual stem is generally applied in non-woody, herbaceous plants as recommended by Barnard et al., 2006. However, to the best of our knowledge, this is not standard practice in woody plants. We acknowledge that it might be more general than we know, as implied by the reviewer, but this is not reflected in the existing literature where the height of coring is rarely provided, and when so, coring is generally performed where stem diameter is measured (i.e. 1.3m in the metric system, and at 4.5 feet in the imperial system) (e.g. White *et al.*, 1985; Meinzer *et al.*, 1999; Goldsmith *et al.*, 2012; Hervé-Fernández *et al.*, 2016; De Deurwaerder *et al.*, 2018; Muñoz-Villers *et al.*, 2019) (L302:304)**

100-101. What do you mean by "diurnal changes in the soil-plant-atmosphere continuum"? Which changes?

**This statement indeed needs further clarification, which is pursued in the new version of the manuscript. In short, with "diurnal changes in the soil-plant-atmosphere continuum" we imply: changes in water potential differences between leaf and soil along the day (L102:103, but also see Fig 1). These gradients will determine the vertical distribution of root water uptake.**

L113. What exact "water potential gradients" do you refer to?

**Here, we refer to the water potential gradient between soil and the evaporative surfaces (leaves) of the plant. This is added to the manuscript (L102:103, see Fig 1).**

L114-116. Why would you need to use a mixing model, especially since you did not sample soil water and determine its isotopic composition? You may as well simulate a sinusoidal pattern for the $\delta_{xyl}$. Please elaborate/explain.

**We did measure soil water ourselves in this study, however, the obtained recovery rates of the extraction mostly did not reach the requested benchmark of 98% recovery (see Fig S1). This data is therefore not considered for further analysis (L53 in supplementary method A). The soil water isotopic composition used in our theoretical exploration target a model representation that is practically implementable and repeatable. Here, the approach of Phillips and Gregg (2003) presents a widely used and implementable approach to mathematically represent fractional water uptake in the soil. The apparent sinusoidal pattern of the xylem water isotopic composition observed by the reviewer results from the diurnal fluctuations in leaf water potentials and corresponding changes in the distribution of the root water uptake in the various soil layers. Specifically, here, the pattern in leaf water potential is imposed by a bell-shaped sap flow curve obtained from Huang *et al.* (2017). Hence, this apparent sinusoidal pattern naturally emerges from the source mixing model approach and was not hardcoded as such in the model.**

L117-119. You should write the isotopic equations with "$\delta$" instead of "$\delta 2H$" as the model does not focus on 1H2HO, to the contrary of what the authors say. For the model to focus on 1H2HO, it would mean that 1H2HO and 1H218O would follow different physical processes, which is not the case (both isotopologues undergo mass dependent fractionations, i.e. $\varepsilon eq(2H)/\varepsilon eq(18O) \approx 8$ and $\varepsilon k(2H)/\varepsilon k(18O) \approx 0.88$). Also write 2H instead of Deuterium and do it consistently throughout the MS. The latter is just an element's isotope and does not deserve (anymore) its own letter (see IAEA tech reports guidelines).

**We adopted the notation suggested by the reviewer throughout the manuscript, and we rephrased the statement as we indeed focus on the water isotope element instead of the water isotopologue as was inaccurately implied in the original manuscript. [L 198-199]**

L130-131. This assumption is only reasonable when soil water redistribution no longer occurs, e.g., this does not stand shortly after a rain event.
**For the sake of simplicity, we present a model that assumes rain-free periods and prevents soil redistribution, as indicated by our statement L130-132 (original manuscript lines): *"… a reasonable assumption if the isotopic measurements are conducted during rain-free periods, …"*. We acknowledge that this assumption was not presented clearly, which we addressed in the new version of the manuscript. (L205:207)**

L125ff. Report the dimension of each variable and parameter throughout the MS.
**For the readability of the text, we prefer the use of a dedicated table listing the variable/parameter dimensions altogether, as done in Huang *et al.* (2017) (here Table 1, as indicated in the text L197).**

L142. From Eq. (3), I understand that the water "potentials" are in fact "hydraulic heads". This should be clarified.
**This is now clarified in the new version of the manuscript by indication the generic potential term equals hydraulic head or soil matric potential (L219).**

L143-144. Add here that $k_i$ and $\Psi_{S,i,t}$ are also specific to the ith soil layer.
**This is adjusted accordingly.**

L193-196. I am missing background information to understand what the "30 days sequence" of the "model runs of Huang et al. (2017)" refers to. Please elaborate.
**We agree that our statement is unclear for readers that are not familiar with the paper of Huang *et al.* (2017) in which a 30-day drought simulation study of loblolly pine was conducted. An average day within this representation was selected based on both representativeness and data availability. We now elaborate on this topic in more detail in the new version of the manuscript (L285:290).**

L201-203. Why would you need external data (Meissner et al. 2012) and not simply do your model exploration on basis of a synthetic experiment?
**The reviewer is correct, as applying a complete synthetic experiment is indeed possible. We chose to use the Meißner *et al.*, (2012) data as this presents a realistic dataset (in terms of range and variation in both soil water potential and soil water isotopic composition) obtained during field studies, and therefore find it relatable for both interpretation as well as providing insights for model requirements guiding field setups.**

L208-209. I did not hear of such standard practice and I doubt there is. Could you add a reference for this?
**Here, we assume researchers followed standard procedure in using an increment borer in forest inventory, i.e., coring where stem diameter is typically measured (i.e. breast height: at 1.3m according to the metric system, in at 4.5 feet according to the imperial system). This method has been applied multiple times (e.g. see White *et al.*, 1985; Meinzer *et al.*, 1999; Goldsmith *et al.*, 2012; Hervé-Fernández *et al.*, 2016; De Deurwaerder *et al.*, 2018; Muñoz-Villers *et al.*, 2019), but we agree that it does not need to be presented as a standard practice, as several isotope tracing studies applying an increment borer to collect xylem**

cores do not specify the height of coring. We also acknowledge that (a) many studies sample branches (ignoring the effect of evaporative enrichment from the leaves to upstream plant organs), and (b) that our assumption that researchers follow the standard increment borer approach could be incorrect. We rephrased this statement, providing the here presented references in support of the followed approach (L302:304).

L223-225. Split the sentence and add detail. It is hard to understand. Also following the Rayleigh distillation model, the error should always be negative in the case of incomplete water recovery, which does not match to your normal distribution of error in the null model.
**The indicated sentence is split up and clarified. We thank the reviewer for this excellent suggestion. It is true that the expected error should be negative, which we now have implemented in the model structure by using a skew-normal distribution instead of a normal one (L327:332).**

L228-229. How so? And why would it be relevant to take into account the analyzer systematic error at this point of your model testing?
**Analyzers always have an embedded error which is generally very small. But, if known, the user can opt to implement these in SWIFT model. In this study, we consider these errors negligible and have indeed ignored them as it has little relevance in the model testing at this point, as indicated by the reviewer. For sake of clarity, this sentence is now removed.**

L235-244. Are you talking about RWU depth of rooting depths here? How do you define the latter term? Why would you use the direct inference model (which is a very simplistic view on RWU, i.e., one single root sampling from on single layer at a time) if you use a multi-source mixing model (Phillips and Gregg, 2003), which allows the plant to sample simultaneously from different layers? Please explain this apparent contradiction. Overall this section is quite difficult to read and I ask that the authors simplify it.
**Here we are talking about the average depth of RWU (i.e. a weighted mean of the depths of root water uptake, with the root flows at the different depths as weights (now included L335:336), and hence, this section/title will be changed accordingly for clarification (L334). The paragraph was further simplified by textual alteration (L335-345). We applied both methods for completion of the presented study, as combined, they embody 96% of all applied methods (Rothfuss and Javaux, 2017). While direct inference might be considered as very simplistic, to date, it remains the most applied technique in the literature (46% according to Rothfuss and Javaux, 2017). In short, the direct inference approach compares hydrogen and oxygen isotopes between the soil water profile and the xylem water of the stem. The depth of soil water having similar isotope values to the stem water indicate the main depth of soil water sources used by the plant (e.g. see Wang *et al.*, (2010)). This approach does not exclude that the plant can take up water from multiple soil layers but just assumes that the signature found in the xylem reflects the dominant signature of bulk water uptake. It is therefore unclear to us what the reviewer exactly means with "the apparent contradiction".**

L258. Is there a specific reason why you did not use Van Genuchten's soil retention curve?
**There is no specific reason not to use the van Genuchten's soil water retention curve. As we do not know the soil hydraulic properties at the site (soil retention and conductivity curves), we do not have any reason to prefer Clapp and Hornberger (1978) closed-form equation instead of the Mualem-van Genuchten model. We will implement more soil hydraulic models (including Mualem-van Genuchten) in future model versions.**

L309. Delete "kinetic". It is not even sure that you would have fractionation at all, considering that you may boil (==fractionation free process) the water here rather than evaporate it.
**This is adjusted accordingly.**

L321. Fresh weight does not take into account possible loss of water during transport/ storage. You should have weighted the samples prior extraction again.
**We agree with the reviewer that measuring before extraction itself could provide extra information on whether or not water was lost during transport/storage of the samples. We did not do this, and can therefore not provide such insights to the reader.**

**However, as we used glass vials with sealing caps (including sealing rubber, and at least two complete loops of closing coil), water losses during transport and storing should be negligible/absent. Besides, it should be noted that measuring samples after storage, i.e. before extraction, might itself impose sample contamination and inaccurate assessment of the percentage of water extracted by the cryogenic water extraction method. Specifically, frozen vials will attract frost and condense water onto the vial exterior, which can substantially impact the weight of the vial itself. This then should be accounted for, for instance by warming the samples to room temperature before weighing, a practice which arguably is also not recommended.**

L331-336. Since you are measuring with a Picarro, which does not give ratio (but performs already the delta conversion), you need to say that you "corrected the Picarro raw delta readings into calibrated delta values thanks to the values of the aforementioned 'internal laboratory References' expressed on the international V-SMOW scale". No need to display the equation (12) but you may detail these "internal laboratory References" (e.g., value).
**The suggestion of the reviewer is implemented in the new version of the manuscript. (L58:62 in supplementary method A)**

L336-334. Still at this point, I do not know what the difference is between i-H2O-xyl and δxyl: : :If there is none, please use the latter term. In addition, use another letter than ε for the normalized "i-H2O-xyl": it usually stands for isotopic fractionation, defined as the deviation of the fractionation factor to unity. It seems even odd that you would consider such a letter…
**As indicated above, we replaced the symbol of i-H₂O-xyl by δ$_{xyl}$ as suggested by the reviewer. We agree with the reviewer that our choice to use ε here was unfortunate. While 'ε' is commonly used in statistics to indicate bias in the sample set, we now see that this indeed can result in confusion for the isotope community. Hence, another Greek letter (i.e. 'β') is used in the revised manuscript.**

L369. I still do not understand what is the concept of RWU depth if you consider the multi-source mixing model approach.
**As indicated above, we have clarified this definition. Throughout this study, we consider 'average depth of root water uptake' (i.e. a weighted mean of the depth of root water uptake, with the root flows at the different depths as weights), as is now adjusted in the new version of the manuscript (L335:336).**

Fig. 2. Panel (a): how do you come up with a night δxyl at 1.3 m above −60‰ Also, I don't see why panels (a) and (d) look so different for day 1, since if I understand correctly, the cumulative SF is a function of time (if sap flow remains constant).

**The patterns in δ²Hₓ results of both the isotopic composition of the water taken up by the roots at any time – and – the volume displacement of water moving as a slab along the tree stem. At each time step considered, a specific volume of water and isotopic composition is extracted by the plant. This presents δ²Hₓ at stem base which is not limited by the volume of the tree yet. However, this quantity of water is subsequently pushed, as a slab, upwards in the limited volume of the stem, i.e. our model presents a piston-flow approach. At this point, the quantity of water taken up by the plant also impacts the observed δ²Hₓ pattern.**

**Specifically, the water movement within the tree can be visualized by 'a stack of disks' of water each having a time-specific δ²Hₓ, where stack-height is defined by the quantity of water taken up and the stack area corresponds to the lumen area of the tree. Step by step, new disks are introduced at the bottom, pushing previous disks upwards, i.e. water moves as a slab through the stem. When root water uptake activity stops, i.e. sap flow is zero, the stacks remain at their respective position. When measuring at 1.3m height, the entire volume of water taken up in the late afternoon, with values of -66‰ is simply too small to reach the measurement height. For this reason, δ²Hₓ at 1.3m represents the water isotopic composition of water taken up earlier during the day (i.e. around midday), which has a more enriched isotopic composition.**

L371. "isotopic composition of soil water is dominated by depleted deuterium". Please correct phrasing: soil water can be depleted in 2H in comparison to another water volume, but there is no such thing as "depleted 2H".
**This is corrected accordingly.**

L373. An isotopic composition, which is a number, cannot be "enriched". Please correct.
**This is corrected accordingly.**

L375. "depleted deep soil water"
**This is corrected accordingly.**

L384. ": : :RWU originating from deeper, more depleted soil layers". Please correct: water from a given soil layer might be depleted, not the soil layer in itself.
**This is corrected accordingly.**

L399-400. This belongs to the discussion section.
**This sentence is moved to the discussion section.**

L407-418. Nowadays no study is published where RWU depth is investigated with the direct inference method. Analyses are performed with Bayesian mixing models. So I wonder if this section, although interesting theoretically, would benefit practically to the community.
**Indeed, Bayesian mixing approaches become more commonly used in current literature. However, we argue that the potential issues in RWU assessment unraveled in our research apply to all existing literature, of which the direct inference method still embodies the majority of studies (see Rothfuss and Javaux, 2017). For this reason, we are convinced that this section can be relevant when a critical assessment of former studies is pursued.**

L446 and Fig. 4. See my previous comment on the use of "ε". The caption of a figure should not point to another figure or table. Write here the name of the species (no need to write them in the figures though).

**This suggestion is implemented in the figure, and the notation "ε" is replaced by "β".**

L452. Add in the text that growth forms refer to lianas and trees.
**This is adjusted accordingly**

L455-457. This belongs to the discussion section. Also the link between "easily accessible and abundant groundwater reservoir" and the fact that the diurnal intra-individual variance is minimized is not clear. I suggest moving to the discussion and elaborating on this.
**This sentence is no longer included in the paper, as it distracted from the main storyline of the study.**

L471-472 and Table 2. How many individuals (which you could consider as replicates) of each species were sampled during the experiment? Discuss the implication of having n=1 with respect to $\delta_{xyl}$ variance.
**It is true that only one replicate per species was obtained for this study. That was because we did not target the intra- or interspecific variances in $\delta_{xyl}$ in our experimental protocol but instead we investigated the intra-individual $\delta_{xyl}$ variability and the theoretical exploration of a likely cause of this phenomenon.**

L486-492. The authors say that the intrinsic problem of the "isotopic tracing method" is that there is a soil water isotopic gradient in case there is evaporation and under heterogeneous soil water potential gradient? I don't understand this at all (!) The isotopic methodology for studying plant RWU relies on heterogeneous isotopic gradients in soil water. This is a solution, not a problem here…
**We acknowledge that the text was not clearly formulated in support of the argument envisioned. What we wanted to convey is that the soil water conditions required to perform the 'isotopic tracing method', also facilitate a large variance in $\delta^2H_X$, which could have important consequences for the RWU assessment. An altered, non-ambiguous discussion is now presented in the new version of the manuscript. (L518:521, see Fig 1)**

L493-506. I disagree. There is a clear problem in determining fractional RWU profiles on basis of measurements of the transpiration isotopic composition, which is highly temporally dynamic and spatially heterogeneously distributed; many observation of leaf water confirm the non-reaching of isotopic steady state. In addition, how would a "change of cloud cover degree" have an "instantaneous" influence on $\delta xyl$? This contradicts the results of your synthetic experiments, where depending on sap flow rate, there is a marked isotopic memory effect of the antecedent water moving upward it the xylem vessel.
**Cloud cover will result in reduced water environmental demand and thus impact the sap flow velocity (and thus the water and isotope dynamics in the stem). Hence, the cloud cover of a tree will reflect in distinct patterns of RWU uptake dynamics and the bulk isotopic composition of water extracted from different soil layers. The statement does not contradict our model findings but we acknowledge that the presentation could have been more clear. What we wanted to say is that the intra-individual variability of $\delta^2H_X$, according to our model simulations, reflects indeed the past changes of root water uptake dynamics (including due to dynamic changes of environmental demands). This is now clarified in the text (L531:334)**

L516-523. The model provides an explanation, sure, but does not validate your hypotheses from the confrontation with experimental data. This is missing from your study and should be mentioned.

**We fully agree with the reviewer and acknowledge that we have overstated our findings. This version of the manuscript more clearly describes the limitations of our study (L284:286; L361:363);. This is also clarified by the change in paper structure: we now present our model simulations as a potential explanation of the isotopic composition variability observed in the field.**

L534-546. My understanding from the literature is that hydraulic redistribution is intermittent and localized, thus does not affect that much the bulk soil water isotopic composition, rather it affects the direct environment of the roots.
**Correct, we agree with the reviewer that hydraulic redistribution will predominantly impact the rhizosphere of the plant, rather than the isotopic composition of the water in soil layers. This paragraph is rewritten as such (L566:572). The main message within this paragraph, suggesting that hydraulic lift will reduce the $\delta_{xyl}$ variance, remains valid as the variance of the isotopic composition of water accessed by the plant can be reduced.**

L578-587. Not to forget we need to monitor soil water isotopic composition to verify if $\delta_{xyl}$ spreads within the range of isotopic values observed in the soil profile.
**Correct, and we add this suggestion to the new version of the manuscript (L623:626).**

The manuscript by De Deurwaerder and colleagues challenges the idea that, in absence of precipitation or other rapid changes in climate, the water isotope composition of plant xylem should stay fairly constant over diurnal time scales or along stem height. Their analysis is based mostly on a model (!) of root water uptake and isotopic transport within the roots, up to the stem base. Their model considers that (1) the isotope composition of stem water at the base of a tree ($\delta^2 Hx_{(0; t)}$) is the average isotope composition of soil water over the root zone, weighted by the fractional root water uptake rates at each depth (Eqs. 1 or 7) and (2) the isotope composition of stem water at any height h ($\delta^2 Hx_{(h; t)}$) is the isotope composition of stem water at the base, delayed by the travel time $^\tau$ of sap between stem base and height h ($\delta^2 Hx_{(h; t)} = \delta^2 Hx_{(0; t - \tau)}$) (Eq. 9). Soil properties are used as boundary conditions that do not change over the day in terms of soil water potential and isotopic composition. With such model, they predict large diurnal variations of xylem water isotopes at stem base, but also large variations along the stem (see their Figs. 1 and 2). Based on this modeling exercise, and separate observations of the $^2H=^1H$ ratio in water extracted from tree stems and lianas at different heights within a tropical forest canopy, and showing some scatter sometimes larger than 3‰ (the estimated error from water extraction and isotope analysis), they conclude that (1) the common assumption that the isotope composition of stem water is fairly constant over time is violated and (2) it can cause significant biases when using water isotopes to identify plant water origin.

I think it is good that the authors bring forward the point that xylem water may sometimes exhibit rather dynamic variations in its isotope composition. However, I am afraid the proposed model is inadequate and the dataset is too limited for illustrating this point. To me, the study does not prove anything; it shows that there are variations in the data and that there are variations in the model but there is no model-data comparison. Besides, variations in the data are not very large and can be explained by lots of other processes, and variations in the model are mostly caused by its lack of realism. These two points are explained in more detail below.

**We thank the reviewer for his/her detailed assessment of the study and helpful feedback on the manuscript. We do want to stress that our study is based on both a theoretical model exploration and empirical, novel field data. Moreover, we now present 3 independent datasets (datasets collected in French Guiana, in China, and Germany) on a variety of species, which all show pronounced variability in isotopic composition. We do not agree that all these data can be dismissed easily.**

**Additionally, we believe that all models can be criticized due to a lack of realism but their value depends on the insights they bring. In fact, process-based model explorations are proven tools in many scientific fields, because of the insights they provide and not because of their subjective realism. We hope to convince the reviewer that adding supplementary processes would indeed improve model realism and might impact the dynamics and absolute range of $\delta^2 H_X$, but it will not alter our conclusion: large variability of isotopic composition along woody stems is expected in many situations. Moreover, including some of the suggested realism strengthened our results.**

**We agree with the reviewer, however, that the message brought in our original manuscript had to be toned down to better reflect the limitations of the analysis and data. Therefore, we revised our manuscript, providing a more appropriate message by (i) down toning our statements and by (ii) including the positive aspects of diurnal variability in $\delta^2 H_X$, which could present new opportunities in water acquisition and plant performance**

studies (l618:622). At the moment we are unable to fully validate the model as such data does not yet exist to our knowledge. Moreover, the presented model serves as a theoretical exploration of one possible explanation that could cause the observed variance in $\delta^2H_X$. Here we apply generally accepted plant hydraulic processes which show that large variance in $\delta^2H_X$ is expected under the simulated (and realistic) conditions. We remain convinced that our findings, though not conclusive, can help build increased awareness of the potential of diurnal variability which can bias future isotope endeavors. At a minimum, it calls for more research. To meet the concerns of the reviewer, the new version of the manuscript more clearly mentions the limited coupling between model and data (L284:286; L361:363), as also suggested by reviewer 1. We acknowledge that other mechanisms could contribute to the observed $\delta^2H$ variance, and have extended the sections discussing other potential processes (see discussion and new performed analysis assessing if processes other than molecular diffusivity might contribute to the isotope transport along the xylem). Future model developments and targeted datasets are encouraged and are highlighted more in the new version of the manuscript (i.e. L600:603; L623:642; L663:667).

In short, we agree with most of the reviewer's comments, but we do not share his/her conclusion on the data and model.

- The French Guiana dataset (i.e. measuring isotopic composition along stems of lianas and trees) is indeed limited but is the first of its kind and does show intra-individual variations that are too large to be explained by extraction error only. We do not think that variances up to 20‰ $\delta^2H$ in a natural system should be considered as negligible.
- The model lacks realism for certain processes. It however does provide the insight that naturally arising changes in evaporative demand should lead to isotopic composition variability in woody stems due to their coupling to variable isotopic and soil water potential gradients (which are the basis of the isotopic studies). Adding model complexity, as the reviewer suggests, would allow us to refine both the ranges and the dynamics of the variation but will not prevent it. As we illustrate using the reviewer's suggestions below.

Hence our main objective (to illustrate the fact the xylem water isotopic composition does exhibit dynamic variations) still stands.

The dataset accompanying this study only consists of a few water isotope data from tree stems and lianas collected over a couple of days. No soil water data is shown, or even sap flow or rooting depths. I doubt it is the best dataset to test the proposed theory, or draw any conclusion about plant water uptake. The data shown in Fig. 5 is interesting but it comes from another study (Zhao L, Wang L, Liu X, Xiao H, Ruan Y, Zhou M (2014) The patterns and implications of diurnal variations in the d-excess of plant water, shallow soil water and air moisture. Hydrology and Earth System Sciences,18, 4129–4151). Many processes (stem evaporation, different proportions of storage tissues or even atmospheric vapour use) and measurement artefacts (during sampling and transport, water extraction, isotopic analysis…) could explain significant variations in the water composition of stems from trees and lianas of different statures. Accounting for uncertainties in the extraction and analysis is certainly not enough.

We more clearly discuss such limitations in the new version of the manuscript (L284:286; L361:363).

**This study indeed presents data collected by Zhao *et al.*, (2014). We note that this study is in collaboration with the authors who collected the data (please see the author list). The data we present show high temporal xylem water observations not presented in Zhao *et al.* (2014). This type of data is very rare in literature. Zhao *et al.* (2014) focused their paper on d-excess variability throughout the day, which is a derivative of $\delta^2H$ and $\delta^{18}O$ data. In our paper, we provide the raw $\delta^2H$ and $\delta^{18}O$ temporal data. Now also an additional dataset, collected in Germany is included in our study,** also **showing pronounced variance in $\delta^2H$ and $\delta^{18}O$ data.**

**Finally, we note that the factors the reviewer mentions were included in our discussion section. Not all of them will be an issue, while others will exacerbate bias. We will expand this discussion section to address the concerns of the reviewer. Here in short:**

- **Stem evaporation: This is indeed a good suggestion for non-woody plants. However, we target woody/lignified plants (this might have not been stated clearly enough, but should be now in the new version of the manuscript). Stem evaporation, especially when measuring relatively low in the stem (at 1.3m), should be negligible (see supplementary method A, L29-31).**
- **Different proportions of storage tissues: We fully agree with the reviewer that this presents a potential explanation of observed patterns, as discussed in the discussion section "iii. *Storage tissue and phloem enrichment*" (L580:603).**
- **Atmospheric vapor use: The reviewer presents another excellent argument of why applying the 'tracer isotopic technique' and sampling protocols should be re-assessed, addressing large variances in $\delta^2H_x$ and all potential contributing factors.**
- **Sampling protocol and extraction procedure: While sampling protocols and extraction procedures are never perfect, our extraction protocol is based on best practice as suggested by Orlowski *et al.*, (2013). Hence, considering extraction error rates of 3‰ are very cautious estimates, and actual error is most likely much less. Here we remark that despite these cautious estimates, we observe significant variability (as compared to the null model), which is remarkable and should be reported.**

More importantly, I find the modelling analysis flawed and totally unrealistic. As explained above, the proposed model simulates water isotope gradients along the stem based on the average travel time of sap between two stem points (i.e. assuming the water isotope composition of xylem water at height h is that at stem base at an earlier time corresponding to the travel time between stem base and height h). By doing so, the model neglects the mixing of water isotope by diffusion during water transport. If we neglect pit structure and consider vessels as regular pipes, the Péclet number $\wp$ that compares advection and diffusion is, using their notations: $\wp = SFV \ h = D_l$. Taking an average sap flow velocity of $SFV = 0.3 mh^{-1}$ (see caption of Fig. 2) a typical height (diffusion length) h of 1m and a self-diffusion of liquid water of $D_l = 2.5 \ 10^{-9} m^2 s^{-1}$ this leads a Péclet number $\wp$ around 30000, i.e. high enough to justify neglecting (a posteriori) water mixing by diffusion. However, mixing with storage tissues also occurs and tree sap does not move like a slab. In their model, as soon as transpiration stops, root uptake stops and sap flow at any height stops too so that the $\delta^2H$ of xylem water at any height remains to its value at dusk over the entire night until the following morning plus the time delay $\tau$ (see for example Fig. 2a, the curve for h=1.3m). In reality, at night, sap flow does not stop immediately because plant elastic tissues need to be replenished. Root uptake will continue until full replenishment of the elastic tissues is done. This will contribute to homogenisation of xylem water over night.

Also when sap flow becomes small diffusion is not negligible anymore (low Péclet), which will reinforce the isotopic mixing by water diffusion. In other words, in the real world, xylem water should not exhibit large isotopic gradients along the stem such as shown in their Figs. 1 or 2. Mixing with storage tissues is briefly discussed (section 4.3.iii) but not in the same direction as above. If night-time mixing of xylem water in roots and stems was accounted for, this should strongly minimize the predicted diurnal variations of _2Hx(h; t), even at stem base. Not accounting for diurnal variations in soil conditions (water potential and isotopic composition) is also a strong limitation of the model.

**We thank the reviewer for this excellent suggestion to use the Péclet effect to support neglecting diffusion in the model when sap flow is large enough. We have implemented diffusion in the model and have performed an analysis to evaluate the impact of molecular diffusion at night, when sap flow is zero, as assumed by the model at night (i.e. Péclet number becomes low as advective flow rate goes to zero, while diffusive flow rate remains constant, hence flow is dominated by diffusion at night). As the diffusion coefficient is low (i.e. $D_l = 3 \ 10^{-5} \ cm^2 \ s^{-1}$) the impact of diffusion at night is mostly negligible (12 hours of diffusion results in a smearing of the signature $\pm$ 4 cm). Diffusion causes an increase in the width of the $\delta^2H_X$-baseline drop, which means that the probability of sampling a non-representative section within this $\delta^2H_X$-baseline drop will increase. Including more realism hence increases bias in RWU estimates. It should be noted that diffusion will indeed reduce the absolute range of variability in $\delta^2H_X$ over time (but very slowly and very little), and hence with the height of the plant. However, it will not lead to homogenization overnight (which is by the way not observed in our supporting datasets, Fig 3c). This would require the accumulated impact of diffusion over many days, creating a time-lag between the measured isotopic composition of xylem and soil water, causing a decoupling of the signatures. These results and corresponding figures are implemented in the manuscript (supplementary method B, L466:472). In addition, we like to stress that for simplicity of the theoretical exploration, we deliberately chose for zero sap flow at night, as indicated in the model description. However, the model is flexible, and direct sap flow data complying with the wishes of the user can be implemented without a problem.**

**The impact validation of molecular diffusion did not show strong impacts on $\delta^2H_X$ dynamics along the length of the stem. However, this suggestion instigated a more in-depth assessment if other processes besides molecular diffusivity might contribute to isotope transport through the plant, especially when considering very low sap flow velocities (e.g. variable flow velocities within vessels and among vessels of the xylem network). We extend our study with a new analysis comparing the xylem transport in the model against a recent $^2H$ enrichment study of Marshall _et al._ (2020) (L466:482; L603:615, Fig 6). Marschall _et al._ (2020) applied a novel _in situ_ borehole equilibration technique for continuous monitoring $\delta^2H_X$ dynamics in a _Pinus pinea_ individual. This new analysis highlights the need for an improved understanding of $\delta^2H_X$ uptake and transport along trees. It further emphasizes the currently lack in understanding various important processes, besides diurnal fluctuations in RWU-activity, that might alter $\delta^2H_X$. These processes are currently ignored in the usual approach of using stable water isotopes for RWU assessment. Therefore, we further highlight and discuss the need for more intra-individual physiological and hydrological understanding via targeted studies, for the betterment of the current implementation of the stable water isotopic technique for RWU as well as the presented model.**

As we explored in the discussion section (see section "*Storage tissue and phloem enrichment*"; L580:603), homogenization of xylem water overnight depends on the assumption that storage water is representative of the water taken up by the roots. In fact, for several reasons, the isotopic composition of storage tissues is likely to deviate from the isotopic composition in soil water. This decouples the isotopic signature observed in xylem from the isotopic composition of the water mixture obtained by RWU and exacerbates potential bias in the isotope tracing technique. Unfortunately, empirical data on the isotopic composition of storage tissue is absent in literature to our knowledge, and this hampers the theoretical exploration of such a hypothesis. We highlight more clearly the importance of research targeting evaluation of the impact of storage water use by future studies, which would then allow the implementation of storage tissue in the model (L600:603). However, the presented empirical data do not show any indications of complete homogenization despite obtained from lianas and trees during the dry season (Fig 3c). This might suggest that storage tissue does not completely succeed in homogenizing $\delta^2H_X$ overnight as suggested by the reviewer. Therefore, in our opinion, the diurnal root water uptake fluctuation remains a convincing explanation for the observed variability in $\delta^2H_X$.

Finally, the reviewer is correct in pointing out that the absence of diurnal variations in soil conditions (water potential and isotopic composition) presents a limitation of the model. But this is already discussed in the discussion section "*Temporal and spatial soil dynamics*" (L565:579). However, temporal and spatial soil dynamics are generally very small given (a) the timeframe and (b) conditions in which stable isotopic tracing technique are studied, i.e., one-day sampling during dry conditions without rain are generally preferred. Hence, for all these conditions, the simplification of our model is acceptable in our opinion. Besides, our model implementations are flexible and if variable soil condition data are available, they can easily be implemented.

In conclusion, I find the argument raised by De Deurwaerder and colleagues not supported by their data nor by their model simulations. More realism would need to be brought to the model and the dataset should be complemented with additional information before drawing any conclusion on how variable the isotopic of xylem water in tree stems and lianas is over diurnal time scales or with height.

We understand the reservations of the reviewer for this study, as the coupling between data and the model exploration was not fully possible. Therefore, we toned down statements in the manuscript to better represent the limitations of the data and models. We do not agree that our data can be dismissed so easily: we stress that these are now three independent datasets that show pronounced variability in $\delta^2H_X$, which is illustrated for the first time. We also stress that the processes that reviewer suggested to increase realism do not change our conclusion itself: along the stem of woody plants, we can expect changes in water isotopic composition. We believe that arguments raised by both reviewers present additional incentives to re-assess and therefore further refine and improve the stable isotopic tracer technique.

[revised manuscript text omitted]
 15:00 to assure high sap flow. Since upstream $\delta_{xyl}$ enrichment due to Péclet effect, in close vicinity to evaporative surfaces has been observed in the literature (Dawson & Ehleringer, 1993; Barnard *et al.*, 2006), sampling was restricted to coring of the main stems. The order of sampling, i.e. ascending versus descending heights, was randomized. Tree stem xylem samples were collected with an increment borer (5 mm diameter), resulting in wooden cylinders from which bark and phloem tissues were removed. Coring was performed within the horizontal plane at the predefined heights, oblique to the center of the stem to maximize xylem and minimize heartwood sampling, and slowly to avoid heating the drill head and fractionation. Taking one sample generally took between 5 and 10 minutes. Since coring lianas was not possible, we collected cross-sections of the lianas after removing the bark and phloem tissue with a knife. Soil samples were collected at different depths (0.05, 0.15, 0.30, 0.45, 0.60, 0.90, 1.20, and 1.80m) within close vicinity to the sampled individuals using a soil auger. All materials were thoroughly cleaned between sampling using a dry cloth to avoid cross-contamination. Upon collection, all samples were placed in pre-weighed glass collection vials, using tweezers, to reduce contamination of the sample. Glass vials were immediately sealed with a cap and placed in a cooling box, to avoid water loss during transportation.

[revised manuscript text omitted]

**Method B:**

**Exploring the effect of diffusion on xylem transport of isotopes**

The current version of the model assumes a negligible impact of diffusion on the variance in the isotopic composition of the xylem water in the stem. Here, the validity of this assumption is discussed in more detail. We will use analytical and numerical solutions of the advection-diffusion equation to simulate the transport of isotope within the xylem, followed by a short discussion.

**Theory**

One-dimensional solute flux ($J$) of a solute concentration ($C$) through a pipe can be expressed as the sum of the advection and diffusion processes:

$$J = uC + q \tag{1}$$

where $u$ is the fluid flow velocity and $q$ the diffusion flux.

The one-directional diffusion flux along the direction $x$ can be expressed by Fick's law:

$$q = -D\frac{\partial C}{\partial x} \tag{2}$$

where $D$ ($m^2\ s^{-1}$) is the diffusion constant. The mass conservation can be written:

$$\frac{\partial C}{\partial t} = -\frac{\partial J}{\partial x} \tag{3}$$

*The diffusion equation*

Assuming no flow ($u = 0$) and inserting (2) into (3) we obtain:

$$\frac{\partial C}{\partial t} = D\frac{\partial^2 C}{\partial x^2} \tag{4}$$

Solutions of (4) for an instantaneous point source can be given in the form

$$C(x,t) = \frac{M}{\sqrt{4\pi Dt}} exp\left(-\frac{x^2}{4Dt}\right) \tag{5}$$

where $M$ is the mass of solute injected uniformly across the cross-section of the pipe at $x = 0$. Using the superimposition principle, we can also derive the solution for the one-dimensional stagnant case (an initial step function concentration without advection) as

$$C(x,t) = \frac{C_0}{2} erfc\left(\frac{x}{\sqrt{4\pi Dt}}\right) \tag{6}$$

where $C_0$ is the initial concentration at $x < 0$ and erfc is the complementary error function.

*Advection-diffusion equation*

In the case of flow with velocity, (4) is modified as:

$$\frac{\partial C}{\partial t} = D\frac{\partial^2 C}{\partial x^2} + u\frac{\partial C}{\partial x} \tag{7}$$

The solution for constant concentration at $x = 0$ with initial zero concentration on a semi-
infinite domain, i.e.

$$\begin{cases} C(x,0) = 0, & x > 0 \\ C(0,t) = C_0, & t > 0 \end{cases} \tag{8}$$

is given by (Ogata & Banks, 1961):

$$C(x,t) = \frac{C_0}{2}\left(erfc\left(\frac{x-ut}{\sqrt{4\pi Dt}}\right) + exp\left(\frac{xu}{D}\right)erfc\left(\frac{x+ut}{\sqrt{4\pi Dt}}\right)\right) \tag{9}$$

This solution can describe the dynamic of a solute concentration along the xylem under constant
velocity, with a fixed concentration at the inlet point.

Numerical solutions

Solutions for problems with different boundary conditions and variable velocity are not
available. In order to investigate the case with periodic concentrations at the inlet of the pipe
and periodic velocity we used numerical solutions of the advection-diffusion equation

$$\frac{\partial C}{\partial t} = D\frac{\partial^2 C}{\partial x^2} + u_0 f(t)\frac{\partial C}{\partial x} \tag{10}$$

where $f(t)$ is a periodic function. We used the wrapped normal distribution defined as

$$f(t) = \sum_{i=-100}^{i=100} exp\left[\frac{\left(\frac{2\pi t}{24} - \pi - 2\pi k\right)^2}{2\sigma^2}\right] \tag{11}$$

The boundary conditions at the inlet and outlet are defined as

$$\begin{cases} C = (C_{max} + C_{min})g(t) + C_{min} & x = 0, t > 0 \\ \frac{\partial C}{\partial t} = 0 & x = H, t > 0 \end{cases} \tag{12}$$

where $g(t)$ is another periodic function defined as

$$g(t) = \sum_{i=-100}^{i=100} \exp\left[\frac{\left|\frac{2\pi t}{24} - \pi - 2\pi k\right|^3}{2\sigma^3}\right] \qquad (13)$$

The third power in (13) was chosen to match the diurnal cycle of the isotopic concentration at
the tree base obtained by SWIFT. The equation was solved using the function pdepe
implemented in Matlab (R2019a), explicitly designed to solve initial-boundary value problems
for parabolic-elliptic partial differential equations in 1-D (Skeel & Berzins, 1990).

Unfortunately, numerical solutions of the advection-diffusion equation suffer numerical
oscillation for values of the Péclet number greater than one (Zienkiewicz *et al.*, 2000), so results
are presented for values of diffusivity 50, 100, 200 and 400 cm$^2$ hr$^{-1}$. These values are much
larger than the diffusivity of heavy water and they will produce stronger smoothing.

[Figure]

**Fig B1:** Analytical solutions of advection-diffusion equation on a semi-infinite 1-D domain
(Eq. (9)) with 12 ‰ step-change in isotope signature for different values of flow velocity and
diffusivity. The plots show the impact of diffusion on the isotopic composition of xylem water.
Colored lines show the solution at different time intervals: 0, 12, 24, 48, and 96 hr. Note that
the values of diffusivity are much higher than these reported for heavy water (e.g. D=0.1 cm$^2$
h$^{-1}$; Meng et al., 2018)

[Figure]

**Fig B2:** Numerical solutions of advection-diffusion equation on a finite 1-D domain (Eq. (10-13)) with 12 ‰ step-change in isotope signature for different values of diffusivity along the length of the xylem. The periodic forcing used in the simulations are shown in panel a and b. Panels c and d show the solutions for two different time of the day. Colored lines show the solution at different diffusivity (see legend in d). Note that the values of diffusivity are much higher than these reported for heavy water (e.g. D=0.1 cm² h⁻¹; Meng et al., 2018).

**Results and Discussion**

The diffusivity of $^2$H in water depends on temperature: at 20 °C is D = 6.87 10$^{-2}$ cm² hr$^{-1}$, at 40 °C is D = 1.37 10$^{-1}$ cm² hr$^{-1}$ (Meng et al., 2018). Another process that can cause substantial mixing is the random movement of particles in the xylem network. Within each vessel, the flow is laminar, but in vessels with a larger diameter, velocity is higher than in vessels with a smaller diameter. According to the Hagen–Poiseuille law, the flow is proportional to the fourth power of diameter (hence, the velocity is proportional to diameter square). Therefore, the variable velocity experienced by the particles in the xylem network can generate substantial random motion in the transport of a solute in a similar manner of diffusion in a porous media.

Molecular diffusivity results in a relatively negligible impact of diffusion on the variance in $^2$H when high sap flux densities are considered, as shown in Fig B1. For example, for diffusivity of 0.1 cm² hr$^{-1}$, after 96 hours, diffusion results in smearing in a range ± 10cm (Fig. B1a). The case with a flow velocity of 25 cm hr$^{-1}$, comparable to the velocity of sap in xylem, shows that the transport of the solute is minimally affected by diffusion (Fig B1 a and c). In order to appreciate the effect of diffusion, the diffusivity needs to increase three orders of magnitude (Fig B1 b and d). However, because homogenization increases with time, the impact of diffusion on $\delta^2$H dynamics can be non-negligible for very low sap flux velocities.

Numerical solutions with the periodic forcing (Fig B2 a and b), show that for high values of diffusivity there could be a substantial smoothing in the peak (Fig B2 c and d). The smoothing progress along the path-length of the flow. However, note that a very high value of diffusivity ($>400$ cm$^2$ hr$^{-1}$) is required for complete homogenization above 10 m.

For the general application to isotope transport in xylem with variable input concentrations and variable sap flow velocity, diffusion can cause a smoothing of the peak and a consequent increase in the width of the $\delta^2$H$_X$-baseline drop. Therefore, the probability of sampling a non-representative section within this $\delta^2$H$_X$-baseline might increase, which means that neglecting diffusion could lead towards a conservative assessment of the bias in RWU estimates. However, the minimal reduction of the peak in $\delta^2$H$_X$ over time might lead to reducing the variability in time and space compared to the case with no diffusion. In conclusion, while diffusion does affect both the absolute range of $\delta^2$H$_X$ variance and the width of the $\delta^2$H$_X$-baseline drop (i.e. increased probability of extracting biased samples), the impact is small in the lower part of the tree and over the timeframe and sap flow flux considered in this study. Hence, for this study, diffusion will not result in the complete homogenization of the $\delta^2$H$_X$ along the length of the studied trees, consistent with empirical datasets (Fig 3c, Fig S2.).

processes besides molecular diffusivity might contribute to isotope transport through the plant,
especially when very low sap flow velocities are considered. Specifically, the experiment
follows the impact of a stepwise $^2$H enrichment of the source water, i.e. from $\delta^2H=-59.28 \pm$
0.24 ‰ to $\delta^2H=290.57 \pm 3.08‰$ (see Fig 6), on the $\delta^2H_X$ dynamics in a pine tree (*Pinus pinea*
*L.*). The tree was placed in a large pot, with the root system fully submerged in aerated water
(using mini-pumps) and subjected to artificial light conditions (12h light, 12h dark, light
transition at 7:00 o'clock). $\delta^2H_X$ was monitored continuously and *in situ* at two sampling
heights, 0.15 cm, and 0.65 cm, respectively, using a novel borehole technique. Concomitant,
sap flow velocity was measured using a sap flow sensor (heat pulse velocity sensor, Edaphic
Scientific, Australia), installed at 0.85m height, and perpendicular to the upper borehole. For
specific details of this experiment, we refer to Marshall *et al.* (2020).

In this setup, roots are submerged in a uniform isotopic solution, so the SWIFT model
parameterization of soil and root is not necessary. The isotopic composition of the source water
will, therefore, almost instantly reflect the $\delta^2H$ at the stem base. The impact of diffusion could
not be considered negligible as sap flow velocities are very low (daily mean $SF_V = 0.97 \pm 0.39$
cm h$^{-1}$) and the experiment lasted out 38 days before equilibrium was reached between the
δ2HX of the source water and the δ2HX in both boreholes. For simulating the isotopic
dynamics, we used an analytical solution of the advection-diffusion, as described in
supplementary methods B, coupled to the SWIFT model. Model parameters, velocity, and
diffusion were fitted by visual inspection independently for the two heights to match the initial
increase in isotope signature.
Note that the studied tree shows strong tapering (diam. at 0.15cm = 9.9cm; diam. at 0.65cm =
8.0cm), causing an acceleration of the sap flow along the pathway length as a same volume of
water is propelled through a diminishing cross-area. This is also reflected in the allocated
velocity parameters.

**Sensitivity analyses**
We first assessed model sensitivity to (bio)physical variables by modifying model parameters
of soil type, sap flow, and root properties as compared to the standard parameterization (given
in Table S1). The following sensitivity analyses were considered:

**Soil type:** The soil moisture content overall soil layers ($\theta_{S,i,t}$) can be deduced from the
considered Meißner et al. (2012) $\Psi_{S,i,t}$ profile (see Fig. S8 and Table S1) using the Clapp
& Hornberger (1978) equation:

[revised manuscript text omitted]

**Figures and tables**

[Figure]

**Fig. S1.** Oxygen isotope composition ($\delta^{18}O$, in ‰ V-SMOW) of bulk soil water sampled at different depths (red), xylem water of lianas (orange) and trees (green), and from bulk stream (blue) and bulk precipitation water (cyan) in Laussat, French Guiana. Different soil $\delta^{18}O$ composition symbols indicate the extraction recovery rates, where 98% presents the generally pursued benchmark. Shaded areas show the Q25-Q75 intervals for lianas and trees in orange and green respectively.

[Figure]

**Fig. S2.** Field measurements of normalized intra-individual $\delta^2 H_X$ ($\beta^2 H_X$) for six lianas (panel a) and six trees (panel b). Individuals are provided in different colors; liana species: ■ *Coccoloba sp.*, ■ sp.2, ■ sp.3, ■ *cf. rotundifolium Rich.*, ■ *Maripa cf violacea*, ■ *Maripa sp.*; tree species: ■ *Coussapoa sp.*, ■ *Vouacapoua americana*, ■ *Erisma nitidum*, ■ *Micropholis guyanensis*, ■ *Tapirira guyanensis*, ■ *Albizia pedicellaris*. Error whiskers are the combination of potential extraction and measurement errors of the isotope analyzer. The former presents a positive skew-normal distribution $SN_{empirical}(\xi = 0‰, \omega = 3‰, \alpha = +\infty)$. The full grey envelope delineates the acceptable variance from the stem mean (i.e. 3‰) according to the standard assumption of no variance along the length of a lignified plant, i.e the null model.

[Figure]

**Fig. S3**. High temporal field measurements of normalized $\delta^2H$ composition of xylem water ($\beta^2H_X$) of two trees (red, stem samples), two shrubs (blue, stem samples) and two herbs (green, root samples) species sampled in the Heihe River Basin (northwestern China) shown for the respective measurement periods. Timing and location of sampling are provided in the panel titles. Horizontal dark grey colored envelope delineates the acceptable variance from the stem mean (i.e. 3‰) according to the standard assumption of no variance along the length of a lignified plant. Light grey vertical envelopes mark the nighttime periods. The table provides the maximum measured diurnal $\delta^2H_X$ range per species.

[Figure]

**Fig. S4**. High temporal field measurements of normalized $\delta^{18}O$ composition of xylem water ($\beta^{18}O_X$) of two trees (red, stem samples), two shrubs (blue, stem samples) and two herbs (green, root samples) in the Heihe River Basin (northwestern China) shown for the respective measurement period. Timing and location of sampling are provided in the panel title. Horizontal dark grey colored envelope delineates the acceptable variance from the stem mean (i.e. 0.3‰) according to the standard assumption of no variance along the length of a lignified plant. Light grey vertical envelopes mark the nighttime periods. The table provides the maximum measured diurnal $\delta^{18}O_X$ range per species.

[Figure]

**Fig. S5**. High temporal field measurements of normalized $\delta^2$H composition of xylem water ($\beta^2 H_X$) of three *Abies alba* individuals (blue, branch samples) and three *Fagus sylvatica* individuals (red, branch samples) sampled during a drought period in July 2017 in the "Freiamt" field site in south-west Germany. Horizontal dark grey colored envelope delineates the acceptable variance from the stem mean (i.e. 3‰) according to the standard assumption of no variance along the length of a lignified plant. Light grey vertical envelopes mark the nighttime periods.

[Figure]

**Fig. S6**. High temporal field measurements of normalized $\delta^{18}O$ composition of xylem water
($\beta^{18}O_X$) of three *Abies alba* individuals (blue, branch samples) and three *Fagus sylvatica*
individuals (red, branch samples) sampled during a drought period in July 2017 in the "Freiamt"
field site in south-west Germany. Horizontal dark grey colored envelope delineates the
acceptable variance from the stem mean (i.e. 0.3‰) according to the standard assumption of no
variance along the length of a lignified plant. Light grey vertical envelopes mark the nighttime
periods.

[Figure]

**Fig S7:** Sap flow rate (*SF*, blue line), $\delta^2$H composition of xylem water at stem base ($\
[revised manuscript text omitted]
 RB, Canadell J, Ehleringer JR, Mooney HA, Sala OE, Schulze ED. 1996.** A global analysis of root distributions for terrestrial biomes. *Oecologia* **108**: 389–411.

**Leuschner C, Coners H, Icke R. 2004.** In situ measurement of water absorption by fine roots of three temperate trees: species differences and differential activity of superficial and deep roots. *Tree Physiology* **24**: 1359–1367.

**Meinzer FC, Goldstein G, Andrade JL. 2001**. Regulation of water flux through tropical forest canopy trees: do universal rules apply? *Tree physiology* **21**: 19–26.

**Meißner M, Köhler M, Schwendenmann L, Hölscher D. 2012.** Partitioning of soil water among canopy trees during a soil desiccation period in a temperate mixed forest. *Biogeosciences* **9**: 3465–3474.

**Rüdinger M, Hallgren SW, Steudle E, Schulze E-D. 1994.** Hydraulic and osmotic properties of spruce roots. *Journal of Experimental Botany* **45**: 1413–1425.

**Sands R, Fiscus EL, Reid CPP. 1982.** Hydraulic properties of pine and bean roots with varying degrees of suberization, vascular differentiation and mycorrhizal infection. *Functional Plant Biology* **9**: 559–569.

**Steudle E, Meshcheryakov AB. 1996.** Hydraulic and osmotic properties of oak roots. *Journal of Experimental Botany* **47**: 387–401.

**Zanne AE, Westoby M, Falster DS, Ackerly DD, Loarie SR, Arnold SEJ, Coomes DA. 2010.** Angiosperm wood structure: global patterns in vessel anatomy and their relation to wood density and potential conductivity. *American Journal of Botany* **97**: 207–215.

---

## Author Response (AR2)

Dear Biogeosciences editor,

We are grateful for the involvement and suggestions of both the referees and editor, which have so far greatly improved the quality of our manuscript. The new textual suggestions of referee #1 have now also been implemented, as detailed below (responses to reviewers in bold). We hope that these adjustments in the manuscript allow publication in *Biogeosciences*.

Yours sincerely,

The authors
* * *
**REVIEWER #1 COMMENTS**

Remove consistently "deuterium" from the manuscript.

➔ **The word 'deuterium' is removed consistently from both the manuscripts and the supplementary data.**

L29/60. Add "relative" before "root water uptake profiles"

➔ **The word 'relative' has now been added following the reviewer suggestion. [L29, 61, 869]**

L395. You are almost there ☺. It should read something like "and where soil water is more depleted in [2H] in comparison with the soil layers above" (i.e., not δ2H)

➔ **This sentence has been altered and now reeds as: 'Early in the morning, when transpiration is low, most of the RWU occurs in deeper layers, where soil matric potential is less negative and where soil water is more depleted in $^2H$ compared with the soil layers above (Fig. S8a-b).' [L394:396]**

L398/403/536-537. A δ value cannot be enriched.

➔ **These sentences have now been altered, highlighting that not the δ-value, but the sampled water is enriched. [L398, 403, 534:537, 567:568]**

L456. "to access more shallow and enriched soil layer[ water]"

➔ **This sentence now reads as "Higher $SF_t$ requires more negative $\widehat{\Psi}_{X,0,t}$, enabling the plant to access the enriched soil water of more shallow soil layers". [L455:456]**

L478-479. Correct terminology "…isotope…"

[revised manuscript text omitted]
 15:00 to assure high sap flow. Since upstream $\delta_{xyl}$ enrichment due to Péclet effect, in close vicinity to evaporative surfaces has been observed in the literature (Dawson & Ehleringer, 1993; Barnard *et al.*, 2006), sampling was restricted to coring of the main stems. The order of sampling, i.e. ascending versus descending heights, was randomized. Tree stem xylem samples were collected with an increment borer (5 mm diameter), resulting in wooden cylinders from which bark and phloem tissues were removed. Coring was performed within the horizontal plane at the predefined heights, oblique to the center of the stem to maximize xylem and minimize heartwood sampling, and slowly to avoid heating the drill head and fractionation. Taking one sample generally took between 5 and 10 minutes. Since coring lianas was not possible, we collected cross-sections of the lianas after removing the bark and phloem tissue with a knife. Soil samples were collected at different depths (0.05, 0.15, 0.30, 0.45, 0.60, 0.90, 1.20, and 1.80m) within close vicinity to the sampled individuals using a soil auger. All materials were thoroughly cleaned between sampling using a dry cloth to avoid cross-contamination. Upon collection, all samples were placed in pre-weighed glass collection vials, using tweezers, to reduce contamination of the sample. Glass vials were immediately sealed with a cap and placed in a cooling box, to avoid water loss during transportation.

**Sample processing**
Sample processing was performed as in De Deurwaerder *et al.* (2018). Specifically, all fresh samples were weighed, transported in a cooler, and frozen before cryogenic vacuum distillation (CVD). Water was extracted from the samples via CVD (4 h at 105°C). Water recovery rates were calculated from the fresh weight, weight after extraction, and oven-dry weight (48 h at 105°C). Samples were removed from the analysis whenever weight loss resulting from the extraction process was below 98% (after Araguás-Araguás *et al.*, 1998). Nearly all soil samples fell below this benchmark and were therefore excluded from further analysis (Fig S1). The isotope composition of the water in the samples was measured by a Wavelength-Scanned-

Cavity Ring-Down Spectrometer (WS-CRDS, L2120-i, Picarro, California, USA) coupled with a vaporizing module (A0211 High Precision Vaporizer) through a micro combustion module to avoid organic contamination (Martin-Gomez *et al.*, 2015; Evaristo *et al.*, 2016). Post-processing of raw δ-readings into calibrated δ-values was performed using SICalib (version 2.16; Gröning, 2011) and internal laboratory references, i.e. Lab1 ($\delta^2$H: 7.74±0.4‰; $\delta^{18}$O: 5.73±0.06‰,), Lab3 ($\delta^2$H: -146.98±0.4‰; $\delta^{18}$O: -20.01±0.06‰,) and quality assurance samples ($\delta^2$H: -48.68±0.4‰; $\delta^{18}$O: -7.36±0.06‰). Calibrated δ-values are expressed on the international V-SMOW scale.

**Table A1.** Sampled liana and tree individuals, provided with their species, respective diameter at breast height (DBH, in cm) and their $\delta^2H$

and $\delta^{18}O$ ranges (in ‰, VSMOW) measured per individual.

| Code | Growth form | DBH [cm] | Family | Species name | $\delta^2H_X$-range [in ‰, VSMOW] | $\delta^{18}O_X$-range [in ‰, VSMOW] |
|---|---|---|---|---|---|---|
| SP1 | Tree | 15.6 | Moraceae | *Coussapoa sp.* | -30.1; -25.5 | -2.8; -2.6 |
| SP2 | Tree | 50.9 | Fabaceae | *Vouacapoua americana* | -23.9; -18.1 | -3.1; -2.2 |
| SP3 | Tree | 44.6 | Vochysiaceae | *Erisma nitidum* | -27.7; -20.8 | -3.2; -1.9 |
| SP4 | Tree | 26.1 | Sapotaceae | *Micropholis guyanensis* | -29.8; -28.0 | -3.0; -2.9 |
| SP5 | Tree | 21.0 | Anacardiaceae | *Tapirira guyanensis* | -31.1; -18.0 | -3.2; -2.2 |
| SP6 | Tree | 49.7 | Fabaceae | *Albizia pedicellaris* | -26.9; -22.1 | -3.2; -2.6 |
| SP1 | Liana | 2.8 | Polygonaceae | *Coccoloba sp.* | -27.9; -20.7 | -3.9; -2.3 |
| SP2 | Liana | 2.7 | Convolvulaceae | *sp.* | -29.3; -24.0 | -4.4; -2.9 |
| SP3 | Liana | 0.8 | Moraceae | *sp.* | -40.8; -22.6 | -4.5; -2.3 |
| SP4 | Liana | 3.8 | Combretaceae | *cf. rotundifolium Rich.* | -23.6; -15.2 | -2.9; -2.0 |
| SP5 | Liana | 0.7 | Convolvulaceae | *Maripa cf violacea* | -31.6; -19.7 | -3.8; -2.7 |
| SP6 | Liana | 3.8 | Convolvulaceae | *Maripa sp.* | -35.3; -24.4 | -4.8; -3.1 |

**Method B:**

105**Exploring the effect of diffusion on xylem transport of isotopes**

107The current version of the model assumes a negligible impact of diffusion on the variance in
108the isotopic composition of the xylem water in the stem. Here, the validity of this assumption
109is discussed in more detail. We will use analytical and numerical solutions of the advection-
110diffusion equation to simulate the transport of isotope within the xylem, followed by a short
111discussion.

113**Theory**

114One-dimensional solute flux ($J$) of a solute concentration ($C$) through a pipe can be expressed
115as the sum of the advection and diffusion processes:

116$$J = uC + q \tag{1}$$

117where $u$ is the fluid flow velocity and $q$ the diffusion flux.

118The one-directional diffusion flux along the direction $x$ can be expressed by Fick's law:

119$$q = -D\frac{\partial C}{\partial x} \tag{2}$$

120where $D$ (m$^2$ s$^{-1}$) is the diffusion constant. The mass conservation can be written:

121$$\frac{\partial C}{\partial t} = -\frac{\partial J}{\partial x} \tag{3}$$

123*The diffusion equation*

124Assuming no flow ($u = 0$) and inserting (2) into (3) we obtain:

125$$\frac{\partial C}{\partial t} = D\frac{\partial^2 C}{\partial x^2} \tag{4}$$

126Solutions of (4) for an instantaneous point source can be given in the form

127$$C(x,t) = \frac{M}{\sqrt{4\pi Dt}}exp\left(-\frac{x^2}{4Dt}\right) \tag{5}$$

128where $M$ is the mass of solute injected uniformly across the cross-section of the pipe at $x = 0$.
129Using the superimposition principle, we can also derive the solution for the one-dimensional
130stagnant case (an initial step function concentration without advection) as

$$C(x,t) = \frac{C_0}{2} erfc\left(\frac{x}{\sqrt{4\pi Dt}}\right) \tag{6}$$

where $C_0$ is the initial concentration at $x < 0$ and erfc is the complementary error function.

*Advection-diffusion equation*

In the case of flow with velocity, (4) is modified as:

$$\frac{\partial c}{\partial t} = D\frac{\partial^2 c}{\partial x^2} + u\frac{\partial c}{\partial x} \tag{7}$$

The solution for constant concentration at $x = 0$ with initial zero concentration on a semi-infinite domain, i.e.

$$\begin{cases} C(x,0) = 0, & x > 0 \\ C(0,t) = C_0, & t > 0 \end{cases} \tag{8}$$

is given by (Ogata & Banks, 1961):

$$C(x,t) = \frac{C_0}{2}\left(erfc\left(\frac{x-ut}{\sqrt{4\pi Dt}}\right) + exp\left(\frac{xu}{D}\right) erfc\left(\frac{x+ut}{\sqrt{4\pi Dt}}\right)\right) \tag{9}$$

This solution can describe the dynamic of a solute concentration along the xylem under constant velocity, with a fixed concentration at the inlet point.

Numerical solutions

Solutions for problems with different boundary conditions and variable velocity are not available. In order to investigate the case with periodic concentrations at the inlet of the pipe and periodic velocity we used numerical solutions of the advection-diffusion equation

$$\frac{\partial c}{\partial t} = D\frac{\partial^2 c}{\partial x^2} + u_0 f(t)\frac{\partial c}{\partial x} \tag{10}$$

where $f(t)$ is a periodic function. We used the wrapped normal distribution defined as

$$f(t) = \sum_{i=-100}^{i=100} exp\left[\frac{\left(\frac{2\pi t}{24} - \pi - 2\pi k\right)^2}{2\sigma^2}\right] \tag{11}$$

The boundary conditions at the inlet and outlet are defined as

$$\begin{cases} C = (C_{max} + C_{min})g(t) + C_{min} & x = 0, t > 0 \\ \frac{\partial c}{\partial t} = 0 & x = H, t > 0 \end{cases} \tag{12}$$

where $g(t)$ is another periodic function defined as

$$g(t) = \sum_{i=-100}^{i=100} \exp\left[\frac{\left|\frac{2\pi t}{24} - \pi - 2\pi k\right|^3}{2\sigma^3}\right] \qquad (13)$$

The third power in (13) was chosen to match the diurnal cycle of the isotopic concentration at the tree base obtained by SWIFT. The equation was solved using the function *pdepe* implemented in Matlab (R2019a), explicitly designed to solve initial-boundary value problems for parabolic-elliptic partial differential equations in 1-D (Skeel & Berzins, 1990).

Unfortunately, numerical solutions of the advection-diffusion equation suffer numerical oscillation for values of the Péclet number greater than one (Zienkiewicz *et al.*, 2000), so results are presented for values of diffusivity 50, 100, 200 and 400 cm$^2$ hr$^{-1}$. These values are much larger than the diffusivity of heavy water and they will produce stronger smoothing.

[Figure]

**Fig B1:** Analytical solutions of advection-diffusion equation on a semi-infinite 1-D domain (Eq. 9) with 12 ‰ step-change in isotope signature for different values of flow velocity and diffusivity. The plots show the impact of diffusion on the isotopic composition of xylem water. Colored lines show the solution at different time intervals: 0, 12, 24, 48, and 96 hr. Note that the values of diffusivity are much higher than these reported for heavy water (e.g. D=0.1 cm$^2$ h$^{-1}$; Meng et al., 2018)

[Figure]

**Fig B2:** Numerical solutions of advection-diffusion equation on a finite 1-D domain (Eq. 10-13) with 12 ‰ step-change in isotope signature for different values of diffusivity along the length of the xylem. The periodic forcing used in the simulations are shown in panel a and b. Panels c and d show the solutions for two different time of the day. Colored lines show the solution at different diffusivity (see legend in d). Note that the values of diffusivity are much higher than these reported for heavy water (e.g. D=0.1 cm$^2$ h$^{-1}$; Meng et al., 2018).

**Results and Discussion**

The diffusivity of $^2$H in water depends on temperature: at 20 ºC is D = 6.87 10$^{-2}$ cm² hr$^{-1}$, at 40 ºC is D = 1.37 10$^{-1}$ cm² hr$^{-1}$ (Meng et al., 2018). Another process that can cause substantial mixing is the random movement of particles in the xylem network. Within each vessel, the flow is laminar, but in vessels with a larger diameter, velocity is higher than in vessels with a smaller diameter. According to the Hagen–Poiseuille law, the flow is proportional to the fourth power of diameter (hence, the velocity is proportional to diameter square). Therefore, the variable velocity experienced by the particles in the xylem network can generate substantial random motion in the transport of a solute in a similar manner of diffusion in a porous media.

Molecular diffusivity results in a relatively negligible impact of diffusion on the variance in $^2$H when high sap flux densities are considered, as shown in Fig B1. For example, for diffusivity of 0.1 cm² hr$^{-1}$, after 96 hours, diffusion results in smearing in a range ± 10cm (Fig. B1a). The case with a flow velocity of 25 cm hr$^{-1}$, comparable to the velocity of sap in xylem, shows that the transport of the solute is minimally affected by diffusion (Fig B1 a and c). In order to appreciate the effect of diffusion, the diffusivity needs to increase three orders of magnitude (Fig B1 b and d). However, because homogenization increases with time, the impact of diffusion on $\delta^2$H dynamics can be non-negligible for very low sap flux velocities.

Numerical solutions with the periodic forcing (Fig B2 a and b), show that for high values of diffusivity there could be a substantial smoothing in the peak (Fig B2 c and d). The smoothing progress along the path-length of the flow. However, note that a very high value of diffusivity (>400 cm$^2$ hr$^{-1}$) is required for complete homogenization above 10 m.

For the general application to isotope transport in xylem with variable input concentrations and variable sap flow velocity, diffusion can cause a smoothing of the peak and a consequent increase in the width of the $\delta^2$H$_X$-baseline drop. Therefore, the probability of sampling a non-representative section within this $\delta^2$H$_X$-baseline might increase, which means that neglecting diffusion could lead towards a conservative assessment of the bias in RWU estimates. However, the minimal reduction of the peak in $\delta^2$H$_X$ over time might lead to reducing the variability in time and space compared to the case with no diffusion. In conclusion, while diffusion does affect both the absolute range of $\delta^2$H$_X$ variance and the width of the $\delta^2$H$_X$-baseline drop (i.e. increased probability of extracting biased samples), the impact is small in the lower part of the tree and over the timeframe and sap flow flux considered in this study. Hence, for this study, diffusion will not result in the complete homogenization of the $\delta^2$H$_X$ along the length of the studied trees, consistent with empirical datasets (Fig 3c, Fig S2.).

mechanics. *Butterworth-heinemann.*

**Method C:**

**A detailed description of the performed transport dynamics and sensitivity analyses.**

**Transport dynamics**

The intact-root greenhouse experiment of Marshall *et al.* (2020) allows assessment if other processes besides molecular diffusivity might contribute to isotope transport through the plant, especially when very low sap flow velocities are considered. Specifically, the experiment follows the impact of a stepwise $^2$H enrichment of the source water, i.e. from $\delta^2H$=-59.28 ± 0.24 ‰ to $\delta^2H$=290.57 ± 3.08‰ (see Fig 6), on the $\delta^2H_X$ dynamics in a pine tree (*Pinus pinea L.*). The tree was placed in a large pot, with the root system fully submerged in aerated water (using mini-pumps) and subjected to artificial light conditions (12h light, 12h dark, light transition at 7:00 o'clock). $\delta^2H_X$ was monitored continuously and *in situ* at two sampling heights, 0.15 cm, and 0.65 cm, respectively, using a novel borehole technique. Concomitant, sap flow velocity was measured using a sap flow sensor (heat pulse velocity sensor, Edaphic Scientific, Australia), installed at 0.85m height, and perpendicular to the upper borehole. For specific details of this experiment, we refer to Marshall *et al.* (2020).

In this setup, roots are submerged in a uniform isotopic solution, so the SWIFT model parameterization of soil and root is not necessary. The isotopic composition of the source water will, therefore, almost instantly reflect the $\delta^2H$ at the stem base. The impact of diffusion could not be considered negligible as sap flow velocities are very low (daily mean $SF_V = 0.97 \pm 0.39$ cm h$^{-1}$) and the experiment lasted out 38 days before equilibrium was reached between the $\delta^2H_X$ of the source water and the $\delta^2H_X$ in both boreholes. For simulating the isotopic dynamics, we used an analytical solution of the advection-diffusion, as described in supplementary methods B, coupled to the SWIFT model. Model parameters, velocity, and diffusion were fitted by visual inspection independently for the two heights to match the initial increase in isotope signature. Note that the studied tree shows strong tapering (diam. at 0.15cm = 9.9cm; diam. at 0.65cm = 8.0cm), causing an acceleration of the sap flow along the pathway length as a same volume of water is propelled through a diminishing cross-area. This is also reflected in the allocated velocity parameters.

**Sensitivity analyses**

We first assessed model sensitivity to (bio)physical variables by modifying model parameters of soil type, sap flow, and root properties as compared to the standard parameterization (given in Table S1). The following sensitivity analyses were considered:

**Soil type:** The soil moisture content overall soil layers ($\theta_{S,i,t}$) can be deduced from the considered Meißner et al. (2012) $\Psi_{S,i,t}$ profile (see Fig. S8 and Table S1) using the Clapp & Hornberger (1978) equation:

[revised manuscript text omitted]

**Figures and tables**

[Figure]

**Fig. S1.** Oxygen isotope composition ($\delta^{18}O$, in ‰ V-SMOW) of bulk soil water sampled at different depths (red), xylem water of lianas (orange) and trees (green), and from bulk stream (blue) and bulk precipitation water (cyan) in Laussat, French Guiana. Different soil $\delta^{18}O$ composition symbols indicate the extraction recovery rates, where 98% presents the generally pursued benchmark. Shaded areas show the Q25-Q75 intervals for lianas and trees in orange and green respectively.

[Figure]

**Fig. S2.** Field measurements of normalized intra-individual $\delta^2H_X$ ($\beta^2H_X$) for six lianas (panel a) and six trees (panel b). Individuals are provided in different colors; liana species: ■ *Coccoloba sp.*, ■ sp.2, ■ sp.3, ■ *cf. rotundifolium Rich.*, ■ *Maripa cf violacea*, ■ *Maripa sp.*; tree species: ■ *Coussapoa sp.*, ■ *Vouacapoua americana*, ■ *Erisma nitidum*, ■ *Micropholis guyanensis*, ■ *Tapirira guyanensis*, ■ *Albizia pedicellaris*. Error whiskers are the combination of potential extraction and measurement errors of the isotope analyzer. The former presents a positive skew-normal distribution SN$_{empirical}$($\xi$ =0‰, $\omega$=3‰, $\alpha$=+∞). The full grey envelope delineates the acceptable variance from the stem mean (i.e. 3‰) according to the standard assumption of no variance along the length of a lignified plant, i.e the null model.

[Figure]

**Fig. S3**. High temporal field measurements of normalized $\delta^2H$ composition of xylem water ($\beta^2H_X$) of two trees (red, stem samples), two shrubs (blue, stem samples) and two herbs (green, root samples) species sampled in the Heihe River Basin (northwestern China) shown for the respective measurement periods. Timing and location of sampling are provided in the panel titles. Horizontal dark grey colored envelope delineates the acceptable variance from the stem mean (i.e. 3‰) according to the standard assumption of no variance along the length of a lignified plant. Light grey vertical envelopes mark the nighttime periods. The table provides the maximum measured diurnal $\delta^2H_X$ range per species.

[Figure]

**Fig. S4**. High temporal field measurements of normalized $\delta^{18}O$ composition of xylem water ($\beta^{18}O_X$) of two trees (red, stem samples), two shrubs (blue, stem samples) and two herbs (green, root samples) in the Heihe River Basin (northwestern China) shown for the respective measurement period. Timing and location of sampling are provided in the panel title. Horizontal dark grey colored envelope delineates the acceptable variance from the stem mean (i.e. 0.3‰) according to the standard assumption of no variance along the length of a lignified plant. Light grey vertical envelopes mark the nighttime periods. The table provides the maximum measured diurnal $\delta^{18}O_X$ range per species.

[Figure]

**Fig. S5**. High temporal field measurements of normalized δ²H composition of xylem water
(β²Hₓ) of three *Abies alba* individuals (blue, branch samples) and three *Fagus sylvatica*
individuals (red, branch samples) sampled during a drought period in July 2017 in the "Freiamt"
field site in south-west Germany. Horizontal dark grey colored envelope delineates the
acceptable variance from the stem mean (i.e. 3‰) according to the standard assumption of no
variance along the length of a lignified plant. Light grey vertical envelopes mark the nighttime
periods.

[Figure]

**Fig. S6**. High temporal field measurements of normalized $\delta^{18}O$ composition of xylem water ($\beta^{18}O_X$) of three *Abies alba* individuals (blue, branch samples) and three *Fagus sylvatica* individuals (red, branch samples) sampled during a drought period in July 2017 in the "Freiamt" field site in south-west Germany. Horizontal dark grey colored envelope delineates the acceptable variance from the stem mean (i.e. 0.3‰) according to the standard assumption of no variance along the length of a lignified plant. Light grey vertical envelopes mark the nighttime periods.

[Figure]

**Fig S7:** Sap flow rate (*SF*, blue line), $\delta^2$H composition of xylem water at stem base ($\
[revised manuscript text omitted]
 RB, Canadell J, Ehleringer JR, Mooney HA, Sala OE, Schulze ED. 1996.** A global analysis of root distributions for terrestrial biomes. *Oecologia* **108**: 389–411.

**Leuschner C, Coners H, Icke R. 2004.** In situ measurement of water absorption by fine roots of three temperate trees: species differences and differential activity of superficial and deep roots. *Tree Physiology* **24**: 1359–1367.

**Meinzer FC, Goldstein G, Andrade JL. 2001**. Regulation of water flux through tropical forest canopy trees: do universal rules apply? *Tree physiology* **21**: 19–26.

**Meißner M, Köhler M, Schwendenmann L, Hölscher D. 2012.** Partitioning of soil water among canopy trees during a soil desiccation period in a temperate mixed forest. *Biogeosciences* **9**: 3465–3474.

**Rüdinger M, Hallgren SW, Steudle E, Schulze E-D. 1994.** Hydraulic and osmotic properties of spruce roots. *Journal of Experimental Botany* **45**: 1413–1425.

**Sands R, Fiscus EL, Reid CPP. 1982.** Hydraulic properties of pine and bean roots with varying degrees of suberization, vascular differentiation and mycorrhizal infection. *Functional Plant Biology* **9**: 559–569.

**Steudle E, Meshcheryakov AB. 1996.** Hydraulic and osmotic properties of oak roots. *Journal of Experimental Botany* **47**: 387–401.

**Zanne AE, Westoby M, Falster DS, Ackerly DD, Loarie SR, Arnold SEJ, Coomes DA. 2010.** Angiosperm wood structure: global patterns in vessel anatomy and their relation to wood density and potential conductivity. *American Journal of Botany* **97**: 207–215.